# The Bias-Variance Tradeoff in Data-Driven Optimization: A Local Misspecification Perspective

**Haixiang Lan[1]\*, Luofeng Liao[1]\*, Adam N. Elmachtoub[1],**
**Christian Kroer[1], Henry Lam[1], Haofeng Zhang[1,2]**
[1]Department of Industrial Engineering and Operations Research, Columbia University
[2]Morgan Stanley
{hl3725,ll3530,ae2516,ck2945,khl2114,hz2553}@columbia.edu

## Abstract

Data-driven stochastic optimization is ubiquitous in machine learning and operational decision-making problems. Sample average approximation (SAA) and model-based approaches such as estimate-then-optimize (ETO) or integrated estimation-optimization (IEO) are all popular, with model-based approaches being able to circumvent some of the issues with SAA in complex context-dependent problems. Yet the relative performance of these methods is poorly understood, with most results confined to the dichotomous cases of the model-based approach being either well-specified or misspecified. We develop the first results that allow for a more granular analysis of the relative performance of these methods under a local misspecification setting, which models the scenario where the model-based approach is nearly well-specified. By leveraging tools from contiguity theory in statistics, we show that there is a bias-variance tradeoff between SAA, IEO, and ETO under local misspecification, and that the relative importance of the bias and the variance depends on the degree of local misspecification. Moreover, we derive explicit expressions for the decision bias, which allows us to characterize (un)impactful misspecification directions, and provide further geometric understanding of the variance.

## 1 Introduction

Data-driven stochastic optimization arises ubiquitously in machine learning and operational decision-making problems. Generally, this problem takes the form $\operatorname{argmin}_{\boldsymbol{w}} \mathbb{E}_Q\left[c(\boldsymbol{w}, \boldsymbol{z})\right]$ where $\boldsymbol{w}$ represents the decision that we aim to optimize, and $\boldsymbol{z}$ is a random variable (or vector) drawn from an unknown distribution $Q$. The non-linear cost function $c$ is given and can represent a loss function in machine learning, financial portfolio losses, or resource allocation costs. In this paper, we focus on the setting where the expectation $\mathbb{E}_Q$ is unknown, but we observe data from $Q$.

A natural approach is to use empirical optimization, known as sample average approximation (SAA) [Shapiro et al., 2021], which approximates the unknown expectation with the empirical counterpart from the data. This approach is straightforward, but may not be suitable for complex scenarios in constrained and contextual optimization, when one needs to obtain a feature-dependent decision (i.e., a decision as a "function" of the features) and maintain feasibility [Hu et al., 2022]. In such cases, model-based approaches provide a workable alternative. A model-based approach fits a parametric distribution class to the data, say $\{P_{\boldsymbol{\theta}} : \boldsymbol{\theta} \in \Theta\}$, and this fitted distribution is then injected into the downstream optimization to obtain a decision. Just as in standard machine-learning problems, this parametrization helps maintain generalizability from supervised data.

---

\*Equal Contribution and Corresponding Authors.

39th Conference on Neural Information Processing Systems (NeurIPS 2025).

Our focus in this work is the statistical performance of model-based approaches for data-driven optimization with nonlinear cost objectives, as compared to SAA. More specifically, we study the question of *how to fit* the data into the parametric distribution models. There have been two major methods proposed in the literature: *Estimation-then-Optimize (ETO)* and *Integrated Estimation-Optimization (IEO)*. ETO separates the fitting step from the downstream optimization, by simply fitting $P_\theta$ via maximum likelihood estimation (MLE). IEO, on the other hand, integrates the downstream objective with the estimation process, by selecting the distribution parameter $\theta$ that minimizes the empirical expected cost. Conceptually, ETO can readily leverage existing machine learning tools and fits the model disjointly from the downstream decision task, while IEO attempts to take into account the downstream decision task (in many cases with a nontrivial additional computational expense). Intuitively, then, IEO should have better statistical performance than ETO in terms of the ultimate objective value of the chosen decision.

Our main goal is to dissect the bias-variance tradeoff between ETO, IEO, and also SAA, especially under the setting of *model misspecification*. In particular, our study reveals not only the variance arising from data noise, but also the bias of the resulting decision elicited under model misspecification. This allows us to gain insight on how the direction and amount of misspecification impacts decision quality. More concretely, a well-specified model means that in the estimation-optimization pipeline, the chosen parametric class $\{P_\theta : \theta \in \Theta\}$ contains the ground-truth distribution $Q$ – a case that is rarely seen in reality. In other words, model misspecification arises generically, the question only being by how much. Unfortunately, the theoretical understanding of the statistical performance among the various estimation-optimization approaches, especially in relation to this misspecification, has been rather limited. Elmachtoub et al. [2023] compare these approaches in large-sample regimes via stochastic dominance, but their analysis divides into the cases of well-specification and misspecification, each with different asymptotic scaling. Unfortunately, there is no smooth transition in between that captures the impact of varying the misspecification amount. Elmachtoub et al. [2025] attempt to address this issue by deriving finite-sample bounds that depend on the sample size and misspecification amount.

In this paper, we remedy the shortcomings in the above literature by leveraging the notion of *local misspecification*, originated from contiguity theory in statistics [Le Cam and Yang, 2000, Copas and Eguchi, 2001, Andrews et al., 2020], to derive large-sample results in relation to both the amount and *direction* of misspecification. Our results explicitly show the decision bias and variance, and its resulting regret, that arises from misspecification. This allows us to smoothly compare ETO, IEO and SAA. We show the following results. When model misspecification is severe relative to the data noise level, SAA performs better than IEO, and IEO performs better than ETO, in terms of both bias and regret. This matches the intuition described previously that IEO should outperform ETO by integrating the estimation-optimization pipeline. On the other hand, when the misspecification amount is mild, the performance ordering is reversed, which generalizes previous similar findings in Elmachtoub et al. [2023] that focused on zero misspecification. Most importantly, in the most relevant case where the misspecification is roughly similar to the data noise, which we call the *balanced* case, the ordering of the methods exhibits a bias-variance tradeoff: SAA performs the best on bias, whereas ETO performs the best on variance, and IEO is in the middle for both metrics. This defies a universal performance ordering, but also points to the need for a deep understanding of the characteristics of the bias term in relation to the misspecification direction. Table 1 summarizes our performance ordering findings.

|  | mild ($\alpha > 1/2$) | | | balanced ($\alpha = 1/2$) | | | severe ($0 < \alpha < 1/2$) | | |
|---|---|---|---|---|---|---|---|---|---|
|  | bias | variance | regret | bias | variance | regret | bias | variance | regret |
| ETO | $\approx 0$ | best | best | worst | best | depends | worst | $\approx 0$ | worst |
| IEO | $\approx 0$ | middle | middle | middle | middle | depends | middle | $\approx 0$ | middle |
| SAA | $\approx 0$ | worst | worst | best ($\approx 0$) | worst | depends | best ($\approx 0$) | $\approx 0$ | best |

Table 1: Summary of our results on performance orderings. "$\approx 0$" means asymptotically negligible. $\alpha$ is a parameter that signals the misspecification amount relative to the data noise level and will be detailed later.

Our next contribution is to provide an explicit formula for the bias attributed to model misspecification, which allows us to gain insights into how the bias is impacted by the misspecification direction. We went beyond the classical local minimax theory by showing the *non-regularity* of the ETO and

IEO estimators, which is rarely seen in standard statistical literature [van der Vaart, 2000]. In the severely misspecified regime, where there is no available contiguity theory tools, we develop a novel technique to characterize and compare the asymptotics of the three estimators. We further identify sufficient conditions on *approximately impactless* misspecification directions – model misspecification directions that result in bias that is first-order negligible compared to the data noise variance. In general, this direction is orthogonal to the difference between the influence functions of solutions obtained from the considered estimation-optimization pipeline and SAA. Here, influence functions are interpreted as the gradients with respect to the underlying data distribution, and they appear not only in the bias but also variance comparisons. This characterization in particular suggests how SAA is always (and naturally) the best in terms of bias (see Table 1), but also how the biases for IEO and ETO magnify when the obtained solution has a different influence function from that of SAA. Moreover, to enhance the transparency of our characterization, we show that a sufficient condition for being approximately impactless is to be in the linear span of the score function of the parametric model. While this latter condition is imposed purely on the parametric model (i.e., not the downstream optimization), it already shows the intriguing phenomenon where model misspecification could be insignificant in impacting the performance of the ultimate decision.

## 1.1 Related Works

**Data-Driven Stochastic Optimization.** Data-driven optimization, a cornerstone in machine learning and operations research, addresses problems where decision are informed from optimization problems with parameters or distributions learned from data. Existing popular methods include SAA [Shapiro et al., 2021] and distributionally robust optimization (DRO) [Delage and Ye, 2010]. Recently, there has been a growing interest in an integrated framework that combines predictive modeling of unknown parameters with downstream optimization tasks [Kao et al., 2009, Donti et al., 2017]. When the cost function is linear, Elmachtoub and Grigas [2022] propose a "Smart-Predict-then-Optimize" (SPO) approach that integrates prediction with optimization to improve decision-making. Recent literature explore further properties of the SPO approach [Mandi et al., 2020, Blondel et al., 2020, Ho-Nguyen and Kılınç-Karzan, 2022, Liu and Grigas, 2022, Liu et al., 2023, El Balghiti et al., 2023]. Hu et al. [2022] further compare the performances of different data-driven approaches in the linear cost function setting. In the context of non-linear cost functions, Grigas et al. [2021] propose an integrated approach tailored to discrete distributions, and Lam [2021] compares SAA with DRO and Bayesian extensions.

**Local Misspecification.** Model misspecification is extensively studied in the statistics and econometrics and machine learning literature [Marsden et al., 2021]. In this paper we focus on local misspecification, where the magnitude of misspecification vanishes as the sample size grows. Newey [1985] analyzes the asymptotic power properties of the generalized method of moments tests under a sequence of local misspecified alternatives. Kang and Schafer [2007] design a doubly robust procedure to estimate the population mean under the local misspecified models with incomplete data. Copas and Eguchi [2001, 2005] discuss the impact of local misspecification on the sensitivity of likelihood-based statistical inference under the asymptotic framework. Local misspecification is also discussed in robust estimation [Kitamura et al., 2013, Armstrong et al., 2023], causal inference [Conley et al., 2012, Fan et al., 2022], econometrics [Bugni et al., 2012, Andrews et al., 2017, 2020, Bugni and Ura, 2019, Armstrong and Kolesár, 2021, Bonhomme and Weidner, 2022, Candelaria and Zhang, 2024] and reinforcement learning [Dong et al., 2023]. To the best of our knowledge, we are the first to study the impact of local misspecification in data-driven optimization.

## 2 Settings and Methodologies

## 2.1 Data-Driven Stochastic Optimization

Consider a data-driven optimization problem in the following form:

$$\boldsymbol{w}^* = \underset{\boldsymbol{w} \in \Omega}{\operatorname{argmin}} \left\{ v_0(\boldsymbol{w}) := \mathbb{E}_Q \left[ c(\boldsymbol{w}, \boldsymbol{z}) \right] \right\} \tag{1}$$

where $\Omega$ is an open subset in $\mathbb{R}^{d_w}$, $\boldsymbol{z} \in \mathcal{Z} \subset \mathbb{R}^{d_z}$ is the uncertain parameter with unknown data-generating distribution $Q$, $c(\cdot, \cdot)$ is a known *non-linear* cost function, and $v_0(\cdot)$ is the expectation of the cost function under ground-truth distribution $Q$ under a decision $\boldsymbol{w}$. We are given independent

and identically distributed (i.i.d.) data $\boldsymbol{z}_1, ..., \boldsymbol{z}_n$ drawn from $Q$, and the goal is to approximate the optimal decision $\boldsymbol{w}^*$ using the data.

In model-based approaches, we use a parametric distribution family $\{P_{\boldsymbol{\theta}}, \boldsymbol{\theta} \in \Theta\}$ where $\boldsymbol{\theta} \in \Theta \subset \mathbb{R}^{d_\theta}$ is the model parameter and $\Theta$ is an open subset of $\mathbb{R}^{d_\theta}$. To explain further, we define the *oracle* solution $\boldsymbol{w}_{\boldsymbol{\theta}}$ by

$$\boldsymbol{w}_{\boldsymbol{\theta}} \in \operatorname*{argmin}_{\boldsymbol{w} \in \Omega} \{v(\boldsymbol{w}, \boldsymbol{\theta}) := \mathbb{E}_{P_{\boldsymbol{\theta}}}[c(\boldsymbol{w}, \boldsymbol{z})]\}, \tag{2}$$

where $v(\boldsymbol{w}, \boldsymbol{\theta})$ is the expected cost function under the distribution $P_{\boldsymbol{\theta}}$. Depending on the choice of the model, the ground-truth distribution $Q$ may or may not be in the parametric family $\{P_{\boldsymbol{\theta}} : \boldsymbol{\theta} \in \Theta\}$. We say $\{P_{\boldsymbol{\theta}} : \boldsymbol{\theta} \in \Theta\}$ is well-specified if there exists $\boldsymbol{\theta}_0 \in \boldsymbol{\theta}$ such that $P_{\boldsymbol{\theta}_0} = Q$. We say $\{P_{\boldsymbol{\theta}} : \boldsymbol{\theta} \in \Theta\}$ is misspecified if it is not well-specified.

**Notations.** We denote $\mathbb{E}_{\tilde{P}}[\cdot]$ and $\operatorname{var}_{\tilde{P}}$ as the expectation and variance under the distribution $\tilde{P}$. We denote $\mathbb{E}_{\boldsymbol{\theta}}[\cdot]$ as $\mathbb{E}_{P_{\boldsymbol{\theta}}}[\cdot]$ and $\operatorname{var}_{\boldsymbol{\theta}}(\cdot) = \operatorname{var}_{P_{\boldsymbol{\theta}}}(\cdot)$ in the parametric case. For a symmetric matrix $\boldsymbol{A}$, we write $\boldsymbol{A} \geq \boldsymbol{0}$ if it is positive semi-definite and $\boldsymbol{A} > \boldsymbol{0}$ if it is positive definite. For two symmetric matrices $\boldsymbol{A}_1$ and $\boldsymbol{A}_2$, we write $\boldsymbol{A}_1 \geq \boldsymbol{A}_2$ if $\boldsymbol{A}_1 - \boldsymbol{A}_2 \geq \boldsymbol{0}$ and $\boldsymbol{A}_1 > \boldsymbol{A}_2$ if $\boldsymbol{A}_1 - \boldsymbol{A}_2 > \boldsymbol{0}$. For a matrix $\boldsymbol{A} \in \mathbb{R}^{m \times n}$, we define the column span of $\boldsymbol{A}$ as $\operatorname{col}(\boldsymbol{A}) = \{\boldsymbol{A}\boldsymbol{x} : \boldsymbol{x} \in \mathbb{R}^n\}$. For a vector $\boldsymbol{x} \in \mathbb{R}^n$ and a positive semi-definite matrix $\boldsymbol{A} \in \mathbb{R}^{n \times n}$, we define the matrix-induced norm $\|\boldsymbol{x}\|_{\boldsymbol{A}} := \sqrt{\boldsymbol{x}^\top \boldsymbol{A} \boldsymbol{x}}$. We define $\xrightarrow{P^n}$ as convergence in distribution under the measure $P^n$. Precisely, $X_n \xrightarrow{P^n} X$ if $P^n(X_n \leq t) \to \mathbb{P}(X \leq t)$ for all continuous points of the distribution of $X$ where $\mathbb{P}$ denotes the distribution of $X$. For a sequence of random variables $\{X_n\}_{n=1}^\infty$, we say $X_n = O_{P^n}(1)$ if it is stochastically bounded under the probability measure $P^n$, i.e., for all $\delta > 0$ there exists $M$ and $N \in \mathbb{N}$ such that for all $n \geq N$, $P^n(|X_n| > M) \leq \delta$. We say $X_n = o_{P^n}(1)$ if it converges to zero in probability under the probability measure $P^n$, i.e., $X_n = o_{P^n}(1)$ if for all $\varepsilon > 0$, $\lim_{n \to \infty} P^n(|X_n| > \varepsilon) = 0$. More generally, we denote $X_n = O_{P^n}(a_n)$ if $X_n/a_n = O_{P^n}(1)$ and we denote $X_n = o_{P^n}(a_n)$ if $X_n/a_n = o_{P^n}(1)$. For two random variables $X_1$, $X_2$ with distribution $\mathbb{P}_1$ and $\mathbb{P}_2$, we say $X_1$ is (first-order) stochastically dominated by $X_2$, denoted as $X_1 \preceq_{\mathrm{st}} X_2$, if for all $x \in \mathbb{R}$, it satisfies that $\mathbb{P}_1(X_1 > x) \leq \mathbb{P}_2(X_2 > x)$.

## 2.2 Three Data-Driven Methods

We consider three popular approaches for solving (1) in a data-driven fashion.

*Sample Average Approximation (SAA).* SAA simply replaces $\mathbb{E}_Q$ in (1) with its empirical counterpart. More precisely, we solve $\hat{\boldsymbol{w}}^{\mathrm{SAA}} := \operatorname{argmin}_{\boldsymbol{w} \in \Omega} \{\hat{v}_0(\boldsymbol{w}) := \frac{1}{n} \sum_{i=1}^n c(\boldsymbol{w}, \boldsymbol{z}_i)\}$ .

*Estimate-Then-Optimize (ETO).* ETO uses maximum likelihood estimation (MLE) to infer $\boldsymbol{\theta}$ by solving $\hat{\boldsymbol{\theta}}^{\mathrm{ETO}} = \sup_{\boldsymbol{\theta} \in \Theta} \frac{1}{n} \sum_{i=1}^n \log p_{\boldsymbol{\theta}}(\boldsymbol{z}_i)$ Here $p_{\boldsymbol{\theta}}$ is the probability density or mass function. Then we plug $\hat{\boldsymbol{\theta}}^{\mathrm{ETO}}$ into (2): $\hat{\boldsymbol{w}}^{\mathrm{ETO}} := \boldsymbol{w}_{\hat{\boldsymbol{\theta}}^{\mathrm{ETO}}} = \operatorname{argmin}_{\boldsymbol{w} \in \Omega} v(\boldsymbol{w}, \boldsymbol{\theta}^{\mathrm{ETO}})$.

*Integrated-Estimation-Optimization (IEO).* IEO selects the $\boldsymbol{\theta}$ that performs best on the empirical cost function $\hat{v}_0(\cdot)$ evaluated at $\boldsymbol{w}_{\boldsymbol{\theta}}$. More precisely, we solve $\inf_{\boldsymbol{\theta} \in \Theta} \hat{v}_0(\boldsymbol{w}_{\boldsymbol{\theta}})$ and get a solution $\hat{\boldsymbol{\theta}}^{\mathrm{IEO}}$. Then we use the plug-in estimator $\hat{\boldsymbol{w}}^{\mathrm{IEO}} := \boldsymbol{w}_{\hat{\boldsymbol{\theta}}^{\mathrm{IEO}}} = \operatorname{argmin}_{\boldsymbol{w} \in \Omega} v(\boldsymbol{w}, \boldsymbol{\theta}^{\mathrm{IEO}})$.

Among the three methods, SAA is model-free while ETO and IEO are model-based. ETO separates estimation (via MLE) with downstream optimization, while IEO integrates the latter into the estimation process. Our primary focus is to statistically compare these three data-to-decision pipelines in the so-called locally misspecified regime, which we discuss next.

Throughout the paper we assume certain classical technical assumptions on the cost $c$ and the distribution $P_{\boldsymbol{\theta}}$ to ensure the asymptotic normality of certain $M$-estimators under $P_{\boldsymbol{\theta}_0}$ and the Cramer-Rao lower bound. In particular, they require relevant population minimizers of $\min_{\boldsymbol{w}} \mathbb{E}_{\boldsymbol{\theta}_0}[c(\boldsymbol{w}, \boldsymbol{z})]$, $\min_{\boldsymbol{\theta}} \mathbb{E}_{\boldsymbol{\theta}_0}[-\log p_{\boldsymbol{\theta}}(\boldsymbol{z})]$ and $\min_{\boldsymbol{\theta}} \mathbb{E}_{\boldsymbol{\theta}_0}[c(\boldsymbol{w}_{\boldsymbol{\theta}}, \boldsymbol{z})]$ are uniquely attained in the interior of the parameter spaces. See Assumptions 3 and 4 in the Appendix for precise statements.

## 2.3 Local Misspecification

We first explain local misspecification intuitively before providing formal definitions. Recall the ground-truth data generating distribution $Q$, and our parametric distribution family (i.e., model)

$\{P_{\boldsymbol{\theta}} : \boldsymbol{\theta} \in \Theta\}$. At a high level, we assume that for a finite data size $n$, $Q$ could deviate from the model in a certain "direction". However, as $n$ is sufficiently large, we expect to have a more accurate model, and $Q$ will approach a distribution in $\{P_{\boldsymbol{\theta}} : \boldsymbol{\theta} \in \Theta\}$, which we denote as $P_{\boldsymbol{\theta}_0}$ for some $\boldsymbol{\theta}_0 \in \Theta$. In other words, $\{P_{\boldsymbol{\theta}} : \boldsymbol{\theta} \in \Theta\}$ is misspecified to $Q$ in a "local" sense, and such misspecification will ultimately vanish. From now on, we use $P_{\boldsymbol{\theta}_0}$ to represent this distribution. Note that neither $P_{\boldsymbol{\theta}_0}$ nor $\boldsymbol{\theta}_0$ are known in practice.

To introduce the local misspecification regime, we first formally define a notion called local perturbation for describing the deviation between two distributions. We work with a general form of local perturbation [van der Vaart, 2000, Fan et al., 2022, Duchi and Ruan, 2021, Duchi, 2021] where the ground truth distribution $Q$ is related to $P_{\boldsymbol{\theta}_0}$ through a general tilt the distribution (we will provide many classical examples in Appendix A.2):

**Definition 1** (Local Perturbation). Consider a scalar function $u(\boldsymbol{z}) : \mathcal{Z} \to \mathbb{R}$ with zero mean $\mathbb{E}_{\boldsymbol{\theta}_0}[u] = 0$ and finite second order moment $\mathbb{E}_{\boldsymbol{\theta}_0}[u^2]$. We define a tilted distribution $Q_t$ for $t \in \mathbb{R}$ with probability density (mass) function $q_t$ with respect to the dominated measure (note that $Q_t$ is not necessarily in the parametric family $\{P_{\boldsymbol{\theta}} : \boldsymbol{\theta} \in \Theta\}$) with $q_0 = p_{\boldsymbol{\theta}_0}$. We further assume for all $t$, $q_t$ is differentiable for almost every $\boldsymbol{z}$, as well as the quadratic mean differentiability condition:

$$\int \left( \sqrt{q_t} - \sqrt{p_{\boldsymbol{\theta}_0}} - \frac{1}{2} t u \sqrt{p_{\boldsymbol{\theta}_0}} \right)^2 d\boldsymbol{z} = o(t^2).$$

Note that when $t = 0$, $q_0 = p_{\boldsymbol{\theta}_0}$.

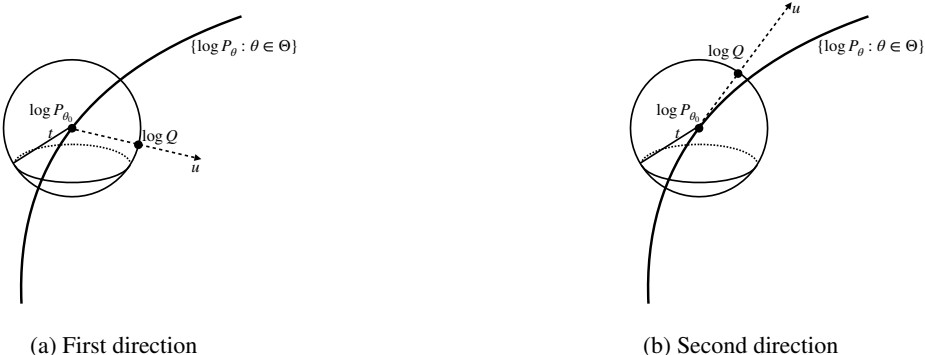

(a) First direction          (b) Second direction

Figure 1: Local Misspecification

The local perturbation in Definition 1 is standard in classical asymptotic statistics [van der Vaart, 2000] and consists of two crucial elements, the scalar function $u(\boldsymbol{z})$ and the real value $t$. Intuitively, we can think of $t$ as the degree of perturbation and the function $u(\boldsymbol{z})$ as a certain direction of perturbation. Figure 1 presents such a geometric interpretation. The parametric family $\{P_{\boldsymbol{\theta}} : \boldsymbol{\theta} \in \Theta\}$ could be viewed as a "curve" in the distribution space. Each point on the curve corresponds to a distribution in the parametric family. If we fix the value $t$ and let the direction $u(\boldsymbol{z})$ range over all possible directions, then $Q$ will lie within a neighborhood of radius $t$ around this curve. In this sense, the perturbation quantity $tu(\boldsymbol{z})$ acts like the "vector" pointing from $P_{\boldsymbol{\theta}_0}$ to $Q$ where the "length" is $t$ and the "direction" is given by the vector $u(\boldsymbol{z})$. Below we will discuss the "local" case where the radius $t$ vanishes as the sample size goes to infinity. In particular, in Figure 1 (b), the "direction" $u(\boldsymbol{z})$ is tangent to the curve. We will discuss this special case in the latter part of this paper (Theorem 4).

Local misspecification refers to the situation where, as the sample size $n$ increases, the sequence $Q_t$, where $t$ depends on $n$, approaches the model. When the ground-truth $Q_t$ lies outside the parametric family, this misspecification adds errors to the standard inference errors on the order $\Theta(1/\sqrt{n})$, and when these two error levels coincide at the same order, we call this scenario *balanced* misspecification. More broadly, we consider local misspecification with $t = \Theta(1/n^\alpha)$ where $\alpha \in (0, \infty)$, which leads to the following definition. For $n$ i.i.d. data $\{\boldsymbol{z}_i\}_{i=1}^n$ sampled from $Q$, let $Q^n := Q^{\otimes n}$ be the $n$-fold product measure of $Q$ denoting their joint distribution, and analogously for $P_{\boldsymbol{\theta}_0}$.

**Definition 2** (Three Local Misspecification Regimes). Let $P^n = P_{\boldsymbol{\theta}_0}^{\otimes n}$. The tilted distribution $Q_t$ is defined in Definition 1. Suppose $\alpha > 0$ and the joint distribution of the $n$ i.i.d. data is $Q^n := Q_{1/n^\alpha}^{\otimes n}$.

1. (Mild). We call the case when $\alpha > 1/2$ the mildly misspecified regime.

2. (Balanced). We call the case when $\alpha = 1/2$ the balanced misspecified regime.

3. (Severe). We call the case when $0 < \alpha < 1/2$ the severely misspecified regime.

We note that the local misspecification regime described above should not be taken literally to imply that the data-generating process depends on the sample size. Instead, it is an information theoretic device to analyze and compare the local behavior of estimators in situations where the influence of misspecification is comparable to the order of the statistical error. In other words, we are interested in the more realistic setting where both misspecification and statistical error are small at a similar level, instead of assuming vanishing statistical error but fixed misspecification as in previous work [Elmachtoub et al., 2023, 2025].

## 3 Main Results

We derive theoretical results to compare the asymptotic performances of the three methods, SAA, ETO, and IEO, that encompass the three local misspecification regimes in Definition 2. We first list out several standard assumptions. We define $s_{\boldsymbol{\theta}_0}(\boldsymbol{z}) = \nabla_{\boldsymbol{\theta}} \log p_{\boldsymbol{\theta}_0}(\boldsymbol{z})$ as the score function at $\boldsymbol{\theta}_0$ mapping from $\mathcal{Z} \to \mathbb{R}^{d_\theta}$. Recall that $v(\boldsymbol{w}, \boldsymbol{\theta}) = \int c(\boldsymbol{w}, \boldsymbol{z}) p_{\boldsymbol{\theta}}(\boldsymbol{z}) d\boldsymbol{z}$.

**Assumption 1** (Smoothness). *Assume that*

1. *The function $v(\boldsymbol{w}, \boldsymbol{\theta})$ is twice continuously differentiable with respect to $(\boldsymbol{w}, \boldsymbol{\theta})$ at $(\boldsymbol{w}_{\boldsymbol{\theta}_0}, \boldsymbol{\theta}_0)$ with a Hessian matrix, denoted by $\begin{bmatrix} \boldsymbol{V} & \boldsymbol{\Sigma} \\ \boldsymbol{\Sigma}^\top & * \end{bmatrix}$, where $*$ denotes a matrix that is not of interest. Assume $\boldsymbol{\Sigma} \in \mathbb{R}^{d_w \times d_\theta}$ is full-rank and $\boldsymbol{V} \in \mathbb{R}^{d_w \times d_w}$ is invertible.*

2. *The function $\boldsymbol{\theta} \mapsto \boldsymbol{w}_{\boldsymbol{\theta}}$ is well-defined on a neighborhood of $\boldsymbol{\theta}_0$, twice continuously differentiable at $\boldsymbol{\theta}_0$ with a full-rank gradient matrix $\nabla_{\boldsymbol{\theta}} \boldsymbol{w}_{\boldsymbol{\theta}}|_{\boldsymbol{\theta}=\boldsymbol{\theta}_0} \in \mathbb{R}^{d_\theta \times d_w}$.*

3. *The Fisher information matrix $\boldsymbol{I} := \mathbb{E}_{\boldsymbol{\theta}_0}[s_{\boldsymbol{\theta}_0}(\boldsymbol{z}) s_{\boldsymbol{\theta}_0}(\boldsymbol{z})^\top] \in \mathbb{R}^{d_\theta \times d_\theta}$ is well-defined and invertible.*

Note that the matrices above are fixed quantities and are not related to whether the model is well-specified or misspecified. These matrices are critical for characterizing the sensitivity of the target stochastic optimization problem. We also define $\boldsymbol{\Phi} = \nabla^2_{\boldsymbol{\theta}_1 \boldsymbol{\theta}_1} \int c(\boldsymbol{w}_{\boldsymbol{\theta}_1}, \boldsymbol{z}) p_{\boldsymbol{\theta}_0}(\boldsymbol{z}) d\boldsymbol{z}|_{\boldsymbol{\theta}_1 = \boldsymbol{\theta}_0}$. The following lemma provides closed-form expressions for the gradient $\nabla_{\boldsymbol{\theta}} \boldsymbol{w}_{\boldsymbol{\theta}}$ and matrices $\boldsymbol{\Sigma}, \boldsymbol{\Phi}$.

**Lemma 1.** *Under Assumption 1, it holds that*

$$\nabla_{\boldsymbol{\theta}} \boldsymbol{w}_{\boldsymbol{\theta}}|_{\boldsymbol{\theta}=\boldsymbol{\theta}_0} = -\boldsymbol{\Sigma}^\top \boldsymbol{V}^{-1}, \quad \boldsymbol{\Sigma} = \mathbb{E}_{\boldsymbol{\theta}_0}\left[\nabla_{\boldsymbol{w}} c(\boldsymbol{w}_{\boldsymbol{\theta}_0}, \boldsymbol{z}) s_{\boldsymbol{\theta}_0}(\boldsymbol{z})^\top\right], \quad \boldsymbol{\Phi} = \boldsymbol{\Sigma}^\top \boldsymbol{V}^{-1} \boldsymbol{\Sigma}.$$

Next, we introduce the influence function which is a key ingredient in our derived formulas. Originating from robust statistics [Hampel, 1974], it is the functional derivative of an estimator with respect to the data distribution. In our context, this refers to the derivative of the *decision* obtained from the estimation-optimization pipeline. Specifically, the influence functions for SAA, IEO and ETO are respectively

$$\mathrm{IF}^{\mathrm{SAA}}(\boldsymbol{z}) = -\boldsymbol{V}^{-1} \nabla_{\boldsymbol{w}} c(\boldsymbol{w}_{\boldsymbol{\theta}_0}, \boldsymbol{z}),$$
$$\mathrm{IF}^{\mathrm{IEO}}(\boldsymbol{z}) = \boldsymbol{V}^{-1} \boldsymbol{\Sigma} \boldsymbol{\Phi}^{-1} \nabla_{\boldsymbol{\theta}} c(\boldsymbol{w}_{\boldsymbol{\theta}_0}, \boldsymbol{z}),$$
$$\mathrm{IF}^{\mathrm{ETO}}(\boldsymbol{z}) = -\boldsymbol{V}^{-1} \boldsymbol{\Sigma} \boldsymbol{I}^{-1} s_{\boldsymbol{\theta}_0}(\boldsymbol{z}),$$

all of which are $\mathbb{R}^{d_w}$-valued. Regarding notations, $\nabla_{\boldsymbol{w}} c(\boldsymbol{w}_{\boldsymbol{\theta}_0}, \boldsymbol{z})$ is the gradient of the map $\boldsymbol{w} \mapsto c(\boldsymbol{w}, \boldsymbol{z})$ at $\boldsymbol{w} = \boldsymbol{w}_{\boldsymbol{\theta}_0}$, and $\nabla_{\boldsymbol{\theta}} c(\boldsymbol{w}_{\boldsymbol{\theta}_0}, \boldsymbol{z})$ is the gradient of the map $\boldsymbol{\theta} \mapsto c(\boldsymbol{w}_{\boldsymbol{\theta}}, \boldsymbol{z})$ at $\boldsymbol{\theta} = \boldsymbol{\theta}_0$.

Finally, we introduce *regret* as the criterion to evaluate the quality of a decision $\boldsymbol{w}$. Since we conduct local asymptotic analysis, the definition of regret is slightly different from the classical definition in asymptotic or finite-sample analysis [Lam, 2021, Elmachtoub and Grigas, 2022], as it needs to account for the changing sample size. We define $v_n$ and $\boldsymbol{w}_n^*$ as follows:

$$\boldsymbol{w}_n^* := \operatorname*{argmin}_{\boldsymbol{w} \in \Omega} v_n(\boldsymbol{w}) := \mathbb{E}_{Q^n}\left[\frac{1}{n} \sum_{i=1}^n c(\boldsymbol{w}, \boldsymbol{z}_i)\right].$$

In the local misspecification setting, when the sample size is $n$, the data distribution is given by $Q^n$. Hence, $v_n(\boldsymbol{w})$ represents the ground-truth expected cost, and $\boldsymbol{w}_n^*$ is interpreted as the corresponding optimal solution at sample size $n$.

**Definition 3** (Regret). For any distribution $Q^n$ and any $\boldsymbol{w} \in \Omega$, the regret of $\boldsymbol{w}$ at sample size $n$ is defined as
$$R_{Q^n}(\boldsymbol{w}) = v_n(\boldsymbol{w}) - v_n(\boldsymbol{w}_n^*).$$

In the rest of this section, we conduct a comprehensive analysis on the regrets using the three estimation-optimization methods, for the three misspecification regimes introduced in Definition 2.

## 3.1 Balanced Misspecification

To state our main results, we define, for $\square \in \{\text{SAA}, \text{IEO}, \text{ETO}\}$,
$$N^\square := N(0, \text{var}(\text{IF}^\square(\boldsymbol{z}))),$$
$$\boldsymbol{b}^\square := \mathbb{E}_{\boldsymbol{\theta}_0}[u(\boldsymbol{z})(\text{IF}^\square(\boldsymbol{z}) - \text{IF}^{\text{SAA}}(\boldsymbol{z}))] \in \mathbb{R}^{d_w},$$
$$R^\square := \tfrac{1}{2}\left\| \boldsymbol{b}^\square \right\|_{\boldsymbol{V}}^2 \in \mathbb{R},$$

where $N^\square$ is the normal distribution with zero mean and covariance matrix $\text{var}(\text{IF}^\square(\boldsymbol{z}))$. Note that unless otherwise specified, $\text{var}(\cdot)$ should always be interpreted as $\text{var}_{P_{\boldsymbol{\theta}_0}}(\cdot)$.

**Theorem 1** (Asymptotics under Balanced Misspecification). *Suppose Assumptions 1, 3, and 4 hold. In the balanced regime in Definition 2, under $Q^n$, for $\square \in \{\text{SAA}, \text{ETO}, \text{IEO}\}$,*
$$\sqrt{n}(\hat{\boldsymbol{w}}^\square - \boldsymbol{w}_n^*) \overset{Q^n}{\to} N^\square + \boldsymbol{b}^\square,$$
$$nR_{Q^n}(\hat{\boldsymbol{w}}^\square) \overset{Q^n}{\to} \frac{1}{2}\left(N^\square + \boldsymbol{b}^\square\right)^\top \boldsymbol{V} \left(N^\square + \boldsymbol{b}^\square\right).$$
*In terms of bias,* $0 = \left\|\boldsymbol{b}^{\text{SAA}}\right\|_{\boldsymbol{V}} \leq \left\|\boldsymbol{b}^{\text{IEO}}\right\|_{\boldsymbol{V}} \leq \left\|\boldsymbol{b}^{\text{ETO}}\right\|_{\boldsymbol{V}}$. *In terms of variance,* $\text{var}\left(N^{\text{SAA}}\right) \geq \text{var}\left(N^{\text{IEO}}\right) \geq \text{var}\left(N^{\text{ETO}}\right)$.

Theorem 1 states that when $\alpha = 1/2$, i.e., the degree of misspecification is of the same order as the statistical error, the gap between the data-driven and optimal decisions is asymptotically normal. Moreover, this normal has mean zero for SAA (note $\boldsymbol{b}^{\text{SAA}} = \boldsymbol{0}$ and $R^{\text{SAA}} = 0$), but generally nonzero for ETO and IEO. More importantly, in the asymptotic limit, $\boldsymbol{b}^\square$ represents the bias coming from model misspecification as it involves $u$, while $N^\square$ captures the data noise variability. We highlight that the dependence on $u$ in $\boldsymbol{b}$ directly implies that the ETO and IEO estimator are *non-regular* in the sense of van der Vaart [2000]. The theorem shows that in terms of bias, SAA generally outperforms IEO which in turn outperforms ETO. On the the hand, the ordering is reversed for variance. As a result there is no universal ordering for the overall error in general. The next theorem lifts further to compare the regrets of SAA, ETO and IEO under the balanced regime.

**Theorem 2** (Regret Comparisons under Balanced Misspecification). *Let*
$$\mathbb{G}^\square := \frac{1}{2}(N^\square + \boldsymbol{b}^\square)^\top \boldsymbol{V} \left(N^\square + \boldsymbol{b}^\square\right)$$
*denote the limiting regret distribution of* $\square \in \{\text{SAA}, \text{ETO}, \text{IEO}\}$ *in Theorem 1. Then* $\mathbb{E}[\mathbb{G}^\square] = \mathbb{E}[\frac{1}{2}(N^\square)^\top \boldsymbol{V} N^\square] + \frac{1}{2}(\boldsymbol{b}^\square)^\top \boldsymbol{V} \boldsymbol{b}^\square$. *Moreover,*
$$\mathbb{E}\left[\frac{1}{2}\left(N^{\text{ETO}}\right)^\top \boldsymbol{V} N^{\text{ETO}}\right] \leq \mathbb{E}\left[\frac{1}{2}\left(N^{\text{IEO}}\right)^\top \boldsymbol{V} N^{\text{IEO}}\right] \leq \mathbb{E}\left[\frac{1}{2}\left(N^{\text{SAA}}\right)^\top \boldsymbol{V} N^{\text{SAA}}\right],$$
$$\frac{1}{2}\left(\boldsymbol{b}^{\text{SAA}}\right)^\top \boldsymbol{V} \boldsymbol{b}^{\text{SAA}} \leq \frac{1}{2}\left(\boldsymbol{b}^{\text{IEO}}\right)^\top \boldsymbol{V} \boldsymbol{b}^{\text{IEO}} \leq \frac{1}{2}\left(\boldsymbol{b}^{\text{ETO}}\right)^\top \boldsymbol{V} \boldsymbol{b}^{\text{ETO}}.$$

Like Theorem 1, while Theorem 2 shows a lack of universal ordering for regrets, it depicts a decomposition of the asymptotic distribution of the regret into two parts where two opposite orderings emerge. In particular, it suggests that while ETO is best in terms of variance, and SAA best in terms of bias, IEO is in between and could potentially induce a lower decision error compared to the other two methods.

Another important insight from Theorems 1 and 2 regards the explicit form of the bias and variance. For this, let us introduce the analogous results for severe and mild misspecification regimes and discuss the formulas along the way.

## 3.2 Severe Misspecification

We first formally describe the $O(1/\sqrt{n})$ order of the statistical error via the following assumptions borrowed from Fang et al. [2023]. The assumption is natural because it says the empirical part deviates from the expected part at the rate $O(1/\sqrt{n})$. .

**Assumption 2** (Statistical Error Order). *For i.i.d.* $\{z_i\}_{i=1}^n$ *with joint distribution* $Q^n$, *let*

$$\boldsymbol{\theta}_n^{\mathrm{KL}} := \underset{\boldsymbol{\theta} \in \Theta}{\operatorname{argmax}}\, \mathbb{E}_{Q^n} \left[ \frac{1}{n} \sum_{i=1}^n \log p_{\boldsymbol{\theta}}(\boldsymbol{z}_i) \right],$$

$$\boldsymbol{\theta}_n^* := \underset{\boldsymbol{\theta} \in \Theta}{\operatorname{argmin}}\, \mathbb{E}_{Q^n} \left[ \frac{1}{n} \sum_{i=1}^n c(\boldsymbol{w}_{\boldsymbol{\theta}}, \boldsymbol{z}_i) \right].$$

*Assume that* $\left\| \hat{\boldsymbol{w}}^{\mathrm{ETO}} - \boldsymbol{w}_{\boldsymbol{\theta}_n^{\mathrm{KL}}} \right\|$, $\left\| \hat{\boldsymbol{w}}^{\mathrm{IEO}} - \boldsymbol{w}_{\boldsymbol{\theta}_n^*} \right\|$ *and* $\left\| \hat{\boldsymbol{w}}^{\mathrm{SAA}} - \boldsymbol{w}_n^* \right\|$ *are all of order* $O_{Q^n}(1/\sqrt{n})$. *Moreover, assume that the matrix* $\nabla^2_{\boldsymbol{w}\boldsymbol{w}} v_n(\boldsymbol{w}_n^*) \to \boldsymbol{V}$ *as* $n \to \infty$.

**Theorem 3** (Asymptotics under Severe Misspecification). *Suppose Assumptions 1, 2, 3 and 4 hold. In the severely misspecified case, under* $Q^n$, *for* $\square \in \{\mathrm{SAA}, \mathrm{ETO}, \mathrm{IEO}\}$,

$$n^\alpha (\hat{\boldsymbol{w}}^\square - \boldsymbol{w}_n^*) \xrightarrow{p} \boldsymbol{b}^\square,$$

$$n^{2\alpha} R_{Q^n}(\hat{\boldsymbol{w}}^\square) \xrightarrow{p} R^\square = \frac{1}{2} \|\boldsymbol{b}^\square\|_{\boldsymbol{V}}^2.$$

*In terms of variance,* $0 = \mathrm{var}\left(\boldsymbol{b}^{\mathrm{SAA}}\right) = \mathrm{var}\left(\boldsymbol{b}^{\mathrm{IEO}}\right) = \mathrm{var}\left(\boldsymbol{b}^{\mathrm{ETO}}\right)$. *The comparison of bias has the same form as the regret (stated in the next theorem).*

**Theorem 4** (Bias/Regret Comparisons under Severe Misspecification). *Under the same setting as in Theorem 3, we have* $0 = \left\|\boldsymbol{b}^{\mathrm{SAA}}\right\|_{\boldsymbol{V}} \le \left\|\boldsymbol{b}^{\mathrm{IEO}}\right\|_{\boldsymbol{V}} \le \left\|\boldsymbol{b}^{\mathrm{ETO}}\right\|_{\boldsymbol{V}}$ *and* $0 = R^{\mathrm{SAA}} \le R^{\mathrm{IEO}} \le R^{\mathrm{ETO}}$.

Theorems 3 and 4 stipulate that, once the degree of misspecification is larger than the statistical error $(0 < \alpha < 1/2)$, SAA will dominate ETO which will further dominate IEO. This is because in this regime only the bias surfaces, and this ordering is in line with the bias ordering in Theorems 1 and 2.

In both the balanced and the severe misspecification cases, the bias term can be significant relative to the variance. In all cases, the bias has the form $\boldsymbol{b}^\square = \mathbb{E}_{\boldsymbol{\theta}_0}[u(\boldsymbol{z})(\mathrm{IF}^\square(\boldsymbol{z}) - \mathrm{IF}^{\mathrm{SAA}}(\boldsymbol{z}))]$, an inner product between the misspecification direction and the difference of influence functions between the considered estimation-optimization pipeline and SAA. Note that the latter difference is always zero for SAA, which coincides with the model-free nature of SAA that elicits zero bias. On the other hand, for either IEO or ETO, the bias effect can be minimized if the misspecification direction is orthogonal to the influence function difference. While this characterization is generally opaque, the following provides a more manageable sufficient condition.

**Theorem 5** (Approximately Impactless Misspecification Direction). *Let the assumptions in Theorem 3 hold. If* $u(\cdot) \in \left\{\boldsymbol{\beta}^\top \boldsymbol{s}_{\boldsymbol{\theta}_0}(\cdot) : \boldsymbol{\beta} \in \mathbb{R}^{d_{\boldsymbol{\theta}}}\right\}$, *then* $\boldsymbol{b}^{\mathrm{ETO}}$ *(and thus* $\boldsymbol{b}^{\mathrm{IEO}}$ *and* $\boldsymbol{b}^{\mathrm{SAA}}$*)* $= \boldsymbol{0}$.

Theorem 5 states that if the misspecification direction is in the linear span of the score function of the imposed model at $\boldsymbol{\theta}_0$, the asymptotic bias is zero for all methods. Figure 1 (b) illustrates such a direction, where $u(\boldsymbol{z})$ is tangential to the model $\{P_{\boldsymbol{\theta}}\}$ at $P_{\boldsymbol{\theta}_0}$. In this case, the interesting direction of misspecification $u(\boldsymbol{z})$ aligns with the parametric information and couples the influence function of ETO and SAA. As a result, ETO can induce zero decision bias by merely conducting MLE to infer $\boldsymbol{\theta}$, even though the model is misspecified.

Note that the condition in Theorem 5 depends only on the parametric model, but not the downstream optimization problem. Nonetheless, it already allows us to understand and provide examples where a model misspecification can be significant, namely shooting outside the parametric model, yet the impact on the bias of the resulting decision is negligible. We provide an explicit example as follows.

**Example 1.** Consider the distribution family $\{P_{\boldsymbol{\theta}} : \boldsymbol{\theta} \in \mathbb{R}\}$ to be normal distributions with variance 1, $N(\boldsymbol{\theta}, 1)$, where $\boldsymbol{\theta}_0 = 0$. We define the tilted distribution $Q_t(\boldsymbol{z})$ with density $q_t(\boldsymbol{z}) \propto (1 + t\boldsymbol{z})_+ e^{-\boldsymbol{z}^2/2}$. In this case, $Q_t(\boldsymbol{z})$ satisfies Definition 1 and the conditions in Theorem 5, but $Q_t \notin \{P_{\boldsymbol{\theta}} : \boldsymbol{\theta} \in \mathbb{R}\}$.

In Example 1, the parametric family is a normal location family, while at $\boldsymbol{\theta}_0 = 0$, the perturbed distribution family $Q_t$ is never normally distributed for all $t > 0$. The direction of perturbation

(misspecification) at $\boldsymbol{\theta}_0$ here is $u(\boldsymbol{z}) = \boldsymbol{s}_{\boldsymbol{\theta}_0}(\boldsymbol{z}) = \boldsymbol{z}$. In other words, even if the ground truth distribution of uncertain parameters is complicated, model-based approaches under a simplified and misspecified parametric family can still be employed with a satisfying decision regret performance.

### 3.3 Mild Misspecification

Finally, we establish results under mild misspecification.

**Theorem 6** (Asymptotics and Comparisons under Mild Misspecification)**.** *Suppose Assumptions 1, 3 and 4 hold. In the mildly misspecified case, under $Q^n$, for $\square \in \{\mathrm{SAA}, \mathrm{ETO}, \mathrm{IEO}\}$,*

$$\sqrt{n}(\hat{\boldsymbol{w}}^{\square} - \boldsymbol{w}_n^*) \overset{Q^n}{\rightsquigarrow} N^{\square} \,,$$

$$nR_{Q^n}(\hat{\boldsymbol{w}}^{\square}) \overset{Q^n}{\rightsquigarrow} \frac{1}{2}(N^{\square})^{\top} \boldsymbol{V} N^{\square}.$$

*Moreover, in terms of bias, all three estimators have asymptotically zero biases. In terms of variance,* $\mathrm{var}\left(N^{\mathrm{SAA}}\right) \geq \mathrm{var}\left(N^{\mathrm{IEO}}\right) \geq \mathrm{var}\left(N^{\mathrm{ETO}}\right)$. *In terms of regret, it holds that*

$$\frac{1}{2}(N^{\mathrm{ETO}})^{\top} \boldsymbol{V} N^{\mathrm{ETO}} \preceq_{\mathrm{st}} \frac{1}{2}(N^{\mathrm{IEO}})^{\top} \boldsymbol{V} N^{\mathrm{IEO}} \preceq_{\mathrm{st}} \frac{1}{2}(N^{\mathrm{SAA}})^{\top} \boldsymbol{V} N^{\mathrm{SAA}} \,.$$

Theorem 6 shows that the obtained solutions (consequently also the regret) exhibit asymptotic behaviors in accordance with the universal ordering of ETO best, then IEO, and then SAA. In this regime, the bias is negligible, while the variance is the dominant term, and the regret distribution is related to the variance. The phenomenon also holds in the well-specified regime stated as follows.

**Proposition 1** (Asymptotics under Well-Specification[Elmachtoub et al., 2023])**.** *In the well-specified case where $Q = P_{\boldsymbol{\theta}_0}$, under Assumptions 1, 3 and 4, for $\square \in \{\mathrm{SAA}, \mathrm{ETO}, \mathrm{IEO}\}$, we have*

$$\sqrt{n}\left(\hat{\boldsymbol{w}}^{\square} - \boldsymbol{w}^*\right) = \frac{1}{\sqrt{n}} \sum_{i=1}^{n} \mathrm{IF}^{\square}(\boldsymbol{z}_i) + o_{P^n}(1).$$

*Moreover,* $\mathrm{var}\left(\mathrm{IF}^{\mathrm{SAA}}(\boldsymbol{z})\right) \geq \mathrm{var}\left(\mathrm{IF}^{\mathrm{IEO}}(\boldsymbol{z})\right) \geq \mathrm{var}\left(\mathrm{IF}^{\mathrm{ETO}}(\boldsymbol{z})\right)$. *If $d_{\boldsymbol{\theta}} = d_{\boldsymbol{w}}$ and $\boldsymbol{\Sigma}$ is a square and full-rank matrix, then* $\mathrm{var}\left(\mathrm{IF}^{\mathrm{SAA}}(\boldsymbol{z})\right) = \mathrm{var}\left(\mathrm{IF}^{\mathrm{IEO}}(\boldsymbol{z})\right)$.

## 4 Numerical Experiments

In this section, we validate our findings by conducting numerical experiments on the newsvendor problem, a classic example in operations research with non-linear cost objectives. We show and compare the performances of the three data-driven methods in the finite-sample regimes under different local misspecification settings, including different directions and degrees of misspecification. The experimental results in the finite-sample regime are consistent with our asymptotic comparisons. All computations were carried out on a personal desktop computer without GPU acceleration.

The newsvendor problem has the objective function $c(\boldsymbol{w}, \boldsymbol{z}) = \boldsymbol{a}^{\top} (\boldsymbol{w} - \boldsymbol{z})^{+} + \boldsymbol{d}^{\top} (\boldsymbol{z} - \boldsymbol{w})^{+}$, where for each $j \in [d_z]$: (1) $z^{(j)}$ is the customers' random demand of product $j$; (2) $w^{(j)}$ is the decision variable, the ordering quantity for product $j$; (3) $a^{(j)}$ is the holding cost for product $j$; (4) $d^{(j)}$ is the backlogging cost for product $j$. We assume the random demand for each product are independent and the holding cost and backlogging cost is uniform among all products by setting $a^{(j)} = 5$ and $d^{(j)} = 1$ for all $j \in [d_z]$.

We describe the local misspecified setting by using the framework of Example 5 and building a model and generating a random demand dataset as follows. We denote the training dataset as $\left\{z_i^{(j)}\right\}_{i=1}^{n}$, where $n$ is the training sample size. The model assumes that the demand for each product $j \in [d_z]$ is normally distributed with the distribution $N(j\theta, 1)$ where $\theta$ is unknown and needs to be learned. We first describe the well-specified setting, where the demand distribution for product $j$ is $N(3j, 1)$. In this case, the probability density function of the random demand of each product is $p_j(z^{(j)}) \propto \exp(-(z^{(j)} - 3j)^2/2)$. To describe the local misspecification, we need to specify the direction and degree of misspecification, i.e., the expression of $u(\boldsymbol{z})$ and $\alpha$ in Section 2.3. We set (1) $\alpha = 0.1$ to denote the severely misspecified setting, (2) $\alpha = 0.5$ to denote the balanced

setting and $\alpha = 2$ to denote the mildly misspecified setting. We discuss two types of directions: (1) $u(\boldsymbol{z}) = \prod_{j=1}^{d_z} \left( z^{(j)} \right)^2$; (2) $u(\boldsymbol{z}) = \prod_{j=1}^{d_z} \left( z^{(j)} - 3j \right)^2 / 2$.

We show experimental results in Figure 2 - Figure 3 to support our theoretical results in Section 3, using the mean, median, 25-th quantile, 75-th quantile and histograms of the regret. When $u(\boldsymbol{z}) = \prod_{j=1}^{d_z} \left( z^{(j)} \right)^2$ and $u(\boldsymbol{z}) = \prod_{j=1}^{d_z} \left( z^{(j)} - 3j \right)^2 / 2$, in the mildly specified case, ETO has a lower regret than IEO, and IEO has a lower regret than SAA. However, in the severely misspecified regime, the ordering of the three methods flips. This is consistent with our theoretical comparison results in Theorems 6 and 4. In the balanced regime, experimental results show that IEO has the lowest regret among the three methods. This is also consistent with the theoretical insight in Theorem 2 that IEO has the advantage of achieving bias-variance trade-off in terms of the decisions and regrets.

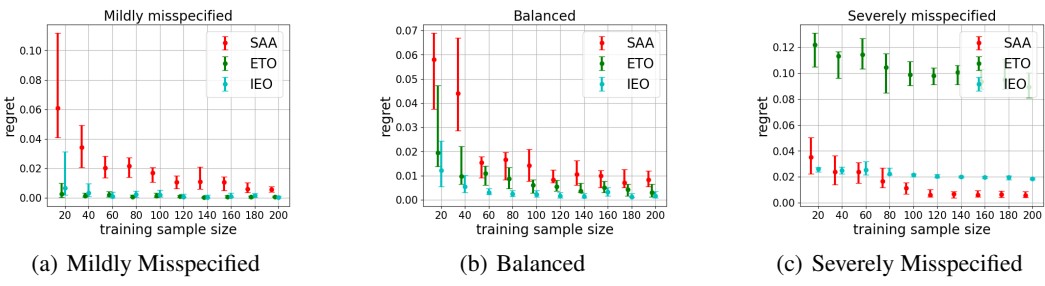

(a) Mildly Misspecified        (b) Balanced        (c) Severely Misspecified

Figure 2: The direction of misspecification satisfies $u(\boldsymbol{z}) = \prod_{j=1}^{d_z} \left( z^{(j)} \right)^2$.

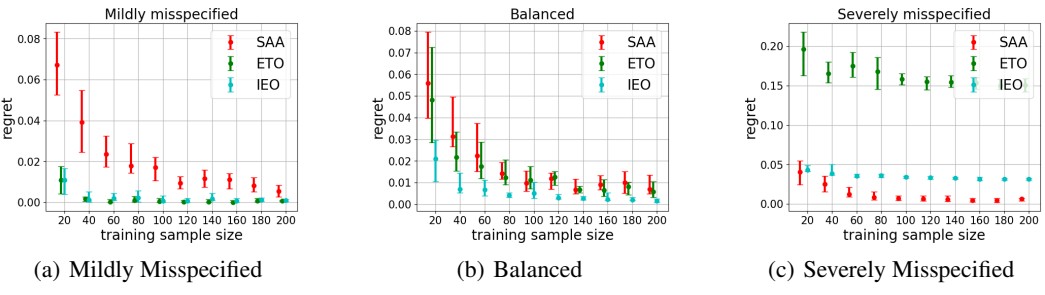

(a) Mildly Misspecified        (b) Balanced        (c) Severely Misspecified

Figure 3: The direction of misspecification satisfies $u(\boldsymbol{z}) = \prod_{j=1}^{d_z} \left( z^{(j)} - 3j \right)^2 / 2$.

## 5 Conclusions and Discussions

In this paper, we present the first results on analyzing local model misspecification in data-driven optimization. Our framework captures scenarios where the model-based approach is nearly well-specified, moving beyond the existing dichotomy of well-specified and misspecified models. We conduct a detailed analysis of the relative performances of SAA, ETO, and IEO, providing insights into their bias, variance, and regret. By classifying local misspecification into three regimes, our analysis illustrates how varying degrees of misspecification impacts performance. In particular, we show that in the balanced misspecification case, ETO exhibits the best variance, SAA exhibits the best bias, while IEO entails a bias-variance tradeoff that can potentially result in lower overall decision errors than both ETO and SAA. Additionally, we derive closed-form expressions for decision bias and variance. From this, we show how the orthogonality between the misspecification direction and the difference of influence functions can lead to bias cancellation, and provide more transparent sufficient condition for such phenomenon in relation to the tangentiality on the score function. Technically, we leverage and generalize tools from contiguity theory in statistics to establish the performance orderings and the clean, interpretable bias and variance expressions. Future research directions include extending our framework to contextual or constrained optimization problems, where challenges like feasibility guarantees and model complexity become increasingly significant.

## Acknowledgments and Disclosure of Funding

We gratefully acknowledge support from the National Science Foundation grant IIS-2238960, the Office of Naval Research awards N00014-22-1-2530 and N00014-23-1-2374, InnoHK initiative, the Government of the HKSAR, Laboratory for AI-Powered Financial Technologies, and Columbia SEAS Innovation Hub Award. The authors thank the anonymous reviewers for their constructive comments, which have greatly improved the quality of our paper.

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

# A  Additional Examples and Techniques Details

## A.1  Examples of Data-Driven Optimization in Operations Research wtih Nonlinear Cost Objectives

We now give two canonical examples of stochastic optimization problems in operations research with non-linear cost objectives.

**Example 2** (Multi-Product Newsvendor Problem). The newsvendor problem has the objective function $c(\boldsymbol{w}, \boldsymbol{z}) = \boldsymbol{a}^\top (\boldsymbol{w} - \boldsymbol{z})^+ + \boldsymbol{d}^\top (\boldsymbol{z} - \boldsymbol{w})^+$, where for each $j \in [d_z]$: (1) $z^{(j)}$ is the customers' random demand of product $j$, ; (2) $w^{(j)}$ is the decision variable, the ordering quantity for product $j$; (3) $a^{(j)}$ is the holding cost for product $j$; (4) $d^{(j)}$ is the backlogging cost for product $j$ and (5) the goal is to minimize the expected total cost.

Consider another classical problem in operations research, the portfolio optimization problem [Kallus and Mao, 2022, Grigas et al., 2021, Elmachtoub et al., 2023].

**Example 3** (Portfolio Optimization). Let $d_w = d_z + 1$ and denote the cost function as $c(\boldsymbol{w}, \boldsymbol{z}) = \gamma \left( \boldsymbol{w}^\top (\boldsymbol{z}, -1) \right)^2 + \exp \left( -\boldsymbol{w}^\top (\boldsymbol{z}, 0) \right)$. The decision $\boldsymbol{w}$ satisfies $\left( w^{(1)}, w^{(2)}, ..., w^{(d_w-1)} \right) \in \mathbb{R}^{d_w-1}$, denoting the investment fraction on products $1, 2, ...d_z$ (i.e., $d_w - 1$) and $w^{(d_w)}$ is an auxiliary decision variable. The first component represents the risk (variance) of the portfolio and the second component represents the exponential utility of the portfolio.

## A.2  Further Examples of Local Misspecification

**Example 4** (Parametric Perturbation: Quadratic Mean Differentiability (QMD) family). Suppose $P^n = P_{\boldsymbol{\theta}_0}^{\otimes n}$ for some fixed $\boldsymbol{\theta}_0$. Consider a sequence of vectors in $\mathbb{R}^{d_\theta}$, say $\{\boldsymbol{h}_n\}_{n=1}^\infty$. Suppose the joint distribution of $\{\boldsymbol{z}_i\}_{i=1}^n$, $Q^n$, is also in the parametric family, but is of the form $P_{\boldsymbol{\theta}_0+\boldsymbol{h}_n}^{\otimes n}$. If there exists a score function $\dot{\boldsymbol{\ell}}_{\boldsymbol{\theta}}(\boldsymbol{z}) : \mathcal{Z} \to \mathbb{R}^{d_\theta}$ with $\mathbb{E}_{\boldsymbol{\theta}_0}[\dot{\boldsymbol{\ell}}_{\boldsymbol{\theta}_0}(\boldsymbol{z})] = \boldsymbol{0}$ such that

$$\int \left( \sqrt{p_{\boldsymbol{\theta}_0+\boldsymbol{h}_n}} - \sqrt{p_{\boldsymbol{\theta}_0}} - \frac{1}{2} \dot{\boldsymbol{\ell}}_{\boldsymbol{\theta}_0}^\top \boldsymbol{h}_n \sqrt{p_{\boldsymbol{\theta}_0}} \right)^2 dz = o(\|\boldsymbol{h}_n\|^2), \boldsymbol{h}_n \to \boldsymbol{0}.$$

In particular, in our framework, we focus on the case where $\boldsymbol{h}_n = \boldsymbol{h}/n^\alpha$ for a fixed vector $\boldsymbol{h}$. When $\alpha = 1/2$, van der Vaart [2000] shows that the likelihood ratio between $Q^n$ and $P^n$ satisfies:

$$\log \frac{dQ^n(\boldsymbol{z}_1, ..., \boldsymbol{z}_n)}{dP^n(\boldsymbol{z}_1, ..., \boldsymbol{z}_n)} = \frac{1}{\sqrt{n}} \sum_{i=1}^n \boldsymbol{h}^\top \dot{\boldsymbol{\ell}}_{\boldsymbol{\theta}_0}(\boldsymbol{z}_i) - \frac{1}{2}\boldsymbol{h}^\top \boldsymbol{I}\boldsymbol{h} + o_{P^n}(\|\boldsymbol{h}\|^2),$$

where $\boldsymbol{I} := \mathbb{E}_{\boldsymbol{\theta}_0}[\dot{\boldsymbol{\ell}}_{\boldsymbol{\theta}_0} \dot{\boldsymbol{\ell}}_{\boldsymbol{\theta}_0}^\top]$ denotes the Fisher information. In other words, under $P^n$

$$\log \frac{dQ^n(\boldsymbol{z}_1, ..., \boldsymbol{z}_n)}{dP^n(\boldsymbol{z}_1, ..., \boldsymbol{z}_n)} \overset{P^n}{\to} N(-\frac{1}{2}\boldsymbol{h}^\top \boldsymbol{I}\boldsymbol{h}, \boldsymbol{h}^\top \boldsymbol{I}\boldsymbol{h}),$$

where the limiting distribution is a Gaussian distribution with mean $-\frac{1}{2}\boldsymbol{h}^\top \boldsymbol{I}\boldsymbol{h}$ and variance $\boldsymbol{h}^\top \boldsymbol{I}\boldsymbol{h}$.

In the previous example, the ground truth distribution $Q$ is still in $\{P_{\boldsymbol{\theta}} : \boldsymbol{\theta} \in \Theta\}$ but is in the local neighbourhood of $P_{\boldsymbol{\theta}_0}$. The more common and interesting examples are when $Q \notin \{P_{\boldsymbol{\theta}} : \boldsymbol{\theta} \in \Theta\}$ as discussed in examples below.

**Example 5** (Semi-parametric Local Perturbation: Part I). Suppose $P_{\boldsymbol{\theta}_0}$ is a given distribution, and $u(\boldsymbol{z}) : \mathcal{Z} \to \mathbb{R}$ is an unobserved random variable with $\mathbb{E}_{\boldsymbol{\theta}_0}[u] = 0$ and a finite variance $\mathbb{E}_{\boldsymbol{\theta}_0}[u^2]$. For a scalar $t$ in a neighborhood of zero, we define the tilted distribution of $P_{\boldsymbol{\theta}_0}$, called $Q_t$, as

$$dQ_t(\boldsymbol{z}) = \frac{\exp(tu(\boldsymbol{z}))}{C_t} dP_{\boldsymbol{\theta}_0}(\boldsymbol{z})$$

where $C_t = \int \exp(tu(\boldsymbol{z})) dP_{\boldsymbol{\theta}_0}(\boldsymbol{z}) < \infty$ is a normalization constant. Clearly $Q_{t=0} = P_{\boldsymbol{\theta}_0}$.

**Lemma 2** (Log Likelihood Ratio Property in Example 5). *Under Definition 2, when $\alpha = 1/2$, i.e., $Q^n = Q_{1/\sqrt{n}}^{\otimes n}$, the log-likelihood ratio between $Q^n$ and $P^n$ satisfies:*

$$\log \frac{dQ^n(\boldsymbol{z}_1, ..., \boldsymbol{z}_n)}{dP^n(\boldsymbol{z}_1, ..., \boldsymbol{z}_n)} = \frac{1}{\sqrt{n}} \sum_{i=1}^n u(\boldsymbol{z}_i) - \frac{1}{2}\mathbb{E}_{\boldsymbol{\theta}_0}[u^2] + o_{P^n}(1) .$$

*This implies, under $P^n$,* $\log \frac{dQ^n(z_1,...,z_n)}{dP^n(z_1,...,z_n)} \xrightarrow{P^n} N(-\frac{1}{2}\mathbb{E}_{\boldsymbol{\theta}_0}[u^2], \mathbb{E}_{\boldsymbol{\theta}_0}[u^2])$.

**Example 6** (Semi-parametric Local Perturbation: Part II)**.** Consider the random variable $u(\boldsymbol{z}) : \mathcal{Z} \to \mathbb{R}$ with a zero mean, $\mathbb{E}_{\boldsymbol{\theta}_0}[u] = 0$, and finite second moment, say $\mathbb{E}_{\boldsymbol{\theta}_0}[u^2]$. Now we define the tilted distribution:

$$dQ_t(\boldsymbol{z}) = \frac{[1 + tu(\boldsymbol{z})]_+}{C_t} dP_{\boldsymbol{\theta}_0}(\boldsymbol{z}) \text{ where } C_t = \int [1 + tu(\boldsymbol{z})]_+ dP_{\boldsymbol{\theta}_0}(\boldsymbol{z}).$$

In particular, in our framework we focus on the case where $Q = Q_{1/n^\alpha}$ and $Q^n := Q_{1/n^\alpha}^{\otimes n}$. When $\alpha = 1/2$, by Duchi [2021], the log likelihood ratio satisfies

$$\log \frac{dQ^n(z_1,...,z_n)}{dP^n(z_1,...,z_n)} = \frac{1}{\sqrt{n}} \sum_{i=1}^n u(z_i) - \frac{1}{2}\mathbb{E}_{\boldsymbol{\theta}_0}[u^2] + o_{P^n}(1).$$

In other words, under $P^n$,

$$\log \frac{dQ^n(z_1,...,z_n)}{dP^n(z_1,...,z_n)} \xrightarrow{P^n} N(-\frac{1}{2}\mathbb{E}_{\boldsymbol{\theta}_0}[u^2], \mathbb{E}_{\boldsymbol{\theta}_0}[u^2]).$$

**Example 7** (Semi-parametric Local Perturbation: Part III)**.** Consider the function $g : \mathbb{R} \to [-1, 1]$ be any three-times continuously differentiable function where $g(x) = x$ for $x \in [-1/2, 1/2]$ and $g' \geq 0$ and the first three derivatives of $g$ are bounded. Consider the random variable $u(\boldsymbol{z}) : \mathcal{Z} \to \mathbb{R}$ with a zero mean $\mathbb{E}_{\boldsymbol{\theta}_0}[u] = \boldsymbol{0}$ and finite second moment, say $\mathbb{E}_{\boldsymbol{\theta}_0}[u^2]$. Now, for $\boldsymbol{t} \in \mathbb{R}$, we define the tilted distribution

$$dQ_t(\boldsymbol{z}) = \frac{1 + g(tu(\boldsymbol{z}))}{C_t} dP_{\boldsymbol{\theta}_0}(\boldsymbol{z}) \text{ where } C_t = 1 + \int g(tu(\boldsymbol{z})) dP_{\boldsymbol{\theta}_0}(\boldsymbol{z}).$$

In particular, in our framework we focus on the case where $Q = Q_{1/n^\alpha}$ and $Q^n := Q_{1/n^\alpha}^{\otimes n}$. When $\alpha = 1/2$, by Duchi and Ruan [2021], the log likelihood ratio satisfies the following Property:

$$\log \frac{dQ^n(z_1,...,z_n)}{dP^n(z_1,...,z_n)} = \frac{1}{\sqrt{n}} \sum_{i=1}^n u(z_i) - \frac{1}{2}\mathbb{E}_{\boldsymbol{\theta}_0}[u^2] + o_{P^n}(1).$$

In other words, under $P^n$,

$$\log \frac{dQ^n(z_1,...,z_n)}{dP^n(z_1,...,z_n)} \xrightarrow{P^n} N(-\frac{1}{2}\mathbb{E}_{\boldsymbol{\theta}_0}[u^2], \mathbb{E}_{\boldsymbol{\theta}_0}[u^2]).$$

**Example 8** (Semi-parametric Local Perturbation: Part IV (QMD Family))**.** Consider a scalar function $u(\boldsymbol{z}) : \mathcal{Z} \to \mathbb{R}$ with zero mean $\mathbb{E}_{\boldsymbol{\theta}_0}[u] = 0$ and finite second order moment $\mathbb{E}_{\boldsymbol{\theta}_0}[u^2]$. We define a tilted distribution $Q_t$ for $t \in \mathbb{R}$ with probability density (mass) function $q_t$ with respect to the dominated measure (note that $Q_t$ is not necessarily in the parametric family $\{P_{\boldsymbol{\theta}} : \boldsymbol{\theta} \in \Theta\}$) with $q_0 = p_{\boldsymbol{\theta}_0}$. We further assume the quadratic mean differentiability

$$\int \left( \sqrt{q_t} - \sqrt{p_{\boldsymbol{\theta}_0}} - \frac{1}{2}tu\sqrt{p_{\boldsymbol{\theta}_0}} \right)^2 dz = o(t^2).$$

Note that when $q_0 = p_{\boldsymbol{\theta}_0}$. In particular, in our framework we focus on the case where $Q = Q_{1/n^\alpha}$ and $Q^n := Q_{1/n^\alpha}^{\otimes n}$. When $\alpha = 1/2$, by Duchi [2021], we have

$$\log \frac{dQ^n(z_1,...,z_n)}{dP^n(z_1,...,z_n)} = \frac{1}{\sqrt{n}} \sum_{i=1}^n u(z_i) - \frac{1}{2}\mathbb{E}_{\boldsymbol{\theta}_0}[u^2] + o_{P^n}(1).$$

Note that Example 8 is the most general version and includes Example 6-7 as particular examples under some mild assumptions.

### A.3 Additional Technical Details

We introduce standard regularity assumptions for general $M$-estimation problems in asymptotic statistics [van der Vaart, 2000], which include our SAA, ETO, and IEO methods as examples.

**Assumption 3** (Regularity Assumptions for $M$-estimation). *Suppose the i.i.d. random variables $\{z_i\}_{i=1}^n$ follows a distribution Q. Suppose the function $z \to m_\zeta(z)$ is measurable with respect to $z$ for all $\zeta$ and*

1. *$\sup_\zeta \left| \frac{1}{n} \sum_{i=1}^n m_\zeta(z_i) - \mathbb{E}_Q\left[m_\zeta(z)\right] \right| \xrightarrow{p} 0$,*

2. *there exists $\zeta^* = \operatorname{argmax}_\zeta \mathbb{E}_Q\left[m_\zeta(z)\right]$, for all $\varepsilon > 0$, $\sup_{\zeta : \|\zeta - \zeta^*\| \ge \varepsilon} \mathbb{E}_Q\left[m_\zeta(z)\right] < \mathbb{E}_Q\left[m_{\zeta^*}(z)\right]$,*

3. *the mapping $\zeta \to m_\zeta(z)$ is differentiable at $\zeta^*$ for Q-almost every $z$ with derivative $\nabla_\zeta m_{\zeta^*}(z)$ and such that for every $\zeta_1$ and $\zeta_2$ in a neighbourhood of $\zeta^*$ and a measurable function $K$ with $\mathbb{E}_Q[K(z)^2] < \infty$*
$$|m_{\zeta_1}(z) - m_{\zeta_2}(z)| \le K(z) \|\zeta_1 - \zeta_2\|.$$

4. *assume that the mapping $\zeta \to \mathbb{E}_Q[m_\zeta(z)]$ admits a second-order Taylor expansion at a point of maximum $\zeta^*$ with nonsigular symmetric second order matrix $V_{\zeta^*}$.*

*If the random sequence $\hat{\zeta}_n$ satisfies $\frac{1}{n} \sum_{i=1}^n m_{\hat{\zeta}_n}(z_i) = \sup_\zeta \sum_{i=1}^n m_{\hat{\zeta}}(z_i)$, then $\hat{\zeta}_n \xrightarrow{p} \zeta^*$ and*

$$\sqrt{n}\left(\hat{\zeta}_n - \zeta^*\right) = -V_{\zeta^*}^{-1} \frac{1}{\sqrt{n}} \sum_{i=1}^n \nabla_\zeta m_{\zeta^*}(z_i) + o_Q(1).$$

Throughout this paper, we assume Assumption 3 holds.

- For SAA, consider $m_\zeta(z) = -c(w, z)$ with the parameter $\zeta = w$.
- For ETO, consider $m_\zeta(z) = \log p_\theta(z)$ with parameter $\zeta = \theta$.
- For IEO, consider $m_\zeta(z) = c(w_\theta, z)$ with $\zeta = \theta$.

When we say Assumption 3 holds, it means that Assumption 3 holds for the corresponding $m_\zeta(z)$ in SAA, ETO, and IEO.

**Assumption 4** (Interchangeability). *For any $\theta \in \Theta$ and $w \in \Omega$,*

$$\nabla_\theta \int \nabla_w c(w, z)^\top p_\theta(z) dz = \int \nabla_w c(w, z)^\top \nabla_\theta p_\theta(z) dz,$$

$$\int \nabla_w c(w, z) p_\theta(z) dz|_{w = w^*} = \nabla_w \int c(w, z) p_\theta(z) dz|_{w = w^*}$$

The interchangeability condition in Assumption 4 is a standard assumption in the Cramer-Rao bound [Bickel and Doksum, 2015]. A standard route to check the interchangeability condition is to use the dominated convergence theorem. For instance, we provide a way to check the first interchange equation. If $p_\theta(z)$ is continuously differentiable with respect to $\theta$, and there exists a real-valued function $q(z)$ such that $\int \nabla_w c(w, z)^\top q(z) dz < +\infty$ and $\|\nabla_\theta p_\theta(z)\|_\infty \le q(z)$, then we have $\nabla_\theta \int \nabla_w c(w, z)^\top p_\theta(z) dz = \int \nabla_w c(w, z)^\top \nabla_\theta p_\theta(z) dz$. Other sufficient conditions (more delicate but still based on the dominated convergence theorem) can be found in L'Ecuyer [1990], Asmussen and Glynn [2007], Glasserman [2004].

Next, we present some auxiliary lemmas that are helpful for deriving our theorems.

The first is a classic lemma in asymptotic statistics, called Le Cam's third lemma (Example 6.7 in van der Vaart [2000]).

**Lemma 3** (Le Cam's third lemma). *Let $P^n$ and $Q^n$ be sequences of probability measures on measurable spaces $(\Omega_n, \mathcal{F}_n)$ and let $X_n$ be a sequence of random vectors. Suppose that*

$$\left(X_n, \log \frac{dQ^n}{dP^n}\right) \xrightarrow{P^n} N\left(\begin{pmatrix} \mu \\ -\frac{1}{2}\sigma^2 \end{pmatrix}, \begin{pmatrix} \Sigma & \tau \\ \tau^\top & \sigma^2 \end{pmatrix}\right),$$

*then*

$$X_n \xrightarrow{Q^n} N(\boldsymbol{\mu} + \boldsymbol{\tau}, \boldsymbol{\Sigma}).$$

We now state a auxiliary lemma about the directional differentiability of the optimal solutions to stochastic optimization problems.

**Lemma 4** (Directional differentiability of optimal solutions: Part I). *Consider the distribution $Q_t(\boldsymbol{z})$ in Definition 1. Let*

$$\boldsymbol{w}_t := \operatorname*{argmin}_{\boldsymbol{w} \in \Omega} \mathbb{E}_{Q_t} \left[ c(\boldsymbol{w}, \boldsymbol{z}) \right].$$

*Then under Assumptions 1, 3, and 4,*

$$\lim_{t \to 0} \frac{1}{t} \left( \boldsymbol{w}_t - \boldsymbol{w}_0 \right) = \mathbb{E}_{\boldsymbol{\theta}_0}[u(\boldsymbol{z})\mathrm{IF}^{\mathrm{SAA}}(\boldsymbol{z})].$$

Equipped with the lemma above, we can get the convergence of $n^\alpha \left( \boldsymbol{w}_n^* - \boldsymbol{w}_{\boldsymbol{\theta}_0} \right)$ under the three locally misspecified regimes:

$$\lim_{n \to \infty} n^\alpha \left( \boldsymbol{w}_n^* - \boldsymbol{w}_{\boldsymbol{\theta}_0} \right) = \lim_{t \to 0} \frac{1}{t} \left( \boldsymbol{w}_t - \boldsymbol{w}_0 \right) = \mathbb{E}_{\boldsymbol{\theta}_0}[u(\boldsymbol{z})\mathrm{IF}^{\mathrm{SAA}}(\boldsymbol{z})].$$

*Proof of Lemma 4.* Note that $\boldsymbol{w}_n^*$ is the minimizer of $\mathbb{E}\left[ c(\boldsymbol{w}, \boldsymbol{z}) \right]$ under $Q^n$ while $\boldsymbol{w}_{\boldsymbol{\theta}_0}$ under $P^n$. We will use the directional differentiablity of optimal solution to derive this fact.

We denote $v(\boldsymbol{w}, Q_t)$ as $\mathbb{E}_{Q_t}[c(\boldsymbol{w}, \boldsymbol{z})]$, $\mathcal{G}(\boldsymbol{w}, t) := \nabla_{\boldsymbol{w}} v(\boldsymbol{w}, Q_t)$ and $\boldsymbol{w}_t := \operatorname{argmin} v(\boldsymbol{w}, Q_t)$. Note that $\mathcal{G}(\boldsymbol{w}_t, t) = 0$ for all $t$. By implicit function theorem,

$$
\begin{aligned}
\lim_{t \to 0} \frac{1}{t} \left( \boldsymbol{w}_t - \boldsymbol{w}_0 \right) &= -[\nabla_{\boldsymbol{w}} \mathcal{G}(\boldsymbol{w}_{\boldsymbol{\theta}_0}, 0)]^{-1} \frac{\partial}{\partial t} \nabla_{\boldsymbol{w}} v(\boldsymbol{w}_{\boldsymbol{\theta}_0}, Q_t)|_{t=0} \\
&= -\nabla_{\boldsymbol{w}\boldsymbol{w}} \mathbb{E}_{\boldsymbol{\theta}_0}[c(\boldsymbol{w}_{\boldsymbol{\theta}_0}, \boldsymbol{z})] \frac{\partial}{\partial t} \nabla_{\boldsymbol{w}} v(\boldsymbol{w}_{\boldsymbol{\theta}_0}, Q_t)|_{t=0} \\
&= -\boldsymbol{V}^{-1} \frac{\partial}{\partial t} \int \nabla_{\boldsymbol{w}} c(\boldsymbol{w}_{\boldsymbol{\theta}_0}, \boldsymbol{z}) dQ_t(\boldsymbol{z})|_{t=0} \\
&= -\boldsymbol{V}^{-1} \int \nabla_{\boldsymbol{w}} c(\boldsymbol{w}_{\boldsymbol{\theta}_0}, \boldsymbol{z}) \frac{\partial}{\partial t} dQ_t(\boldsymbol{z})|_{t=0}.
\end{aligned}
$$

From Definition 1, we know that for almost every $\boldsymbol{z}$, $\frac{\partial}{\partial t} \log q_t(\boldsymbol{z})|_{t=0} = u(\boldsymbol{z})$. Hence, we have for almost every $\boldsymbol{z}$,

$$\frac{\partial}{\partial t} q_t(\boldsymbol{z}) \Big|_{t=0} = q_0(\boldsymbol{z}) u(\boldsymbol{z}). \tag{3}$$

In conclusion,

$$
\begin{aligned}
&-\boldsymbol{V}^{-1} \int \nabla_{\boldsymbol{w}} c(\boldsymbol{w}_{\boldsymbol{\theta}_0}, \boldsymbol{z}) \frac{\partial}{\partial t} dQ_t(\boldsymbol{z})|_{t=0} \\
&= -\boldsymbol{V}^{-1} \int \nabla_{\boldsymbol{w}} c(\boldsymbol{w}_{\boldsymbol{\theta}_0}, \boldsymbol{z}) \left\{ q_0(\boldsymbol{z}) u(\boldsymbol{z}) \right\} d\boldsymbol{z}
\end{aligned}
$$

Since

$$\int \nabla_{\boldsymbol{w}} c(\boldsymbol{w}_{\boldsymbol{\theta}_0}, \boldsymbol{z}) \left[ \int q_0(\boldsymbol{z}) u(\boldsymbol{z}) d\boldsymbol{z} \right] q_0(\boldsymbol{z}) d\boldsymbol{z} = \left[ \int q_0(\boldsymbol{z}) u(\boldsymbol{z}) d\boldsymbol{z} \right] \nabla_{\boldsymbol{w}} \mathbb{E}_{\boldsymbol{\theta}_0}[c(\boldsymbol{w}_{\boldsymbol{\theta}_0}, \boldsymbol{z})] = 0$$

and

$$\int \nabla_{\boldsymbol{w}} c(\boldsymbol{w}_{\boldsymbol{\theta}_0}, \boldsymbol{z}) q_0(\boldsymbol{z}) u(\boldsymbol{z}) d\boldsymbol{z} = \mathbb{E}_{\boldsymbol{\theta}_0}[u(\boldsymbol{z}) \nabla_{\boldsymbol{w}} c(\boldsymbol{w}_{\boldsymbol{\theta}_0}, \boldsymbol{z})],$$

we have

$$-\boldsymbol{V}^{-1} \int \nabla_{\boldsymbol{w}} c(\boldsymbol{w}_{\boldsymbol{\theta}_0}, \boldsymbol{z}) \frac{\partial}{\partial t} dQ_t(\boldsymbol{z})|_{t=0} = -\boldsymbol{V}^{-1} \mathbb{E}_{\boldsymbol{\theta}_0}[u(\boldsymbol{z}) \nabla_{\boldsymbol{w}} c(\boldsymbol{w}_{\boldsymbol{\theta}_0}, \boldsymbol{z})] = \mathbb{E}_{\boldsymbol{\theta}_0}[u(\boldsymbol{z})\mathrm{IF}^{\mathrm{SAA}}(\boldsymbol{z})].$$

Therefore,
$$\lim_{n\to\infty} \sqrt{n}(\boldsymbol{w}_n^* - \boldsymbol{w}_{\boldsymbol{\theta}_0}) = \lim_{t\to0} \frac{1}{t}(\boldsymbol{w}_t - \boldsymbol{w}_0) = \mathbb{E}_{\boldsymbol{\theta}_0}[u(\boldsymbol{z})\mathrm{IF}^{\mathrm{SAA}}(\boldsymbol{z})].$$
More generally, under severely and mildly specified regime, we have further
$$\lim_{n\to\infty} n^\alpha(\boldsymbol{w}_n^* - \boldsymbol{w}_{\boldsymbol{\theta}_0}) = \mathbb{E}_{\boldsymbol{\theta}_0}[u(\boldsymbol{z})\mathrm{IF}^{\mathrm{SAA}}(\boldsymbol{z})].$$

$\square$

We note that Lemma 4 holds for Example 5. To be more specific,
$$q_t(\boldsymbol{z}) = \frac{\exp(tu(\boldsymbol{z}))}{C_t} q_0(\boldsymbol{z}) \text{ where } C_t = \int \exp(tu(\boldsymbol{z}))q_0(\boldsymbol{z})d\boldsymbol{z}.$$
Therefore, the derivative of $q_t(\boldsymbol{z})$ with respect to $t$ is
$$\frac{\partial}{\partial t}q_t(\boldsymbol{z}) = \frac{q_0(\boldsymbol{z})\exp(tu(\boldsymbol{z}))u(\boldsymbol{z})\left[\int \exp(tu(\boldsymbol{z}))q_0(\boldsymbol{z})d\boldsymbol{z}\right]}{\left[\int \exp(tu(\boldsymbol{z}))q_0(\boldsymbol{z})d\boldsymbol{z}\right]^2}$$
$$- \frac{\left[\int \exp(tu(\boldsymbol{z}))q_0(\boldsymbol{z})u(\boldsymbol{z})d\boldsymbol{z}\right]q_0(\boldsymbol{z})\exp(tu(\boldsymbol{z}))}{\left[\int \exp(tu(\boldsymbol{z}))q_0(\boldsymbol{z})d\boldsymbol{z}\right]^2}.$$
At $t = 0$, since $\mathbb{E}_{\boldsymbol{\theta}_0}[u] = 0$, we have for almost every $\boldsymbol{z}$,
$$\frac{\partial}{\partial t}q_t(\boldsymbol{z})\Big|_{t=0} = q_0(\boldsymbol{z})u(\boldsymbol{z}) - \left[\int q_0(\boldsymbol{z})u(\boldsymbol{z})d\boldsymbol{z}\right]q_0(\boldsymbol{z}) = q_0(\boldsymbol{z})u(\boldsymbol{z}).$$
It is also possible to extend the result to other examples under additional regularity assumptions.

For Example 6, the result still holds. Recall that
$$q_t(\boldsymbol{z}) = \frac{[1 + tu(\boldsymbol{z})]_+}{C_t} q_0(\boldsymbol{z}), \qquad C_t = \int [1 + tu(\boldsymbol{z})]_+ q_0(\boldsymbol{z})d\boldsymbol{z}.$$
Hence,
$$\frac{\partial}{\partial t}q_t(\boldsymbol{z})|_{t=0} = q_0(\boldsymbol{z})\frac{C_t\frac{\partial}{\partial t}[1 + tu(\boldsymbol{z})]_+ - [1 + tu(\boldsymbol{z})]_+ \frac{\partial}{\partial t}C_t}{C_t^2}\Big|_{t=0}$$
$$= q_0(\boldsymbol{z})\left(\frac{\partial}{\partial t}[1 + tu(\boldsymbol{z})]_+\Big|_{t=0} - \frac{\partial}{\partial t}C_t\Big|_{t=0}\right)$$
$$= q_0(\boldsymbol{z})\left(u(\boldsymbol{z})\mathbb{1}\{1 + tu(\boldsymbol{z}) \geq 0\}\Big|_{t=0} - \int u(\boldsymbol{z})\mathbb{1}\{1 + tu(\boldsymbol{z}) \geq 0\}\Big|_{t=0}q_0(\boldsymbol{z})d\boldsymbol{z}\right)$$
$$= q_0(\boldsymbol{z})\left(u(\boldsymbol{z}) - \int u(\boldsymbol{z})q_0(\boldsymbol{z})d\boldsymbol{z}\right).$$
The result is the same as (3) and the conclusion of Lemma 4 still holds.

For Example 7, the result still holds. Recall that
$$q_t(\boldsymbol{z}) = \frac{1 + g(tu(\boldsymbol{z}))}{C_t} q_0(\boldsymbol{z}), \qquad C_t = \int (1 + g(tu(\boldsymbol{z}))) q_0(\boldsymbol{z})d\boldsymbol{z}.$$
Hence, by noting that $g'(0) = 1$,
$$\frac{\partial}{\partial t}q_t(\boldsymbol{z})|_{t=0} = q_0(\boldsymbol{z})\frac{C_t\frac{\partial}{\partial t}(1 + g(tu(\boldsymbol{z}))) - (1 + g(tu(\boldsymbol{z})))\frac{\partial}{\partial t}C_t}{C_t^2}\Big|_{t=0}$$
$$= q_0(\boldsymbol{z})\left(\frac{\partial}{\partial t}(1 + g(tu(\boldsymbol{z})))\Big|_{t=0} - \frac{\partial}{\partial t}C_t\Big|_{t=0}\right)$$
$$= q_0(\boldsymbol{z})\left(u(\boldsymbol{z})g'(0) - \int u(\boldsymbol{z})g'(0)q_0(\boldsymbol{z})d\boldsymbol{z}\right)$$
$$= q_0(\boldsymbol{z})\left(u(\boldsymbol{z}) - \int u(\boldsymbol{z})q_0(\boldsymbol{z})d\boldsymbol{z}\right).$$
The result is the same as (3) and the conclusion of Lemma 4 still holds.

We provide another auxiliary lemma similar to Lemma 4.

**Lemma 5** (Directional differentiability of optimal solutions: Part II). *Consider the distribution $Q_t$ in Definition 1 where $Q_0 = P_{\boldsymbol{\theta}_0}$. We denote*

$$\boldsymbol{\theta}_t^{\mathrm{KL}} := \operatorname*{argmax}_{\boldsymbol{\theta} \in \Theta} \mathbb{E}_{Q_t}[\log p_{\boldsymbol{\theta}}(\boldsymbol{z})],$$

$$\boldsymbol{\theta}_t^* := \operatorname*{argmin}_{\boldsymbol{\theta} \in \Theta} \mathbb{E}_{Q_t}[c(\boldsymbol{w}_{\boldsymbol{\theta}}, \boldsymbol{z})].$$

*Then under Assumptions 1, 3, and 4, we have*

$$\nabla_t \boldsymbol{\theta}_t^{\mathrm{KL}} := \lim_{t \to 0} \frac{1}{t} \left( \boldsymbol{\theta}_t^{\mathrm{KL}} - \boldsymbol{\theta}_0 \right) = \mathbb{E}_{\boldsymbol{\theta}_0}[u(\boldsymbol{z}) \mathrm{IF}^{\mathrm{ETO}}(\boldsymbol{z})],$$

$$\nabla_t \boldsymbol{\theta}_t^* := \lim_{t \to 0} \frac{1}{t} \left( \boldsymbol{\theta}_t^* - \boldsymbol{\theta}_0 \right) = \mathbb{E}_{\boldsymbol{\theta}_0}[u(\boldsymbol{z}) \mathrm{IF}^{\mathrm{IEO}}(\boldsymbol{z})].$$

*Proof of Lemma 5.* We denote $v^{\mathrm{KL}}(\boldsymbol{\theta}, Q_t)$ as $\mathbb{E}_{Q_t}[\log p_{\boldsymbol{\theta}}(\boldsymbol{z})]$, $\mathcal{G}^{\mathrm{KL}}(\boldsymbol{\theta}, t) := \nabla_{\boldsymbol{\theta}} v^{\mathrm{KL}}(\boldsymbol{\theta}, Q_t)$ and $\boldsymbol{\theta}_t^{\mathrm{KL}} := \operatorname{argmin} v^{\mathrm{KL}}(\boldsymbol{\theta}, Q_t)$. Note that $\mathcal{G}^{\mathrm{KL}}(\boldsymbol{\theta}_t^{\mathrm{KL}}, t) = 0$ for all $t$. By implicit function theorem,

$$\lim_{t \to 0} \frac{1}{t} \left( \boldsymbol{\theta}_t^{\mathrm{KL}} - \boldsymbol{\theta}_0 \right) = -[\nabla_{\boldsymbol{\theta}} \mathcal{G}^{\mathrm{KL}}(\boldsymbol{\theta}_0, 0)]^{-1} \frac{\partial}{\partial t} \nabla_{\boldsymbol{\theta}} v^{\mathrm{KL}}(\boldsymbol{\theta}_0, Q_t)|_{t=0}$$

$$= -\nabla_{\boldsymbol{\theta}\boldsymbol{\theta}} \mathbb{E}_{\boldsymbol{\theta}_0}[\log p_{\boldsymbol{\theta}_0}(\boldsymbol{z})]^{-1} \frac{\partial}{\partial t} \nabla_{\boldsymbol{\theta}} v^{\mathrm{KL}}(\boldsymbol{\theta}_0, Q_t)|_{t=0}$$

$$= \boldsymbol{I}^{-1} \frac{\partial}{\partial t} \int \boldsymbol{s}_{\boldsymbol{\theta}_0}(\boldsymbol{z}) dQ_t(\boldsymbol{z})|_{t=0}$$

$$= \boldsymbol{I}^{-1} \int \boldsymbol{s}_{\boldsymbol{\theta}_0}(\boldsymbol{z}) \frac{\partial}{\partial t} dQ_t(\boldsymbol{z})|_{t=0}.$$

At $t = 0$, by (3),

$$\frac{\partial}{\partial t} q_t(\boldsymbol{z}) \Big|_{t=0} = q_0(\boldsymbol{z}) u(\boldsymbol{z}).$$

In conclusion,

$$\boldsymbol{I}^{-1} \int \boldsymbol{s}_{\boldsymbol{\theta}_0}(\boldsymbol{z}) \frac{\partial}{\partial t} dQ_t(\boldsymbol{z})|_{t=0}$$

$$= \boldsymbol{I}^{-1} \int \boldsymbol{s}_{\boldsymbol{\theta}_0}(\boldsymbol{z}) \left\{ q_0(\boldsymbol{z}) h^\top u(\boldsymbol{z}) \right\} d\boldsymbol{z}.$$

Since

$$\int \boldsymbol{s}_{\boldsymbol{\theta}_0}(\boldsymbol{z}) \left[ \int q_0(\boldsymbol{z}) u(\boldsymbol{z}) d\boldsymbol{z} \right] q_0(\boldsymbol{z}) d\boldsymbol{z} = \left[ \int q_0(\boldsymbol{z}) u(\boldsymbol{z}) d\boldsymbol{z} \right] \mathbb{E}_{\boldsymbol{\theta}_0} \left[ \boldsymbol{s}_{\boldsymbol{\theta}_0}(\boldsymbol{z}) \right] = \boldsymbol{0}$$

and

$$\int \boldsymbol{s}_{\boldsymbol{\theta}_0}(\boldsymbol{z}) q_0(\boldsymbol{z}) u(\boldsymbol{z}) d\boldsymbol{z} = \mathbb{E}_{\boldsymbol{\theta}_0}[u(\boldsymbol{z}) \boldsymbol{s}_{\boldsymbol{\theta}_0}(\boldsymbol{z})],$$

we have

$$\lim_{t \to 0} \frac{1}{t} \left( \boldsymbol{\theta}_t^{\mathrm{KL}} - \boldsymbol{\theta}_0 \right) = \boldsymbol{I}^{-1} \int \boldsymbol{s}_{\boldsymbol{\theta}_0}(\boldsymbol{z}) \frac{\partial}{\partial t} dQ_t(\boldsymbol{z})|_{t=0} = \boldsymbol{I}^{-1} \mathbb{E}_{\boldsymbol{\theta}_0}[u(\boldsymbol{z}) \boldsymbol{s}_{\boldsymbol{\theta}_0}(\boldsymbol{z})].$$

Similarly, we denote $v^*(\boldsymbol{\theta}, Q_t)$ as $\mathbb{E}_{Q_t}[c(\boldsymbol{w}_{\boldsymbol{\theta}}, \boldsymbol{z})]$, $\mathcal{G}^*(\boldsymbol{\theta}, t) := \nabla_{\boldsymbol{\theta}} v^*(\boldsymbol{\theta}, Q_t)$ and $\boldsymbol{\theta}_t^* := \operatorname{argmin} v^*(\boldsymbol{\theta}, Q_t)$. Note that $\mathcal{G}^*(\boldsymbol{\theta}_t^*, t) = 0$ for all $t$. By implicit function theorem,

$$\lim_{t \to 0} \frac{1}{t} \left( \boldsymbol{\theta}_t^* - \boldsymbol{\theta}_0 \right) = -[\nabla_{\boldsymbol{\theta}} \mathcal{G}^*(\boldsymbol{\theta}_0, 0)]^{-1} \frac{\partial}{\partial t} \nabla_{\boldsymbol{\theta}} v^*(\boldsymbol{\theta}_0, Q_t)|_{t=0}$$

$$= -\nabla_{\boldsymbol{\theta}\boldsymbol{\theta}} \mathbb{E}_{\boldsymbol{\theta}_0}[c(\boldsymbol{w}_{\boldsymbol{\theta}_0}, \boldsymbol{z})]^{-1} \frac{\partial}{\partial t} \nabla_{\boldsymbol{\theta}} v^*(\boldsymbol{\theta}_0, Q_t)|_{t=0}$$

$$= -\boldsymbol{\Phi}^{-1} \frac{\partial}{\partial t} \int \nabla_{\boldsymbol{\theta}} c(\boldsymbol{w}_{\boldsymbol{\theta}_0}, \boldsymbol{z}) dQ_t(\boldsymbol{z})|_{t=0}$$

$$= -\boldsymbol{\Phi}^{-1} \int \nabla_{\boldsymbol{\theta}} c(\boldsymbol{w}_{\boldsymbol{\theta}_0}, \boldsymbol{z}) \frac{\partial}{\partial t} dQ_t(\boldsymbol{z})|_{t=0}.$$

At $t = 0$, by (3),

$$\frac{\partial}{\partial t} q_t(\boldsymbol{z}) \Big|_{t=0} = q_0(\boldsymbol{z}) u(\boldsymbol{z}).$$

In conclusion,

$$- \boldsymbol{\Phi}^{-1} \int \nabla_{\boldsymbol{\theta}} c(\boldsymbol{w}_{\boldsymbol{\theta}_0}, \boldsymbol{z}) \frac{\partial}{\partial t} dQ_t(\boldsymbol{z})|_{t=0}$$

$$= - \boldsymbol{\Phi}^{-1} \int \nabla_{\boldsymbol{\theta}} c(\boldsymbol{w}_{\boldsymbol{\theta}_0}, \boldsymbol{z}) \left\{ q_0(\boldsymbol{z}) u(\boldsymbol{z}) \right\} d\boldsymbol{z}.$$

Since

$$\int \nabla_{\boldsymbol{\theta}} c(\boldsymbol{w}_{\boldsymbol{\theta}_0}, \boldsymbol{z}) q_0(\boldsymbol{z}) u(\boldsymbol{z}) d\boldsymbol{z} = \mathbb{E}_{\boldsymbol{\theta}_0} [u(\boldsymbol{z}) \nabla_{\boldsymbol{\theta}} c(\boldsymbol{w}_{\boldsymbol{\theta}_0}, \boldsymbol{z})],$$

we have

$$\lim_{t \to 0} \frac{1}{t} (\boldsymbol{\theta}_t^* - \boldsymbol{\theta}_0) = - \boldsymbol{\Phi}^{-1} \int \nabla_{\boldsymbol{\theta}} c(\boldsymbol{w}_{\boldsymbol{\theta}_0}, \boldsymbol{z}) \frac{\partial}{\partial t} dQ_t(\boldsymbol{z})|_{t=0} = - \boldsymbol{\Phi}^{-1} \mathbb{E}_{\boldsymbol{\theta}_0} [u(\boldsymbol{z}) \nabla_{\boldsymbol{\theta}} c(\boldsymbol{w}_{\boldsymbol{\theta}_0}, \boldsymbol{z})].$$

$\square$

We remark that Lemma 5 also holds for $Q_t$ in Example 6 and 7.

# B  Proofs

In this section, we supplement the proof of the results in this paper.

*Proof of Theorem 6.* We first notice the fact that, in the mildly misspecified regime, by defining $h_n = 1/(n^{\alpha - 1/2}) = o(1)$, we have

$$\log \frac{dQ^n(\boldsymbol{z}_1, ..., \boldsymbol{z}_n)}{dP^n(\boldsymbol{z}_1, ..., \boldsymbol{z}_n)} = \frac{1}{\sqrt{n}} \sum_{i=1}^{n} h_n u(\boldsymbol{z}_i) - \frac{1}{2} \mathbb{E}_{\boldsymbol{\theta}_0}[u^2] h_n^2 + o_{P^n}(h_n) = o_{P^n}(1).$$

In the mild misspecified case, under $P^n$, we have a joint central limit theorem

$$\begin{bmatrix} \sqrt{n}(\hat{\boldsymbol{w}}^{\square} - \boldsymbol{w}_{\boldsymbol{\theta}_0}) \\ \log \frac{dQ^n}{dP^n} \end{bmatrix} \xrightarrow{P^n} N \left( \begin{bmatrix} 0 \\ 0 \end{bmatrix}, \begin{bmatrix} \mathrm{var}_{\boldsymbol{\theta}_0}(\mathrm{IF}^{\square}(\boldsymbol{z})) & 0 \\ 0 & 0 \end{bmatrix} \right).$$

Using LeCam's third lemma, we change the measure from $P^n$ to $Q^n$ and get that under $Q^n$,

$$\sqrt{n}(\hat{\boldsymbol{w}}^{\square} - \boldsymbol{w}_{\boldsymbol{\theta}_0}) \xrightarrow{Q^n} N(0, \mathrm{var}_{\boldsymbol{\theta}_0}(\mathrm{IF}^{\square}(\boldsymbol{z}))).$$

Using the same technique,

$$n^{\alpha}(\boldsymbol{w}_n^* - \boldsymbol{w}_{\boldsymbol{\theta}_0}) \to \mathbb{E}_{\boldsymbol{\theta}_0}[u(\boldsymbol{z}) \mathrm{IF}^{\mathrm{SAA}}(\boldsymbol{z})],$$

$$\sqrt{n}(\boldsymbol{w}_n^* - \boldsymbol{w}_{\boldsymbol{\theta}_0}) \to \boldsymbol{0}.$$

In conclusion,

$$\sqrt{n}(\hat{\boldsymbol{w}}^{\square} - \boldsymbol{w}_n^*) = \sqrt{n}(\hat{\boldsymbol{w}}^{\square} - \boldsymbol{w}_{\boldsymbol{\theta}_0}) - \sqrt{n}(\boldsymbol{w}_n^* - \boldsymbol{w}_{\boldsymbol{\theta}_0}) \xrightarrow{Q^n} N(0, \mathrm{var}_{\boldsymbol{\theta}_0}(\mathrm{IF}^{\square}(\boldsymbol{z}))).$$

Let us now consider the regret. We use Taylor expansion of the regret with respect to $\boldsymbol{w}$ at $\boldsymbol{w}_n^*$ and note that $\nabla_{\boldsymbol{w}} v_n(\boldsymbol{w}_n^*) = 0$ for every $n$,

$$v_n(\hat{\boldsymbol{w}}^{\square}) - v_n(\boldsymbol{w}_n^*) = \frac{1}{2}(\hat{\boldsymbol{w}}^{\square} - \boldsymbol{w}_n^*)^{\top} \nabla_{\boldsymbol{w}\boldsymbol{w}} v_n(\boldsymbol{w}_n^*)(\hat{\boldsymbol{w}}^{\square} - \boldsymbol{w}_n^*) + o_{Q^n}(\| \hat{\boldsymbol{w}}^{\square} - \boldsymbol{w}_n^* \|^2),$$

$$n(v_n(\hat{\boldsymbol{w}}^{\square}) - v_n(\boldsymbol{w}_n^*)) = \frac{1}{2}\sqrt{n}(\hat{\boldsymbol{w}}^{\square} - \boldsymbol{w}_n^*)^{\top} \nabla_{\boldsymbol{w}\boldsymbol{w}} v_n(\boldsymbol{w}_n^*)\sqrt{n}(\hat{\boldsymbol{w}}^{\square} - \boldsymbol{w}_n^*) + o_{Q^n}(1).$$

By Assumption 2 that $\nabla_{\boldsymbol{w}\boldsymbol{w}} v_n(\boldsymbol{w}_n^*) \to \boldsymbol{V}$, the function $f : \Omega \to \mathbb{R}$ with $f(\cdot) := \frac{1}{2}(\cdot)^{\top} \boldsymbol{V}(\cdot)$ and function sequence $f_n : \Omega \to \mathbb{R}$ with $f_n(\cdot) := \frac{1}{2}(\cdot)^{\top} \nabla_{\boldsymbol{w}\boldsymbol{w}} v_n(\boldsymbol{w}_n^*)(\cdot)$ satisfy: for all sequence $\{\boldsymbol{w}_n\}_{n=1}^{\infty}$, if $\boldsymbol{w}_n \to \boldsymbol{w}$ for some $\boldsymbol{w} \in \Omega$, then $f_n(\boldsymbol{w}_n) \to f(\boldsymbol{w})$ since continuity is preserved under multiplication. Using the extended continuous mapping theorem (Theorem 1.11.1 in van der Vaart and Wellner [1996]), we have under $Q^n$,

$$n(v_n(\hat{\boldsymbol{w}}^{\square}) - v_n(\boldsymbol{w}_n^*)) \xrightarrow{Q^n} \frac{1}{2} N^{\square} \boldsymbol{V} N^{\square}.$$

Moreover, ETO is stochastically dominated by IEO and IEO is stochastically dominated by SAA. $\square$

*Proof of Proposition 1.* The asymptotic normality result is directly from van der Vaart [2000] by noting Lemma 1.

The asymptotic normality of SAA is by Proposition 2A of Elmachtoub et al. [2023]. For ETO and IEO, Proposition 2B and 2C of Elmachtoub et al. [2023] shows that

$$\sqrt{n}\left(\hat{\boldsymbol{\theta}}^{\text{ETO}} - \boldsymbol{\theta}_0\right) \overset{P^n}{\to} N(\boldsymbol{0}, \boldsymbol{I}^{-1}),$$
$$\sqrt{n}\left(\hat{\boldsymbol{\theta}}^{\text{IEO}} - \boldsymbol{\theta}_0\right) \overset{P^n}{\to} N(\boldsymbol{0}, \boldsymbol{\Phi}^{-1}\,\text{var}_{\boldsymbol{\theta}_0}\left(\nabla_{\boldsymbol{\theta}} c(\boldsymbol{w}_{\boldsymbol{\theta}_0}, \boldsymbol{z})\right)\boldsymbol{\Phi}^{-1}).$$

Regarding the notation, $\text{var}_{\boldsymbol{\theta}_0}\left(\nabla_{\boldsymbol{\theta}} c(\boldsymbol{w}_{\boldsymbol{\theta}_0}, \boldsymbol{z}))\right)$ is the variance of the random gradient $\nabla_{\boldsymbol{\theta}} c(\boldsymbol{w}_{\boldsymbol{\theta}}, \boldsymbol{z})$ at $\boldsymbol{\theta} = \boldsymbol{\theta}_0$, under the distribution $P_{\boldsymbol{\theta}_0}$. Note that the subscript $\boldsymbol{\theta}_0$ under the variance is not a variable here. Using the delta method, we have

$$\sqrt{n}\left(\hat{\boldsymbol{\theta}}^{\text{ETO}} - \boldsymbol{\theta}_0\right) \overset{P^n}{\to} N(\boldsymbol{0}, \nabla_{\boldsymbol{\theta}} \boldsymbol{w}_{\boldsymbol{\theta}_0}^{\top} \boldsymbol{I}^{-1} \nabla_{\boldsymbol{\theta}} \boldsymbol{w}_{\boldsymbol{\theta}_0}) = N(\boldsymbol{0}, \boldsymbol{V}^{-1} \boldsymbol{\Sigma} \boldsymbol{I}^{-1} \boldsymbol{\Sigma}^{\top} \boldsymbol{V}^{-1}) = N(0, \text{var}_{\boldsymbol{\theta}_0}(\text{IF}^{\text{ETO}}(\boldsymbol{z}))),$$

$$\sqrt{n}\left(\hat{\boldsymbol{\theta}}^{\text{IEO}} - \boldsymbol{\theta}_0\right) \overset{P^n}{\to} N(\boldsymbol{0}, \nabla_{\boldsymbol{\theta}} \boldsymbol{w}_{\boldsymbol{\theta}_0}^{\top} \boldsymbol{\Phi}^{-1} \text{var}_{\boldsymbol{\theta}_0}\left(\nabla_{\boldsymbol{\theta}} c(\boldsymbol{w}_{\boldsymbol{\theta}_0}, \boldsymbol{z})\right) \boldsymbol{\Phi}^{-1} \nabla_{\boldsymbol{\theta}} \boldsymbol{w}_{\boldsymbol{\theta}_0})$$
$$= N(\boldsymbol{0}, \boldsymbol{V}^{-1} \boldsymbol{\Sigma} \boldsymbol{\Phi}^{-1} \text{var}_{\boldsymbol{\theta}_0}\left(\nabla_{\boldsymbol{\theta}} c(\boldsymbol{w}_{\boldsymbol{\theta}_0}, \boldsymbol{z})\right) \boldsymbol{\Phi}^{-1} \boldsymbol{\Sigma}^{\top} \boldsymbol{V}^{-1})$$
$$= N(0, \text{var}_{\boldsymbol{\theta}_0}(\text{IF}^{\text{IEO}}(\boldsymbol{z}))).$$

The inequality $\text{var}_{\boldsymbol{\theta}_0}(\text{IF}^{\text{ETO}}(\boldsymbol{z})) \leq \text{var}_{\boldsymbol{\theta}_0}(\text{IF}^{\text{IEO}}(\boldsymbol{z})) \leq \text{var}_{\boldsymbol{\theta}_0}(\text{IF}^{\text{SAA}}(\boldsymbol{z}))$ is from Theorem 2 of Elmachtoub et al. [2023]. □

*Proof of Theorem 3.* We use a different decomposition framework this time. We recall

$$\boldsymbol{\theta}_t^{\text{KL}} := \underset{\boldsymbol{\theta} \in \Theta}{\arg\max}\, \mathbb{E}_{Q_t}[\log p_{\boldsymbol{\theta}}(\boldsymbol{z})],$$
$$\boldsymbol{\theta}_t^* := \underset{\boldsymbol{\theta} \in \Theta}{\arg\min}\, \mathbb{E}_{Q_t}[c(\boldsymbol{w}_{\boldsymbol{\theta}}, \boldsymbol{z})],$$
$$\boldsymbol{w}_t^* := \underset{\boldsymbol{w} \in \Omega}{\arg\min}\, \mathbb{E}_{Q_t}[c(\boldsymbol{w}, \boldsymbol{z})].$$

We denote $t_n := 1/n^{\alpha}$. Note that here $\boldsymbol{w}_{t_n}^* = \boldsymbol{w}_n^*$ but generally $\boldsymbol{w}_{\boldsymbol{\theta}_{t_n}^{\text{KL}}} \neq \boldsymbol{w}_n^*, \boldsymbol{w}_{\boldsymbol{\theta}_{t_n}^*} \neq \boldsymbol{w}_n^*$. In this case,

$$\hat{\boldsymbol{w}}^{\text{ETO}} - \boldsymbol{w}_n^* = (\hat{\boldsymbol{w}}^{\text{ETO}} - \boldsymbol{w}_{\boldsymbol{\theta}_{t_n}^{\text{KL}}}) + (\boldsymbol{w}_{\boldsymbol{\theta}_{t_n}^{\text{KL}}} - \boldsymbol{w}_{\boldsymbol{\theta}_0}) - (\boldsymbol{w}_n^* - \boldsymbol{w}_{\boldsymbol{\theta}_0}),$$
$$\hat{\boldsymbol{w}}^{\text{IEO}} - \boldsymbol{w}_n^* = (\hat{\boldsymbol{w}}^{\text{IEO}} - \boldsymbol{w}_{\boldsymbol{\theta}_{t_n}^*}) + (\boldsymbol{w}_{\boldsymbol{\theta}_{t_n}^*} - \boldsymbol{w}_{\boldsymbol{\theta}_0}) - (\boldsymbol{w}_n^* - \boldsymbol{w}_{\boldsymbol{\theta}_0}),$$
$$\hat{\boldsymbol{w}}^{\text{SAA}} - \boldsymbol{w}_n^* = (\hat{\boldsymbol{w}}^{\text{SAA}} - \boldsymbol{w}_{t_n}^*) + (\boldsymbol{w}_{t_n}^* - \boldsymbol{w}_{\boldsymbol{\theta}_0}) - (\boldsymbol{w}_n^* - \boldsymbol{w}_{\boldsymbol{\theta}_0}).$$

We already show in Lemma 4 that

$$n^{\alpha}(\boldsymbol{w}_n^* - \boldsymbol{w}_{\boldsymbol{\theta}_0}) \to \mathbb{E}_{\boldsymbol{\theta}_0}[u(\boldsymbol{z})\text{IF}^{\text{SAA}}(\boldsymbol{z})].$$

Next we give a limit of the middle term, using Taylor expansion. For SAA, the middle term is equal to the third term. For ETO and IEO, $\boldsymbol{w}_{\boldsymbol{\theta}_0} = \boldsymbol{w}_{\boldsymbol{\theta}_t^*}|_{t=0}$ and $\boldsymbol{w}_{\boldsymbol{\theta}_0} = \boldsymbol{w}_{\boldsymbol{\theta}_t^{\text{KL}}}|_{t=0}$.

$$\boldsymbol{w}_{\boldsymbol{\theta}_t^*} - \boldsymbol{w}_{\boldsymbol{\theta}_0} := \nabla_t \boldsymbol{w}_{\boldsymbol{\theta}_t^*} + o(t) = \nabla_{\boldsymbol{\theta}} \boldsymbol{w}_{\boldsymbol{\theta}}^{\top} \nabla_t \boldsymbol{\theta}_t^* + o(t),$$
$$\boldsymbol{w}_{\boldsymbol{\theta}_t^{\text{KL}}} - \boldsymbol{w}_{\boldsymbol{\theta}_0} := \nabla_t \boldsymbol{w}_{\boldsymbol{\theta}_t^{\text{KL}}} + o(t) = \nabla_{\boldsymbol{\theta}} \boldsymbol{w}_{\boldsymbol{\theta}}^{\top} \nabla_t \boldsymbol{\theta}_t^{\text{KL}} + o(t).$$

By Lemma 5, we can get $\nabla_t \boldsymbol{\theta}_t^*$ and $\nabla_t \boldsymbol{\theta}_t^{\text{KL}}$ at $t = 0$:

$$\nabla_t \boldsymbol{\theta}_t^{\text{KL}} = \boldsymbol{I}^{-1} \mathbb{E}_{\boldsymbol{\theta}_0}[u(\boldsymbol{z})\boldsymbol{s}_{\boldsymbol{\theta}_0}(\boldsymbol{z})],$$
$$\nabla_t \boldsymbol{\theta}_t^* = -\boldsymbol{\Phi}^{-1} \mathbb{E}_{\boldsymbol{\theta}_0}[u(\boldsymbol{z})\nabla_{\boldsymbol{\theta}} c(\boldsymbol{w}_{\boldsymbol{\theta}_0}, \boldsymbol{z})].$$

Moreover,

$$\nabla_t \boldsymbol{w}_{\boldsymbol{\theta}_t^{\text{KL}}} = \nabla_{\boldsymbol{\theta}} \boldsymbol{w}_{\boldsymbol{\theta}}^{\top} \nabla_t \boldsymbol{\theta}_t^{\text{KL}} = -\boldsymbol{V}^{-1} \boldsymbol{\Sigma} \boldsymbol{I}^{-1} \mathbb{E}_{\boldsymbol{\theta}_0}(u(\boldsymbol{z})\boldsymbol{s}_{\boldsymbol{\theta}_0}(\boldsymbol{z})) = \mathbb{E}_{\boldsymbol{\theta}_0}(u(\boldsymbol{z})\text{IF}^{\text{ETO}}(\boldsymbol{z})),$$
$$\nabla_t \boldsymbol{w}_{\boldsymbol{\theta}_t^*} = \nabla_{\boldsymbol{\theta}} \boldsymbol{w}_{\boldsymbol{\theta}}^{\top} \nabla_t \boldsymbol{\theta}_t^* = \boldsymbol{V}^{-1} \boldsymbol{\Sigma} \boldsymbol{\Phi}^{-1} \mathbb{E}_{\boldsymbol{\theta}_0}[u(\boldsymbol{z})\nabla_{\boldsymbol{\theta}} c(\boldsymbol{w}_{\boldsymbol{\theta}_0}, \boldsymbol{z})] = \mathbb{E}_{\boldsymbol{\theta}_0}(u(\boldsymbol{z})\text{IF}^{\text{IEO}}(\boldsymbol{z})).$$

Finally, for the middle term,

$$n^{\alpha}(\boldsymbol{w}_{\boldsymbol{\theta}^*_{t_n}} - \boldsymbol{w}_{\boldsymbol{\theta}_0}) \to \mathbb{E}_{\boldsymbol{\theta}_0}(u(\boldsymbol{z})\mathrm{IF}^{\mathrm{IEO}}(\boldsymbol{z})),$$

$$n^{\alpha}(\boldsymbol{w}_{\boldsymbol{\theta}^{\mathrm{KL}}_{t_n}} - \boldsymbol{w}_{\boldsymbol{\theta}_0}) \to \mathbb{E}_{\boldsymbol{\theta}_0}(u(\boldsymbol{z})\mathrm{IF}^{\mathrm{ETO}}(\boldsymbol{z})),$$

$$n^{\alpha}(\boldsymbol{w}^*_{t_n} - \boldsymbol{w}_{\boldsymbol{\theta}_0}) \to \mathbb{E}_{\boldsymbol{\theta}_0}(u(\boldsymbol{z})\mathrm{IF}^{\mathrm{SAA}}(\boldsymbol{z})).$$

For the first term, under Assumption 2 $(\hat{\boldsymbol{w}}^{\mathrm{ETO}} - \boldsymbol{w}_{\boldsymbol{\theta}^{\mathrm{KL}}_{t_n}})$, $(\hat{\boldsymbol{w}}^{\mathrm{IEO}} - \boldsymbol{w}_{\boldsymbol{\theta}^*_{t_n}})$ and $(\hat{\boldsymbol{w}}^{\mathrm{SAA}} - \boldsymbol{w}^*_{t_n})$ are all $O_{Q^n}(1/\sqrt{n})$, then

$$n^{\alpha}(\hat{\boldsymbol{w}}^{\mathrm{ETO}} - \boldsymbol{w}_{\boldsymbol{\theta}^{\mathrm{KL}}_{t_n}}) \xrightarrow{p} 0,$$

$$n^{\alpha}(\hat{\boldsymbol{w}}^{\mathrm{IEO}} - \boldsymbol{w}_{\boldsymbol{\theta}^*_{t_n}}) \xrightarrow{p} 0,$$

$$n^{\alpha}(\hat{\boldsymbol{w}}^{\mathrm{SAA}} - \boldsymbol{w}^*_{t_n}) \xrightarrow{p} 0.$$

When we multiply $n^{\alpha}$, the term shrinks in probability to 0. In conclusion,

$$n^{\alpha}(\hat{\boldsymbol{w}}^{\square} - \boldsymbol{w}^*_n) \xrightarrow{p} \boldsymbol{b}^{\square}.$$

Let us now consider the regret. We use Taylor expansion of the regret with respect to $\boldsymbol{w}$ at $\boldsymbol{w}^*_n$ and note that $\nabla_{\boldsymbol{w}} v_n(\boldsymbol{w}^*_n) = 0$ for every $n$,

$$v_n(\hat{\boldsymbol{w}}^{\square}) - v_n(\boldsymbol{w}^*_n) = \frac{1}{2}(\hat{\boldsymbol{w}}^{\square} - \boldsymbol{w}^*_n)^{\top} \nabla_{\boldsymbol{w}\boldsymbol{w}} v_n(\boldsymbol{w}^*_n)(\hat{\boldsymbol{w}}^{\square} - \boldsymbol{w}^*_n) + o_{Q^n}\left(\left\|\hat{\boldsymbol{w}}^{\square} - \boldsymbol{w}^*_n\right\|^2\right),$$

$$n^{2\alpha}(v_n(\hat{\boldsymbol{w}}^{\square}) - v_n(\boldsymbol{w}^*_n)) = \frac{1}{2}n^{\alpha}(\hat{\boldsymbol{w}}^{\square} - \boldsymbol{w}^*_n)^{\top} \nabla_{\boldsymbol{w}\boldsymbol{w}} v_n(\boldsymbol{w}^*_n)n^{\alpha}(\hat{\boldsymbol{w}}^{\square} - \boldsymbol{w}^*_n) + o_{Q^n}(1).$$

By Assumption 2 that $\nabla_{\boldsymbol{w}\boldsymbol{w}} v_n(\boldsymbol{w}^*_n) \to \boldsymbol{V}$, the function $f : \Omega \to \mathbb{R}$ with $f(\cdot) := \frac{1}{2}(\cdot)^{\top}\boldsymbol{V}(\cdot)$ and function sequence $f_n : \Omega \to \mathbb{R}$ with $f_n(\cdot) := \frac{1}{2}(\cdot)^{\top}\nabla_{\boldsymbol{w}\boldsymbol{w}} v_n(\boldsymbol{w}^*_n)(\cdot)$ satisfy: for all sequence $\{\boldsymbol{w}_n\}_{n=1}^{\infty}$, if $\boldsymbol{w}_n \to \boldsymbol{w}$ for some $\boldsymbol{w} \in \Omega$, then $f_n(\boldsymbol{w}_n) \to f(\boldsymbol{w})$ since continuity is preserved under multiplication. Using the extended continuous mapping theorem (Theorem 1.11.1 in van der Vaart and Wellner [1996]), we have under $Q^n$,

$$n^{2\alpha}(v_n(\hat{\boldsymbol{w}}^{\square}) - v_n(\boldsymbol{w}^*_n)) \xrightarrow{p} \frac{1}{2}\left(\boldsymbol{b}^{\square}\right)^{\top}\boldsymbol{V}\boldsymbol{b}^{\square}.$$

$\square$

*Proof of Theorem 4.* Recall the influence function of SAA, ETO, IEO:

$$\mathrm{IF}^{\mathrm{SAA}}(\boldsymbol{z}) = -\boldsymbol{V}^{-1}\nabla_{\boldsymbol{w}} c(\boldsymbol{w}_{\boldsymbol{\theta}_0}, \boldsymbol{z}),$$

$$\mathrm{IF}^{\mathrm{IEO}}(\boldsymbol{z}) = \boldsymbol{V}^{-1}\boldsymbol{\Sigma}(\boldsymbol{\Sigma}^{\top}\boldsymbol{V}^{-1}\boldsymbol{\Sigma})^{-1}\boldsymbol{\Sigma}^{\top}\boldsymbol{V}^{-1}\nabla_{\boldsymbol{w}} c(\boldsymbol{w}_{\boldsymbol{\theta}_0}, \boldsymbol{z}) = -\boldsymbol{V}^{-1}P_{\boldsymbol{\Sigma},\boldsymbol{V}}\nabla_{\boldsymbol{w}} c(\boldsymbol{w}_{\boldsymbol{\theta}_0}, \boldsymbol{z}),$$

$$\mathrm{IF}^{\mathrm{ETO}}(\boldsymbol{z}) = -\boldsymbol{V}^{-1}\boldsymbol{\Sigma}\boldsymbol{I}^{-1}\boldsymbol{s}_{\boldsymbol{\theta}_0}(\boldsymbol{z}) = -\boldsymbol{V}^{-1}\boldsymbol{\Sigma}\mathbb{E}_{\boldsymbol{\theta}_0}[\boldsymbol{s}_{\boldsymbol{\theta}_0}(\boldsymbol{z})\boldsymbol{s}_{\boldsymbol{\theta}_0}(\boldsymbol{z})^{\top}]\boldsymbol{s}_{\boldsymbol{\theta}_0}(\boldsymbol{z}).$$

For regret comparison, since $\boldsymbol{b}^{\mathrm{SAA}} = \boldsymbol{0}$, we have $R^{\mathrm{SAA}} = 0$. Also, $R^{\mathrm{IEO}} \geq 0$ and $R^{\mathrm{ETO}} \geq 0$.

By noting that $\boldsymbol{V} = \nabla_{\boldsymbol{w}\boldsymbol{w}}\mathbb{E}_{\boldsymbol{\theta}_0}[c(\boldsymbol{w}_{\boldsymbol{\theta}_0}, \boldsymbol{z})]$, we observe that

$$\mathbb{E}_{\boldsymbol{\theta}_0}[u(\boldsymbol{z})\mathrm{IF}^{\mathrm{IEO}}(\boldsymbol{z})]^{\top}\boldsymbol{V}\mathbb{E}_{\boldsymbol{\theta}_0}[u(\boldsymbol{z})\mathrm{IF}^{\mathrm{IEO}}(\boldsymbol{z})] = \mathbb{E}_{\boldsymbol{\theta}_0}[u(\boldsymbol{z})\mathrm{IF}^{\mathrm{IEO}}(\boldsymbol{z})]^{\top}\boldsymbol{V}\mathbb{E}_{\boldsymbol{\theta}_0}[u(\boldsymbol{z})\mathrm{IF}^{\mathrm{SAA}}(\boldsymbol{z})].$$

This is because

$$\mathbb{E}_{\boldsymbol{\theta}_0}[u(\boldsymbol{z})\mathrm{IF}^{\mathrm{IEO}}(\boldsymbol{z})]^{\top}\boldsymbol{V}\mathbb{E}_{\boldsymbol{\theta}_0}[u(\boldsymbol{z})\mathrm{IF}^{\mathrm{IEO}}(\boldsymbol{z})]$$

$$= \left(\mathbb{E}_{\boldsymbol{\theta}_0}[u(\boldsymbol{z})\nabla_{\boldsymbol{w}} c(\boldsymbol{w}_{\boldsymbol{\theta}_0}, \boldsymbol{z})]^{\top}\boldsymbol{V}^{-1}\boldsymbol{\Sigma}\left(\boldsymbol{\Sigma}^{\top}\boldsymbol{V}^{-1}\boldsymbol{\Sigma}\right)^{-1}\boldsymbol{\Sigma}^{\top}\boldsymbol{V}^{-1}\right) \cdot \boldsymbol{V}\cdot$$

$$\left(\boldsymbol{V}^{-1}\boldsymbol{\Sigma}\left(\boldsymbol{\Sigma}^{\top}\boldsymbol{V}^{-1}\boldsymbol{\Sigma}\right)^{-1}\boldsymbol{\Sigma}^{\top}\boldsymbol{V}^{-1}\mathbb{E}_{\boldsymbol{\theta}_0}[u(\boldsymbol{z})\nabla_{\boldsymbol{w}} c(\boldsymbol{w}_{\boldsymbol{\theta}_0}, \boldsymbol{z})]\right)$$

$$= \mathbb{E}_{\boldsymbol{\theta}_0}[u(\boldsymbol{z})\nabla_{\boldsymbol{w}} c(\boldsymbol{w}_{\boldsymbol{\theta}_0}, \boldsymbol{z})]^{\top}\boldsymbol{V}^{-1}\boldsymbol{\Sigma}\left(\boldsymbol{\Sigma}^{\top}\boldsymbol{V}^{-1}\boldsymbol{\Sigma}\right)^{-1}\boldsymbol{\Sigma}^{\top}\boldsymbol{V}^{-1}\mathbb{E}_{\boldsymbol{\theta}_0}[u(\boldsymbol{z})\nabla_{\boldsymbol{w}} c(\boldsymbol{w}_{\boldsymbol{\theta}_0}, \boldsymbol{z})]$$

$$= \mathbb{E}_{\boldsymbol{\theta}_0}[u(\boldsymbol{z})\nabla_{\boldsymbol{w}} c(\boldsymbol{w}_{\boldsymbol{\theta}_0}, \boldsymbol{z})]^{\top}\boldsymbol{V}^{-1}\boldsymbol{V}\boldsymbol{V}^{-1}\boldsymbol{\Sigma}\left(\boldsymbol{\Sigma}^{\top}\boldsymbol{V}^{-1}\boldsymbol{\Sigma}\right)^{-1}\boldsymbol{\Sigma}^{\top}\boldsymbol{V}^{-1}\mathbb{E}_{\boldsymbol{\theta}_0}[u(\boldsymbol{z})\nabla_{\boldsymbol{w}} c(\boldsymbol{w}_{\boldsymbol{\theta}_0}, \boldsymbol{z})]$$

$$= \mathbb{E}_{\boldsymbol{\theta}_0}[u(\boldsymbol{z})\mathrm{IF}^{\mathrm{SAA}}(\boldsymbol{z})]^{\top}\boldsymbol{V}\mathbb{E}_{\boldsymbol{\theta}_0}[u(\boldsymbol{z})\mathrm{IF}^{\mathrm{IEO}}(\boldsymbol{z})]$$

$$= \mathbb{E}_{\boldsymbol{\theta}_0}[u(\boldsymbol{z})\mathrm{IF}^{\mathrm{IEO}}(\boldsymbol{z})]^{\top}\boldsymbol{V}\mathbb{E}_{\boldsymbol{\theta}_0}[u(\boldsymbol{z})\mathrm{IF}^{\mathrm{SAA}}(\boldsymbol{z})].$$

Now let us prove $R^{\text{ETO}} - R^{\text{IEO}} \geq 0$.

$2R^{\text{ETO}}$

$=\mathbb{E}_{\boldsymbol{\theta}_0}[u(\boldsymbol{z})(\text{IF}^{\text{ETO}}(\boldsymbol{z}) - \text{IF}^{\text{SAA}}(\boldsymbol{z}))]^\top \boldsymbol{V} \mathbb{E}_{\boldsymbol{\theta}_0}[u(\boldsymbol{z})(\text{IF}^{\text{ETO}}(\boldsymbol{z}) - \text{IF}^{\text{SAA}}(\boldsymbol{z}))]$

$=\mathbb{E}_{\boldsymbol{\theta}_0}[u(\boldsymbol{z})\text{IF}^{\text{ETO}}(\boldsymbol{z})]^\top \boldsymbol{V} \mathbb{E}_{\boldsymbol{\theta}_0}[u(\boldsymbol{z})\text{IF}^{\text{ETO}}(\boldsymbol{z})]$

$\quad - 2\mathbb{E}_{\boldsymbol{\theta}_0}[u(\boldsymbol{z})\text{IF}^{\text{ETO}}(\boldsymbol{z})]^\top \boldsymbol{V} \mathbb{E}_{\boldsymbol{\theta}_0}[u(\boldsymbol{z})\text{IF}^{\text{SAA}}(\boldsymbol{z})] + \mathbb{E}_{\boldsymbol{\theta}_0}[u(\boldsymbol{z})\text{IF}^{\text{SAA}}(\boldsymbol{z})]^\top \boldsymbol{V} \mathbb{E}_{\boldsymbol{\theta}_0}[u(\boldsymbol{z})\text{IF}^{\text{SAA}}(\boldsymbol{z})]$

$2R^{\text{IEO}}$

$=\mathbb{E}_{\boldsymbol{\theta}_0}[u(\boldsymbol{z})(\text{IF}^{\text{IEO}}(\boldsymbol{z}) - \text{IF}^{\text{SAA}}(\boldsymbol{z}))]^\top \boldsymbol{V} \mathbb{E}_{\boldsymbol{\theta}_0}[u(\boldsymbol{z})(\text{IF}^{\text{IEO}}(\boldsymbol{z}) - \text{IF}^{\text{SAA}}(\boldsymbol{z}))]$

$=\mathbb{E}_{\boldsymbol{\theta}_0}[u(\boldsymbol{z})\text{IF}^{\text{IEO}}(\boldsymbol{z})]^\top \boldsymbol{V} \mathbb{E}_{\boldsymbol{\theta}_0}[u(\boldsymbol{z})\text{IF}^{\text{IEO}}(\boldsymbol{z})]$

$\quad - 2\mathbb{E}_{\boldsymbol{\theta}_0}[u(\boldsymbol{z})\text{IF}^{\text{IEO}}(\boldsymbol{z})]^\top \boldsymbol{V} \mathbb{E}_{\boldsymbol{\theta}_0}[u(\boldsymbol{z})\text{IF}^{\text{SAA}}(\boldsymbol{z})] + \mathbb{E}_{\boldsymbol{\theta}_0}[u(\boldsymbol{z})\text{IF}^{\text{SAA}}(\boldsymbol{z})]^\top \boldsymbol{V} \mathbb{E}_{\boldsymbol{\theta}_0}[u(\boldsymbol{z})\text{IF}^{\text{SAA}}(\boldsymbol{z})]$

$= - \mathbb{E}_{\boldsymbol{\theta}_0}[u(\boldsymbol{z})\text{IF}^{\text{IEO}}(\boldsymbol{z})]^\top \boldsymbol{V} \mathbb{E}_{\boldsymbol{\theta}_0}[u(\boldsymbol{z})\text{IF}^{\text{SAA}}(\boldsymbol{z})] + \mathbb{E}_{\boldsymbol{\theta}_0}[u(\boldsymbol{z})\text{IF}^{\text{SAA}}(\boldsymbol{z})]^\top \boldsymbol{V} \mathbb{E}_{\boldsymbol{\theta}_0}[u(\boldsymbol{z})\text{IF}^{\text{SAA}}(\boldsymbol{z})].$

Hence,

$2R^{\text{ETO}} - 2R^{\text{IEO}}$

$=\mathbb{E}_{\boldsymbol{\theta}_0}[u(\boldsymbol{z})\text{IF}^{\text{ETO}}(\boldsymbol{z})]^\top \boldsymbol{V} \mathbb{E}_{\boldsymbol{\theta}_0}[u(\boldsymbol{z})\text{IF}^{\text{ETO}}(\boldsymbol{z})]$

$\quad - 2\mathbb{E}_{\boldsymbol{\theta}_0}[u(\boldsymbol{z})\text{IF}^{\text{ETO}}(\boldsymbol{z})]^\top \boldsymbol{V} \mathbb{E}_{\boldsymbol{\theta}_0}[u(\boldsymbol{z})\text{IF}^{\text{SAA}}(\boldsymbol{z})] + \mathbb{E}_{\boldsymbol{\theta}_0}[u(\boldsymbol{z})\text{IF}^{\text{IEO}}(\boldsymbol{z})]^\top \boldsymbol{V} \mathbb{E}_{\boldsymbol{\theta}_0}[u(\boldsymbol{z})\text{IF}^{\text{SAA}}(\boldsymbol{z})]$

$=\mathbb{E}_{\boldsymbol{\theta}_0}[u(\boldsymbol{z})s_{\boldsymbol{\theta}_0}(\boldsymbol{z})]^\top \boldsymbol{I}^{-1} \boldsymbol{\Sigma}^\top \boldsymbol{V}^{-1} \boldsymbol{V} \boldsymbol{V}^{-1} \boldsymbol{\Sigma} \boldsymbol{I}^{-1} \mathbb{E}_{\boldsymbol{\theta}_0}[u(\boldsymbol{z})s_{\boldsymbol{\theta}_0}(\boldsymbol{z})]$

$\quad + 2\mathbb{E}_{\boldsymbol{\theta}_0}[u(\boldsymbol{z})s_{\boldsymbol{\theta}_0}(\boldsymbol{z})]^\top \boldsymbol{I}^{-1} \boldsymbol{\Sigma}^\top \boldsymbol{V}^{-1} \boldsymbol{V} \boldsymbol{V}^{-1} \mathbb{E}_{\boldsymbol{\theta}_0}[u(\boldsymbol{z})\nabla_{\boldsymbol{w}} c(\boldsymbol{w}_{\boldsymbol{\theta}_0}, \boldsymbol{z})]$

$\quad + \mathbb{E}[u(\boldsymbol{z})\nabla_{\boldsymbol{w}} c(\boldsymbol{w}_{\boldsymbol{\theta}_0}, \boldsymbol{z})]^\top \boldsymbol{V}^{-1} \boldsymbol{\Sigma} \left(\boldsymbol{\Sigma}^\top \boldsymbol{V}^{-1} \boldsymbol{\Sigma}\right)^{-1} \boldsymbol{\Sigma}^\top \boldsymbol{V}^{-1} \mathbb{E}_{\boldsymbol{\theta}_0}[u(\boldsymbol{z})\nabla_{\boldsymbol{w}} c(\boldsymbol{w}_{\boldsymbol{\theta}_0}, \boldsymbol{z})]$

$=\mathbb{E}_{\boldsymbol{\theta}_0}[u(\boldsymbol{z})s_{\boldsymbol{\theta}_0}(\boldsymbol{z})]^\top \boldsymbol{I}^{-1} \left(\boldsymbol{\Sigma}^\top \boldsymbol{V}^{-1} \boldsymbol{\Sigma}\right) \boldsymbol{I}^{-1} \mathbb{E}_{\boldsymbol{\theta}_0}[u(\boldsymbol{z})s_{\boldsymbol{\theta}_0}(\boldsymbol{z})]$

$\quad + 2\mathbb{E}_{\boldsymbol{\theta}_0}[u(\boldsymbol{z})s_{\boldsymbol{\theta}_0}(\boldsymbol{z})]^\top \boldsymbol{I}^{-1} \boldsymbol{\Sigma}^\top \boldsymbol{V}^{-1} \mathbb{E}_{\boldsymbol{\theta}_0}[u(\boldsymbol{z})\nabla_{\boldsymbol{w}} c(\boldsymbol{w}_{\boldsymbol{\theta}_0}, \boldsymbol{z})]$

$\quad + \mathbb{E}_{\boldsymbol{\theta}_0}[u(\boldsymbol{z})\nabla_{\boldsymbol{w}} c(\boldsymbol{w}_{\boldsymbol{\theta}_0}, \boldsymbol{z})]^\top \boldsymbol{V}^{-1} \boldsymbol{\Sigma} \left(\boldsymbol{\Sigma}^\top \boldsymbol{V}^{-1} \boldsymbol{\Sigma}\right)^{-1} \boldsymbol{\Sigma}^\top \boldsymbol{V}^{-1} \mathbb{E}_{\boldsymbol{\theta}_0}[u(\boldsymbol{z})\nabla_{\boldsymbol{w}} c(\boldsymbol{w}_{\boldsymbol{\theta}_0}, \boldsymbol{z})]$

$=\left\| \left(\boldsymbol{\Sigma}^\top \boldsymbol{V}^{-1} \boldsymbol{\Sigma}\right)^{1/2} \boldsymbol{I}^{-1} \boldsymbol{\Sigma}^\top \boldsymbol{V}^{-1} \mathbb{E}_{\boldsymbol{\theta}_0}[u(\boldsymbol{z})\nabla_{\boldsymbol{w}} c(\boldsymbol{w}_{\boldsymbol{\theta}_0}, \boldsymbol{z})] - \left(\boldsymbol{\Sigma}^\top \boldsymbol{V}^{-1} \boldsymbol{\Sigma}\right)^{-1/2} \boldsymbol{\Sigma}^\top \boldsymbol{V}^{-1} \mathbb{E}_{\boldsymbol{\theta}_0}[u(\boldsymbol{z})\nabla_{\boldsymbol{w}} c(\boldsymbol{w}_{\boldsymbol{\theta}_0}, \boldsymbol{z})] \right\|^2$

$\geq 0.$

The last equality is from the fact that

$$\boldsymbol{x}^\top \boldsymbol{A} \boldsymbol{x} - 2\boldsymbol{x}^\top \boldsymbol{y} + \boldsymbol{y}^\top \boldsymbol{A}^{-1} \boldsymbol{y} = \left(\boldsymbol{A}^{\frac{1}{2}} \boldsymbol{x} - \boldsymbol{A}^{-\frac{1}{2}} \boldsymbol{y}\right)^\top \left(\boldsymbol{A}^{\frac{1}{2}} \boldsymbol{x} - \boldsymbol{A}^{-\frac{1}{2}} \boldsymbol{y}\right).$$

In conclusion, we have

$$R^{\text{ETO}} \geq R^{\text{IEO}} \geq R^{\text{SAA}} = 0.$$

By the definition of $\boldsymbol{b}^\square$ and $R^\square$, we know $\left\|\boldsymbol{b}^\square\right\|_{\boldsymbol{V}} = \sqrt{2R^\square}$. Hence, by the monotonicity of square root function, we have $0 = \left\|\boldsymbol{b}^{\text{SAA}}\right\|_{\boldsymbol{V}} \leq \left\|\boldsymbol{b}^{\text{IEO}}\right\|_{\boldsymbol{V}} \leq \left\|\boldsymbol{b}^{\text{ETO}}\right\|_{\boldsymbol{V}}.$ $\qquad\square$

*Proof of Theorem 5.* Part (i): We note that when $u(\boldsymbol{z}) = \boldsymbol{\beta}^\top s_{\boldsymbol{\theta}_0}(\boldsymbol{z})$ for some $\boldsymbol{\beta} \in \mathbb{R}^{d_\theta}$,

$\boldsymbol{b}^{\text{ETO}}$

$=\mathbb{E}_{\boldsymbol{\theta}_0}[u(\boldsymbol{z})\text{IF}^{\text{ETO}}(\boldsymbol{z})] - \mathbb{E}_{\boldsymbol{\theta}_0}[u(\boldsymbol{z})\text{IF}^{\text{SAA}}(\boldsymbol{z})]$

$=\boldsymbol{V}^{-1} \mathbb{E}_{\boldsymbol{\theta}_0}[\nabla_{\boldsymbol{w}} c(\boldsymbol{w}_{\boldsymbol{\theta}_0}, \boldsymbol{z}) s_{\boldsymbol{\theta}_0}(\boldsymbol{z})^\top] \left(\mathbb{E}_{\boldsymbol{\theta}_0}[s_{\boldsymbol{\theta}_0}(\boldsymbol{z}) s_{\boldsymbol{\theta}_0}(\boldsymbol{z})^\top]\right)^{-1} \mathbb{E}_{\boldsymbol{\theta}_0}[u(\boldsymbol{z}) s_{\boldsymbol{\theta}_0}(\boldsymbol{z})] - \boldsymbol{V}^{-1} \mathbb{E}_{\boldsymbol{\theta}_0}[u(\boldsymbol{z})\nabla_{\boldsymbol{w}} c(\boldsymbol{w}_{\boldsymbol{\theta}_0}, \boldsymbol{z})]$

$=\boldsymbol{V}^{-1} \mathbb{E}_{\boldsymbol{\theta}_0}[\nabla_{\boldsymbol{w}} c(\boldsymbol{w}_{\boldsymbol{\theta}_0}, \boldsymbol{z}) s_{\boldsymbol{\theta}_0}(\boldsymbol{z})^\top] \left(\mathbb{E}_{\boldsymbol{\theta}_0}[s_{\boldsymbol{\theta}_0}(\boldsymbol{z}) s_{\boldsymbol{\theta}_0}(\boldsymbol{z})^\top]\right)^{-1} \mathbb{E}_{\boldsymbol{\theta}_0}[\boldsymbol{\beta}^\top s_{\boldsymbol{\theta}_0}(\boldsymbol{z}) s_{\boldsymbol{\theta}_0}(\boldsymbol{z})] - \boldsymbol{V}^{-1} \mathbb{E}_{\boldsymbol{\theta}_0}[\boldsymbol{\beta}^\top s_{\boldsymbol{\theta}_0}(\boldsymbol{z})\nabla_{\boldsymbol{w}} c(\boldsymbol{w}_{\boldsymbol{\theta}_0}, \boldsymbol{z})]$

$=\boldsymbol{V}^{-1} \mathbb{E}_{\boldsymbol{\theta}_0}[\nabla_{\boldsymbol{w}} c(\boldsymbol{w}_{\boldsymbol{\theta}_0}, \boldsymbol{z}) s_{\boldsymbol{\theta}_0}(\boldsymbol{z})^\top] \left(\mathbb{E}_{\boldsymbol{\theta}_0}[s_{\boldsymbol{\theta}_0}(\boldsymbol{z}) s_{\boldsymbol{\theta}_0}(\boldsymbol{z})^\top]\right)^{-1} \mathbb{E}_{\boldsymbol{\theta}_0}[s_{\boldsymbol{\theta}_0}(\boldsymbol{z}) s_{\boldsymbol{\theta}_0}(\boldsymbol{z})^\top \boldsymbol{\beta}] - \boldsymbol{V}^{-1} \mathbb{E}_{\boldsymbol{\theta}_0}[\nabla_{\boldsymbol{w}} c(\boldsymbol{w}_{\boldsymbol{\theta}_0}, \boldsymbol{z}) s_{\boldsymbol{\theta}_0}(\boldsymbol{z})^\top \boldsymbol{\beta}]$

$=\boldsymbol{V}^{-1} \mathbb{E}_{\boldsymbol{\theta}_0}[\nabla_{\boldsymbol{w}} c(\boldsymbol{w}_{\boldsymbol{\theta}_0}, \boldsymbol{z}) s_{\boldsymbol{\theta}_0}(\boldsymbol{z})^\top] \left(\mathbb{E}_{\boldsymbol{\theta}_0}[s_{\boldsymbol{\theta}_0}(\boldsymbol{z}) s_{\boldsymbol{\theta}_0}(\boldsymbol{z})^\top]\right)^{-1} \mathbb{E}_{\boldsymbol{\theta}_0}[s_{\boldsymbol{\theta}_0}(\boldsymbol{z}) s_{\boldsymbol{\theta}_0}(\boldsymbol{z})^\top] \boldsymbol{\beta} - \boldsymbol{V}^{-1} \mathbb{E}_{\boldsymbol{\theta}_0}[\nabla_{\boldsymbol{w}} c(\boldsymbol{w}_{\boldsymbol{\theta}_0}, \boldsymbol{z}) s_{\boldsymbol{\theta}_0}(\boldsymbol{z})^\top] \boldsymbol{\beta}$

$=\boldsymbol{V}^{-1} \mathbb{E}_{\boldsymbol{\theta}_0}[\nabla_{\boldsymbol{w}} c(\boldsymbol{w}_{\boldsymbol{\theta}_0}, \boldsymbol{z}) s_{\boldsymbol{\theta}_0}(\boldsymbol{z})^\top] \boldsymbol{\beta} - \boldsymbol{V}^{-1} \mathbb{E}_{\boldsymbol{\theta}_0}[\nabla_{\boldsymbol{w}} c(\boldsymbol{w}_{\boldsymbol{\theta}_0}, \boldsymbol{z}) s_{\boldsymbol{\theta}_0}(\boldsymbol{z})^\top] \boldsymbol{\beta}$

$=\boldsymbol{0}.$

$\qquad\square$

To prove Theorem 1, we need a useful result here. When $\alpha = 1/2$, we can show that the log-likelihood ratio is asymptotically normal characterized by the mean and variance of the perturbation direction. This result is used to convert the asymptotics in $P^n$ to $Q^n$ by conducting a change of measure from $P^n$ to $Q^n$, and also contributes to the overall asymptotically normal limit of the decision that encompasses the bias term. It will also be leveraged later to prove results in the mild misspecification case.

**Lemma 6** (Log Likelihood Ratio Property in Definition 1[Duchi, 2021]). *Under Definition 2, when $\alpha = 1/2$, i.e., $Q^n = Q_{1/\sqrt{n}}^{\otimes n}$, the log-likelihood ratio between $Q^n$ and $P^n$ satisfies:*

$$\log \frac{dQ^n(\boldsymbol{z}_1, ..., \boldsymbol{z}_n)}{dP^n(\boldsymbol{z}_1, ..., \boldsymbol{z}_n)} = \frac{1}{\sqrt{n}} \sum_{i=1}^{n} u(\boldsymbol{z}_i) - \frac{1}{2} \mathbb{E}_{\boldsymbol{\theta}_0}[u^2] + o_{P^n}(1).$$

*Proof of Theorem 1.* By Lemma 2 and Proposition 1, under $P^n$, we have a joint central limit theorem

$$\begin{bmatrix} \sqrt{n}(\hat{\boldsymbol{w}}^{\square} - \boldsymbol{w}_{\boldsymbol{\theta}_0}) \\ \log \frac{dQ^n}{dP^n} \end{bmatrix} \overset{P^n}{\to} N\left( \begin{bmatrix} 0 \\ -\frac{1}{2}\mathbb{E}_{\boldsymbol{\theta}_0}(u^2) \end{bmatrix}, \begin{bmatrix} \mathrm{var}_{\boldsymbol{\theta}_0}(\mathrm{IF}^{\square}(\boldsymbol{z})) & \mathbb{E}_{\boldsymbol{\theta}_0}[u(\boldsymbol{z})\mathrm{IF}^{\square}(\boldsymbol{z})] \\ \mathbb{E}_{\boldsymbol{\theta}_0}[u(\boldsymbol{z})\mathrm{IF}^{\square}(\boldsymbol{z})^{\top}] & \mathbb{E}_{\boldsymbol{\theta}_0}(u^2) \end{bmatrix} \right).$$

Using LeCam's third lemma, we change the measure from $P^n$ to $Q^n$ and get that under $Q^n$,

$$\sqrt{n}(\hat{\boldsymbol{w}}^{\square} - \boldsymbol{w}_{\boldsymbol{\theta}_0}) \overset{Q^n}{\to} N(\mathbb{E}_{\boldsymbol{\theta}_0}[u(\boldsymbol{z})\mathrm{IF}^{\square}(\boldsymbol{z})], \mathrm{var}_{\boldsymbol{\theta}_0}(\mathrm{IF}^{\square}(\boldsymbol{z}))).$$

Next, by Lemma 4 (note that this is not a stochastic convergence but deterministic sequence convergence)

$$\sqrt{n}(\boldsymbol{w}_n^* - \boldsymbol{w}_{\boldsymbol{\theta}_0}) \to \mathbb{E}_{\boldsymbol{\theta}_0}[u(\boldsymbol{z})\mathrm{IF}^{\mathrm{SAA}}(\boldsymbol{z})].$$

In conclusion,

$$\sqrt{n}(\hat{\boldsymbol{w}}^{\square} - \boldsymbol{w}_n^*) = \sqrt{n}(\hat{\boldsymbol{w}}^{\square} - \boldsymbol{w}_{\boldsymbol{\theta}_0}) - \sqrt{n}(\boldsymbol{w}_n^* - \boldsymbol{w}_{\boldsymbol{\theta}_0}) \overset{Q^n}{\to} N(\boldsymbol{b}^{\square}, \mathrm{var}_{\boldsymbol{\theta}_0}(\mathrm{IF}^{\square}(\boldsymbol{z}))).$$

Let us now consider the regret. We use Taylor expansion of the regret with respect to $\boldsymbol{w}$ at $\boldsymbol{w}_n^*$ and note that $\nabla_{\boldsymbol{w}} v_n(\boldsymbol{w}_n^*) = 0$ for every $n$,

$$v_n(\hat{\boldsymbol{w}}^{\square}) - v_n(\boldsymbol{w}_n^*) = \frac{1}{2}(\hat{\boldsymbol{w}}^{\square} - \boldsymbol{w}_n^*)^{\top} \nabla_{\boldsymbol{w}\boldsymbol{w}} v_n(\boldsymbol{w}_n^*)(\hat{\boldsymbol{w}}^{\square} - \boldsymbol{w}_n^*) + o_{Q^n}(\left\| \hat{\boldsymbol{w}}^{\square} - \boldsymbol{w}_n^* \right\|^2),$$

$$n(v_n(\hat{\boldsymbol{w}}^{\square}) - v_n(\boldsymbol{w}_n^*)) = \frac{1}{2}\sqrt{n}(\hat{\boldsymbol{w}}^{\square} - \boldsymbol{w}_n^*)^{\top} \nabla_{\boldsymbol{w}\boldsymbol{w}} v_n(\boldsymbol{w}_n^*)\sqrt{n}(\hat{\boldsymbol{w}}^{\square} - \boldsymbol{w}_n^*) + o_{Q^n}(1).$$

By Assumption 2 that $\nabla_{\boldsymbol{w}\boldsymbol{w}} v_n(\boldsymbol{w}_n^*) \to \boldsymbol{V}$, the function $f : \Omega \to \mathbb{R}$ with $f(\cdot) := \frac{1}{2}(\cdot)^{\top}\boldsymbol{V}(\cdot)$ and function sequence $f_n : \Omega \to \mathbb{R}$ with $f_n(\cdot) := \frac{1}{2}(\cdot)^{\top}\nabla_{\boldsymbol{w}\boldsymbol{w}}v_n(\boldsymbol{w}_n^*)(\cdot)$ satisfy: for all sequence $\{\boldsymbol{w}_n\}_{n=1}^{\infty}$, if $\boldsymbol{w}_n \to \boldsymbol{w}$ for some $\boldsymbol{w} \in \Omega$, then $f_n(\boldsymbol{w}_n) \to f(\boldsymbol{w})$ since continuity is preserved under mulptiplication. Using the extended continuous mapping theorem (Theorem 1.11.1 in van der Vaart and Wellner [1996]), we have under $Q^n$,

$$n(v_n(\hat{\boldsymbol{w}}^{\square}) - v_n(\boldsymbol{w}_n^*)) \overset{Q^n}{\to} \frac{1}{2} N^{\square} \boldsymbol{V} N^{\square}.$$

$\square$

*Proof of Theorem 2.* Recall that:

$$\sqrt{n}(\hat{\boldsymbol{w}}^{\square} - \boldsymbol{w}_n^*) \overset{Q^n}{\to} N^{\square} := N(\boldsymbol{b}^{\square}, \mathrm{var}_{\boldsymbol{\theta}_0}(\mathrm{IF}^{\square}(\boldsymbol{z}))).$$

$$n(v_n(\hat{\boldsymbol{w}}^{\square}) - v_n(\boldsymbol{w}_n^*)) \overset{Q^n}{\to} \mathbb{G}^{\square} := \frac{1}{2}(N^{\square})^{\top}\boldsymbol{V} N^{\square}.$$

By denoting $\boldsymbol{b}^{\square}$ as $\mathbb{E}_{\boldsymbol{\theta}_0}(u(\boldsymbol{z})(\mathrm{IF}^{\square}(\boldsymbol{z}) - \mathrm{IF}^{\mathrm{SAA}}(\boldsymbol{z})))$, we can rewrite $\mathbb{G}^{\square}$ as

$$\mathbb{G}^{\square}$$
$$= \frac{1}{2}\left( N(0, \mathrm{var}_{\boldsymbol{\theta}_0}(\mathrm{IF}^{\square}(\boldsymbol{z}))) - \boldsymbol{b}^{\square} \right)^{\top} \boldsymbol{V} \left( N(0, \mathrm{var}_{\boldsymbol{\theta}_0}(\mathrm{IF}^{\square}(\boldsymbol{z}))) - \boldsymbol{b}^{\square} \right)$$
$$= \frac{1}{2}\left[ N(0, \mathrm{var}_{\boldsymbol{\theta}_0}(\mathrm{IF}^{\square}(\boldsymbol{z})))^{\top} \boldsymbol{V} N(0, \mathrm{var}_{\boldsymbol{\theta}_0}(\mathrm{IF}^{\square}(\boldsymbol{z}))) - 2\left( \boldsymbol{b}^{\square} \right)^{\top} \boldsymbol{V} N(0, \mathrm{var}_{\boldsymbol{\theta}_0}(\mathrm{IF}^{\square}(\boldsymbol{z}))) + \left( \boldsymbol{b}^{\square} \right)^{\top} \boldsymbol{V} \boldsymbol{b}^{\square} \right].$$

By taking the expectation, the cross term is zero. Hence,

$$\mathbb{E}\left(\mathbb{G}^{\square}\right) = \frac{1}{2}\left[\mathbb{E}\left[N(0, \text{var}_{\boldsymbol{\theta}_0}(\text{IF}^{\square}(\boldsymbol{z})))^{\top}\boldsymbol{V}N(0, \text{var}_{\boldsymbol{\theta}_0}(\text{IF}^{\square}(\boldsymbol{z})))\right] + \left(\boldsymbol{b}^{\square}\right)^{\top}\boldsymbol{V}\boldsymbol{b}^{\square}\right].$$

Since $\text{var}_{\boldsymbol{\theta}_0}(\text{IF}^{\text{ETO}}(\boldsymbol{z})) \leq \text{var}_{\boldsymbol{\theta}_0}(\text{IF}^{\text{IEO}}(\boldsymbol{z})) \leq \text{var}_{\boldsymbol{\theta}_0}(\text{IF}^{\text{SAA}}(\boldsymbol{z}))$, we know the stochastic dominance of the SAA, IEO and ETO, and their corresponding expectation.

$$N(0, \text{var}_{\boldsymbol{\theta}_0}(\text{IF}^{\text{ETO}}(\boldsymbol{z})))^{\top}\boldsymbol{V}N(0, \text{var}_{\boldsymbol{\theta}_0}(\text{IF}^{\text{ETO}}(\boldsymbol{z})))$$
$$\preceq_{\text{st}}N(0, \text{var}_{\boldsymbol{\theta}_0}(\text{IF}^{\text{IEO}}(\boldsymbol{z})))^{\top}\boldsymbol{V}N(0, \text{var}_{\boldsymbol{\theta}_0}(\text{IF}^{\text{IEO}}(\boldsymbol{z})))$$
$$\preceq_{\text{st}}N(0, \text{var}_{\boldsymbol{\theta}_0}(\text{IF}^{\text{SAA}}(\boldsymbol{z})))^{\top}\boldsymbol{V}N(0, \text{var}_{\boldsymbol{\theta}_0}(\text{IF}^{\text{SAA}}(\boldsymbol{z})))$$

and

$$\mathbb{E}\left[N(0, \text{var}_{\boldsymbol{\theta}_0}(\text{IF}^{\text{ETO}}(\boldsymbol{z})))^{\top}\boldsymbol{V}N(0, \text{var}_{\boldsymbol{\theta}_0}(\text{IF}^{\text{ETO}}(\boldsymbol{z})))\right]$$
$$\leq \mathbb{E}\left[N(0, \text{var}_{\boldsymbol{\theta}_0}(\text{IF}^{\text{IEO}}(\boldsymbol{z})))^{\top}\boldsymbol{V}N(0, \text{var}_{\boldsymbol{\theta}_0}(\text{IF}^{\text{IEO}}(\boldsymbol{z})))\right]$$
$$\leq \mathbb{E}\left[N(0, \text{var}_{\boldsymbol{\theta}_0}(\text{IF}^{\text{SAA}}(\boldsymbol{z})))^{\top}\boldsymbol{V}N(0, \text{var}_{\boldsymbol{\theta}_0}(\text{IF}^{\text{SAA}}(\boldsymbol{z})))\right].$$

From pervious analysis, we already know

$$\left(\boldsymbol{b}^{\text{ETO}}\right)^{\top}\boldsymbol{V}\boldsymbol{b}^{\text{ETO}} \geq \left(\boldsymbol{b}^{\text{IEO}}\right)^{\top}\boldsymbol{V}\boldsymbol{b}^{\text{IEO}} \geq \left(\boldsymbol{b}^{\text{SAA}}\right)^{\top}\boldsymbol{V}\boldsymbol{b}^{\text{SAA}}.$$

Therefore, $\mathbb{E}(\mathbb{G}^{\square})$ consist of two terms. For the first term, ETO is less than IEO, and IEO is less than SAA. For the second term, the direction is flipped. $\square$

Proposition 1 was essentially established by Elmachtoub et al. [2023], but here, we express the asymptotic behaviors of solutions more explicitly in terms of influence functions. Moreover, these more explicit expressions arise from a new projection interpretation of influence functions that allows us to describe the performances geometrically, providing another perspective different from Elmachtoub et al. [2023].

To this end, let $\boldsymbol{P}$ be the projection matrix onto the column span of $\boldsymbol{\Sigma}$ with respect to the norm $\|\boldsymbol{x}\|_{\boldsymbol{V}^{-1}}$, i.e.,

$$\boldsymbol{P}\boldsymbol{x} = \underset{\boldsymbol{y}:\boldsymbol{y}\in\text{col}(\boldsymbol{\Sigma})}{\text{argmin}}\|\boldsymbol{y} - \boldsymbol{x}\|_{\boldsymbol{V}^{-1}}^2,$$

which has a closed-form expression $\boldsymbol{P} = \boldsymbol{\Sigma}\left(\boldsymbol{\Sigma}^{\top}\boldsymbol{V}^{-1}\boldsymbol{\Sigma}\right)^{-1}\boldsymbol{\Sigma}^{\top}\boldsymbol{V}^{-1}$. Second, define the functional $\mathcal{T} : L_2(P_{\boldsymbol{\theta}_0})^{d_w} \to L_2(P_{\boldsymbol{\theta}_0})^{d_w}$ as the projection operator onto the linear function subspace $\{\boldsymbol{A}\boldsymbol{s}_{\boldsymbol{\theta}_0}(\boldsymbol{z}) : \boldsymbol{A} \in \mathbb{R}^{d_w \times d_\theta}\}$, i.e., for a square integrable function $\boldsymbol{f}(\boldsymbol{z}) : \mathcal{Z} \to \mathbb{R}^{d_w}$,

$$\mathcal{T}\boldsymbol{f} = \underset{\boldsymbol{g}:\boldsymbol{g}=\boldsymbol{A}\boldsymbol{s}_{\boldsymbol{\theta}_0}(\boldsymbol{z})}{\text{argmin}}\int \|\boldsymbol{f}(\boldsymbol{z}) - \boldsymbol{g}(\boldsymbol{z})\|^2 p_{\boldsymbol{\theta}_0}(\boldsymbol{z})d\boldsymbol{z}.$$

**Theorem 7.** *Under Assumptions 1, 3 and 4, the influence functions of* IEO *and* ETO *have the following projection interpretation.*

1. $\text{IF}^{\text{IEO}}(\boldsymbol{z}) = -\boldsymbol{V}^{-1}\boldsymbol{P}\nabla_{\boldsymbol{w}}c(\boldsymbol{w}_{\boldsymbol{\theta}_0}, \boldsymbol{z})$,

2. $\text{IF}^{\text{ETO}}(\boldsymbol{z}) = -\boldsymbol{V}^{-1}\mathcal{T}\nabla_{\boldsymbol{w}}c(\boldsymbol{w}_{\boldsymbol{\theta}_0}, \boldsymbol{z})$.

The above theorem points out that the influence functions of IEO and ETO are essentially projections of that of SAA, either in vector or function spaces, shedding light on the ordering of their variances by the contraction properties of projections.

*Proof of Theorem 7.* The fact that
$$\text{IF}^{\text{IEO}}(\boldsymbol{z}) = -\boldsymbol{V}^{-1}\boldsymbol{P}\nabla_{\boldsymbol{w}}c(\boldsymbol{w}_{\boldsymbol{\theta}_0}, \boldsymbol{z})$$
is because
$$\text{IF}^{\text{IEO}}(\boldsymbol{z}) = \boldsymbol{V}^{-1}\boldsymbol{\Sigma}\boldsymbol{\Phi}^{-1}\nabla_{\boldsymbol{\theta}}c(\boldsymbol{w}_{\boldsymbol{\theta}_0}, \boldsymbol{z})$$
$$= \boldsymbol{V}^{-1}\boldsymbol{\Sigma}(\nabla_{\boldsymbol{\theta}}\boldsymbol{w}_{\boldsymbol{\theta}_0}\boldsymbol{V}\nabla_{\boldsymbol{\theta}}\boldsymbol{w}_{\boldsymbol{\theta}_0}^{\top})^{-1}\nabla_{\boldsymbol{\theta}}\boldsymbol{w}_{\boldsymbol{\theta}_0}\nabla_{\boldsymbol{w}}c(\boldsymbol{w}_{\boldsymbol{\theta}_0}, \boldsymbol{z})$$
$$= \boldsymbol{V}^{-1}\boldsymbol{\Sigma}(\boldsymbol{\Sigma}^{\top}\boldsymbol{V}^{-1}\boldsymbol{V}\boldsymbol{V}^{-1}\boldsymbol{\Sigma})^{-1}\left(-\boldsymbol{\Sigma}^{\top}\boldsymbol{V}^{-1}\right)\nabla_{\boldsymbol{w}}c(\boldsymbol{w}_{\boldsymbol{\theta}_0}, \boldsymbol{z})$$
$$= -\boldsymbol{V}^{-1}\boldsymbol{P}_{\boldsymbol{\Sigma},\boldsymbol{V}}\nabla_{\boldsymbol{w}}c(\boldsymbol{w}_{\boldsymbol{\theta}_0}, \boldsymbol{z}).$$

We then show the relationship between $\mathrm{IF}^{\mathrm{ETO}}(z)$ and $\mathrm{IF}^{\mathrm{SAA}}(z)$. Let $\mathcal{T} : L_2(p_{\boldsymbol{\theta}_0})^{d_w} \to L_2(p_{\boldsymbol{\theta}_0})^{d_\theta}$ be the projection matrix on the linear function subspace $\{\boldsymbol{A}s_{\boldsymbol{\theta}_0}(z) : \boldsymbol{A} \in \mathbb{R}^{d_w \times d_\theta}\}$, i.e., for general function $\boldsymbol{f}(z) : \mathcal{Z} \to \mathbb{R}^{d_\theta}$,

$$\mathcal{T}\boldsymbol{f} = \operatorname*{argmin}_{\boldsymbol{g}:\boldsymbol{g}=\boldsymbol{A}s_{\boldsymbol{\theta}_0}(z)} \int \|\boldsymbol{f}(z) - \boldsymbol{g}(z)\|^2 \, p_{\boldsymbol{\theta}_0}(z)dz.$$

The influence function of ETO is also a projection, i.e.,

$$\mathrm{IF}^{\mathrm{ETO}}(z) = \boldsymbol{V}^{-1}\mathcal{T}\nabla_{\boldsymbol{w}}c(\boldsymbol{w}_{\boldsymbol{\theta}_0}, z).$$

The reason is as follows. For ETO, it suffices to prove the following fact:

$$\mathcal{T}\boldsymbol{f} = \mathbb{E}_{\boldsymbol{\theta}_0}(\boldsymbol{f}s_{\boldsymbol{\theta}_0}^\top)\boldsymbol{I}^{-1}s_{\boldsymbol{\theta}_0}(z)$$

since $\boldsymbol{\Sigma} = \mathbb{E}[\nabla_{\boldsymbol{w}}c(\boldsymbol{w}_{\boldsymbol{\theta}_0}, z)s_{\boldsymbol{\theta}_0}(z)]$. To prove the fact, we need to show that $\boldsymbol{A}^* = \mathbb{E}_{\boldsymbol{\theta}_0}(\boldsymbol{f}(z)s_{\boldsymbol{\theta}_0}(z)^\top)\boldsymbol{I}^{-1}$ is the minimizer of the optimization problem

$$\min_{\boldsymbol{A}\in\mathbb{R}^{d_w \times d_\theta}} \int \|\boldsymbol{f}(z) - \boldsymbol{A}s_{\boldsymbol{\theta}_0}(z)\|^2 \, p_{\boldsymbol{\theta}_0}(z)dz.$$

Since this is essentially a quadratic optimization problem, the stationary point is the global minimum. Denote the objective function $h(\boldsymbol{A})$ and we require $\nabla_{\boldsymbol{A}}h(\boldsymbol{A}^*) = 0$. In other words, for all $\tilde{i}, \tilde{j}$, $\partial h(\boldsymbol{A})/\partial A_{\tilde{i},\tilde{j}} = 0$. For simplicity, we write $p_{\boldsymbol{\theta}_0}(z)$ as $p(z)$ and $s_{\boldsymbol{\theta}_0}(z)$ as $s(z)$. Note that

$$h(\boldsymbol{A}) = \int_{\boldsymbol{z}\in\mathcal{Z}} \sum_{i=1}^{d_w} (f_i(z) - \sum_{j=1}^{d_\theta} A_{ij}s_j(z))^2 p(z)d\boldsymbol{z}$$

$$= \sum_{i=1}^{d_w} \int_{\boldsymbol{z}\in\mathcal{Z}} \left[ f_i(z)^2 + (\sum_{j=1}^{d_\theta} A_{ij}s_j(z))^2 - 2f_i(z)\sum_{j=1}^{d_\theta} A_{ij}s_j(z) \right] p(z)d\boldsymbol{z}$$

We have

$$\partial h(\boldsymbol{A}^*)/\partial A_{\tilde{i},\tilde{j}} = \int_{\boldsymbol{z}\in\mathcal{Z}} \left[ -2f_{\tilde{i}}(z)s_{\tilde{j}}(z) + \left[ 2\sum_{j=1}^{d_\theta} A_{\tilde{i}j}s_j(z) \right] \right] p(z)d\boldsymbol{z} = 0.$$

For all $\tilde{i}, \tilde{j}$, we have

$$\int_{\boldsymbol{z}\in\mathcal{Z}} f_{\tilde{i}}(z)s_{\tilde{j}}(z)p(z)dz = \int_{\boldsymbol{z}\in\mathcal{Z}} \sum_{j=1}^{d_\theta} A_{\tilde{i}j}s_j(z)s_{\tilde{j}}(z)p(z)dz.$$

Writing in a matrix form, the left hand side is $\mathbb{E}_{\boldsymbol{\theta}_0}(\boldsymbol{f}(z)s(z)^\top)$. The write hand side is $\mathbb{E}_{\boldsymbol{\theta}_0}[\boldsymbol{A}^*s(z)s(z)^\top] = \boldsymbol{A}^*\mathbb{E}_{\boldsymbol{\theta}_0}(s(z)s(z)^\top) = \boldsymbol{A}^*\boldsymbol{I}$. In conclusion, $\boldsymbol{A}^* = \mathbb{E}(\boldsymbol{f}(z)s(z)^\top)\boldsymbol{I}^{-1}$ and $\mathcal{T}\boldsymbol{f} = \mathbb{E}(\boldsymbol{f}(z)s_{\boldsymbol{\theta}_0}(z)^\top)\boldsymbol{I}^{-1}s_{\boldsymbol{\theta}_0}(z)$. □

*Proof of Lemma 1.* The first identity follows from

$$\boldsymbol{\Sigma}^\top = \nabla_{\boldsymbol{\theta}}\nabla_{\boldsymbol{w}}v(\boldsymbol{w}, \boldsymbol{\theta})|_{\boldsymbol{w}=\boldsymbol{w}_{\boldsymbol{\theta}_0}, \boldsymbol{\theta}=\boldsymbol{\theta}_0}$$

$$= \nabla_{\boldsymbol{\theta}}\mathbb{E}_{\boldsymbol{\theta}}[\nabla_{\boldsymbol{w}}c(\boldsymbol{w}, z)]|_{\boldsymbol{\theta}=\boldsymbol{\theta}_0, \boldsymbol{w}=\boldsymbol{w}_{\boldsymbol{\theta}_0}}$$

$$= \nabla_{\boldsymbol{\theta}} \int \nabla_{\boldsymbol{w}}c(\boldsymbol{w}_{\boldsymbol{\theta}_0}, z)p_{\boldsymbol{\theta}_0}(z)dz$$

$$= \int \nabla_{\boldsymbol{\theta}}p_{\boldsymbol{\theta}_0}(z)\nabla_{\boldsymbol{w}}c(\boldsymbol{w}_{\boldsymbol{\theta}_0}, z)^\top dz$$

$$= \int (\nabla_{\boldsymbol{\theta}}\log p_{\boldsymbol{\theta}_0}(z))\, p_{\boldsymbol{\theta}_0}(z)\, (\nabla_{\boldsymbol{w}}c(\boldsymbol{w}_{\boldsymbol{\theta}_0}, z))^\top dz$$

$$= \mathbb{E}_{\boldsymbol{\theta}_0}[s_{\boldsymbol{\theta}_0}(z)\, (\nabla_{\boldsymbol{w}}c(\boldsymbol{w}_{\boldsymbol{\theta}_0}, z))^\top].$$

For the second identity, by implicit function theorem and applying Barratt [2018], we can prove the first identity

$$0 = \nabla_{\boldsymbol{ww}} v(\boldsymbol{w}, \boldsymbol{\theta}_0)|_{\boldsymbol{w}=\boldsymbol{w}_{\boldsymbol{\theta}_0}} \left(\nabla_{\boldsymbol{\theta}} \boldsymbol{w}_{\boldsymbol{\theta}}|_{\boldsymbol{\theta}=\boldsymbol{\theta}_0}\right)^\top + \nabla_{\boldsymbol{w}} \nabla_{\boldsymbol{\theta}} v(\boldsymbol{w}, \boldsymbol{\theta})|_{\boldsymbol{w}=\boldsymbol{w}_{\boldsymbol{\theta}_0}, \boldsymbol{\theta}=\boldsymbol{\theta}_0},$$

$$\Rightarrow \nabla_{\boldsymbol{\theta}} \boldsymbol{w}_{\boldsymbol{\theta}}|_{\boldsymbol{\theta}=\boldsymbol{\theta}_0} = -\nabla_{\boldsymbol{\theta}} \nabla_{\boldsymbol{w}} v(\boldsymbol{w}, \boldsymbol{\theta})|_{\boldsymbol{w}=\boldsymbol{w}_{\boldsymbol{\theta}_0}, \boldsymbol{\theta}=\boldsymbol{\theta}_0} \cdot \nabla_{\boldsymbol{ww}} v(\boldsymbol{w}, \boldsymbol{\theta}_0)^{-1}|_{\boldsymbol{w}=\boldsymbol{w}_{\boldsymbol{\theta}_0}},$$

$$= -\boldsymbol{\Sigma}^\top \boldsymbol{V}^{-1}.$$

The third identity follows since

$$\begin{aligned}
\boldsymbol{\Phi} &= \nabla_{\boldsymbol{\theta\theta}} \mathbb{E}_{\boldsymbol{\theta}_0}\left[c(\boldsymbol{w}_{\boldsymbol{\theta}_0}, \boldsymbol{z})\right] \\
&= \nabla_{\boldsymbol{\theta}} \left(\nabla_{\boldsymbol{w}} \mathbb{E}_{\boldsymbol{\theta}_0}\left[c(\boldsymbol{w}_{\boldsymbol{\theta}}, \boldsymbol{z})\right] \nabla_{\boldsymbol{\theta}} \boldsymbol{w}_{\boldsymbol{\theta}}^\top\right)|_{\boldsymbol{\theta}=\boldsymbol{\theta}_0} \\
&= \nabla_{\boldsymbol{\theta}} \boldsymbol{w}_{\boldsymbol{\theta}} \nabla_{\boldsymbol{ww}} \mathbb{E}_{\boldsymbol{\theta}_0}\left[c(\boldsymbol{w}_{\boldsymbol{\theta}_0}, \boldsymbol{z})\right] \nabla_{\boldsymbol{\theta}} \boldsymbol{w}_{\boldsymbol{\theta}}^\top \\
&= \left(-\boldsymbol{\Sigma}^\top \boldsymbol{V}^{-1}\right) \boldsymbol{V} \left(-\boldsymbol{\Sigma}^\top \boldsymbol{V}^{-1}\right)^\top \\
&= \boldsymbol{\Sigma}^\top \boldsymbol{V}^{-1} \boldsymbol{\Sigma}
\end{aligned}$$

by noting that $\nabla_{\boldsymbol{w}} \mathbb{E}_{\boldsymbol{\theta}_0}\left[c(\boldsymbol{w}_{\boldsymbol{\theta}_0}, \boldsymbol{z})\right] = \boldsymbol{0}$ since $\boldsymbol{w}_{\boldsymbol{\theta}_0}$ is the minimizer of the function $\boldsymbol{w} \to \mathbb{E}_{\boldsymbol{\theta}_0}\left[c(\boldsymbol{w}, \boldsymbol{z})\right]$.
□

*Proof of Lemma 2.*

$$\log \frac{dQ^n(\boldsymbol{z}_1, ..., \boldsymbol{z}_n)}{dP^n(\boldsymbol{z}_1, ..., \boldsymbol{z}_n)} = \log \prod_{i=1}^n \frac{\exp(u(\boldsymbol{z}_i)/\sqrt{n})}{C_{1/\sqrt{n}}} = \frac{1}{\sqrt{n}} \sum_{i=1}^n u(\boldsymbol{z}_i) - n \log C_{1/\sqrt{n}}.$$

It now suffices to show that $n \log C_{1/\sqrt{n}} = \frac{1}{2} \mathbb{E}_{\boldsymbol{\theta}_0}[u^2] + o_{P^n}(1)$. From the definition of $C_t$, we know

$$C_t = \int \exp(tu(\boldsymbol{z})) dP_{\boldsymbol{\theta}_0}(\boldsymbol{z}).$$

Taking the derivative, we have

$$(C_t)'|_{t=0} = \int \exp(tu(\boldsymbol{z})) u(\boldsymbol{z}) dP_{\boldsymbol{\theta}_0}(\boldsymbol{z})|_{t=0} = \mathbb{E}_{\boldsymbol{\theta}_0}(u(\boldsymbol{z})) = 0.$$

Taking the second order derivative, we have

$$(C_t)''|_{t=0} = \int \exp(tu(\boldsymbol{z})) u(\boldsymbol{z}) u(\boldsymbol{z}) dP_{\boldsymbol{\theta}_0}(\boldsymbol{z})|_{t=0} = \mathbb{E}_{\boldsymbol{\theta}_0}(u^2).$$

By Talor expansion, we have

$$C_t = 1 + \frac{1}{2} \mathbb{E}_{\boldsymbol{\theta}_0}[u^2] t^2 + o(t^2)$$

In conclusion,

$$\begin{aligned}
n \log C_{1/\sqrt{n}} = n \log\left(1 + \frac{1}{2} \frac{1}{\sqrt{n}} \mathbb{E}_{\boldsymbol{\theta}_0}[u^2] \frac{1}{\sqrt{n}} + o\left(\frac{1}{n}\right)\right) &= n\left(\frac{1}{2} \frac{1}{\sqrt{n}} \mathbb{E}_{\boldsymbol{\theta}_0}[u^2] \frac{1}{\sqrt{n}} + o\left(\frac{1}{n}\right)\right) \\
&= \frac{1}{2} \mathbb{E}_{\boldsymbol{\theta}_0}[u^2] + o(1).
\end{aligned}$$

□

