# OpenReview forum: "The Bias-Variance Tradeoff in Data-Driven Optimization: A Local Misspecification Perspective"
_NeurIPS.cc/2025/Conference — NeurIPS 2025 poster_

### Official Review · Reviewer_tsVS · 2025-06-10

**Clarity:** 3
**Significance:** 2
**Originality:** 3
**Rating:** 4
**Confidence:** 2

**Summary:**

This paper presents a theoretical investigation into the relative performance of three widely used approaches in data-driven stochastic optimization: Sample Average Approximation (SAA), Estimate-Then-Optimize (ETO), and Integrated Estimation-Optimization (IEO). The authors rigorously analyze the bias-variance tradeoff associated with these methods under conditions of local model misspecification, a realistic setting in which the true data-generating process deviates slightly from the assumed model.

By employing tools from contiguity theory in statistics, the paper distinguishes itself with a mathematically elegant framework. The analysis reveals that the relative effectiveness of SAA, ETO, and IEO is context-dependent. Specifically, when the degree of local misspecification increases, the tradeoff between variance (due to data noise) and bias (introduced by model misspecification) becomes a critical factor. This highlights that no single method dominates across all regimes; rather, the optimal choice depends on how well the model aligns with reality.

While the theoretical result is interesting, it is not clear how the method can be used and applied in practice because there is no experiment, even on synthetic data. I gave the Borderline accept with less confidence.

**Questions:**

This paper offers interesting theoretical insights that could inspire future empirical or methodological work, but it lacks the experimental evidence necessary to demonstrate relevance or applicability in practice. Adding even a basic synthetic experiment or discussing potential practical guidelines would substantially improve the paper's value to the community.

**Ethical Concerns:**

["NO or VERY MINOR ethics concerns only"]

**Limitations:**

See the weaknesses

**Quality:**

3

**Strengths And Weaknesses:**

Strengths:
* Novelty and Theoretical Depth: The application of contiguity theory to analyze local misspecification and derive asymptotic comparisons is intellectually rigorous and adds depth to the existing literature on decision-making under uncertainty.

* Clear Exposition: The paper is well-structured and the theoretical arguments are laid out in a clear, readable fashion. The definitions and assumptions are reasonable and standard in the field, and the logic flows smoothly.

* Conceptual Insight: The identification of distinct sources of bias and variance—stemming respectively from model mismatch and sample randomness—sheds light on why optimization performance varies across methods and provides a fresh lens to view classical problems.

Weaknesses:
* Lack of Empirical Evaluation: A major shortcoming of the paper is the absence of any empirical validation. While the theoretical results are compelling, it remains unclear how significant the observed effects are in practical settings. Even a simple synthetic example would help to visualize the tradeoffs and enhance the paper’s impact.

* Practical Applicability is Unclear: In real-world scenarios, practitioners often do not know whether or how their models are misspecified. Estimating the degree of local misspecification or the noise level in the data is non-trivial, which complicates the practical utility of the results. The paper stops short of offering actionable guidelines on how a practitioner might choose between SAA, ETO, or IEO based on observable properties of their problem.

* No Discussion on Robustness or Extensions: The paper could be strengthened by discussing how robust the results are to broader forms of misspecification or whether the framework can be extended to incorporate distributional robustness, Bayesian perspectives, or online decision-making settings.

---

> ### Author Rebuttal · Authors · 2025-07-30
>
> We sincerely appreciate your detailed review and thoughtful comments on our paper.
>
> W1: We completely understand your concern, and have run and discussed new experimental results. Please refer to our response to Reviewer JBev for details.
>
> W2: We very much understand the reviewer's point on practical applicability. As in many other theoretical studies, the goal of our paper is to provide insights, in our case the bias-variance dissection and directional impacts of model misspecification, and also to bring in the new theoretical tools that allow us to derive the results and conclude these insights. There is certainly an important question on practical implementation. To this end, in our mind our work is not a one-time study in this problem domain, instead there will be many follow-up works on other aspects including practical and computational issues.
>
> Thus, we are careful not to over-state our contributions (both in the paper and in this response). That being said, our theory does give some clear insights. First, in terms of choosing among the three methods: if the misspecification error is larger than the sampling error, then SAA should be used; if the model is almost well-specified, then ETO should be used; if the misspecification error and the sampling error are of the same scale, or are generally unknown, then IEO is usually a robust choice. Second, our main results reveal that the orderings of the bias and variance of the three methods are universal regardless of (almost) all directions $u(\boldsymbol{z})$, but only depend on the degree of misspecification. That is, the deviating directions $u$ of $Q$ turn out to be an invariant factor under most scenarios.
>
> W3: We will add discussions on the extensions of our framework to other settings in the paper. These extensions, including contextual optimization, constrained optimization and the incorporation of robustness, range from immediately doable to the need of a potentially substantial expansion of our framework: For contextual optimization, the extension is quite immediate. Specifically, we introduce the feature $\boldsymbol{x}$ in additional to the random parameter $\boldsymbol{z}$. We write the joint distribution of $\boldsymbol{x}$ and $\boldsymbol{z}$ as $Q(\boldsymbol{x},\boldsymbol{z})=Q(\boldsymbol{x})Q(\boldsymbol{z}|\boldsymbol{x})$. In this case we need to estimate $Q(\boldsymbol{z}|\boldsymbol{x})$ using parametric family $Q_{\boldsymbol{\theta}}(\boldsymbol{z}|\boldsymbol{x})$. Note that in the contextual case, the SAA method is not interesting because it  allows to choose any map from context to decision, thus easily overfitting the
> finite-sample problem. In this case, IEO and ETO are the more legitimate approaches. In the mildly misspecified case, ETO is better than IEO while in the balanced and misspecified case, IEO is better than ETO.
> For constrained optimization, if the constraints are deterministic, then we expect our analysis and insights to continue to hold, but it would introduce additional technical complexity from the KKT conditions instead of using simpler optimality conditions. On the other hand, if the constraints are stochastic, then there is an additional question of feasibility (e.g., in chance constrained settings where random safety conditions need to be enforced with high probability). These extensions to constrained settings warrant future work in our view. Lastly, to incorporate robustness, a natural consideration is the possibility of distributional shifts. In this setup, we would need to consider the worst-case regret or other decision-theoretic counterparts, and the involved dissection is not only confined to statistical bias and variance but also deterioration of the decision quality coming from the fundamental shifts. This extension is substantial and would be worth another future work.
>
> Q1: As described in our response to W1, we completely understand your concern, and refer you to our new experimental results and discussions in our response to Reviewer JBev.

---

### Official Review · Reviewer_JBev · 2025-07-03

**Clarity:** 3
**Significance:** 3
**Originality:** 3
**Rating:** 4
**Confidence:** 2

**Summary:**

Prior work focused on either well-specified models or fully misspecified models, no theoretical framework to analyze the intermediate and realistic case where the model is locally misspecified. Besides, SAA, ETO, and IEO are widely used, but their performance differences under slight misspecification were not well understood.

The key challenge is to develop a unified theoretical framework for analyzing and comparing data-driven optimization methods under local model misspecification, and proved insights into when each method should be applied.

This paper studies the statistical performance of SAA, ETO, and IEO—under local model misspecification. The authors analyze how bias and variance trade off across different misspecification regimes (mild, balanced, severe). Key contributions include: a unified asymptotic framework capturing these regimes, explicit expressions for decision bias, and sufficient conditions under which misspecification has negligible impact. The results provide guidance on method selection depending on the degree and direction of model misspecification.

**Questions:**

See my questions in the above “Weakness” part and below:

- Could you provide a discussion on how you expect these asymptotic results to manifest in finite-sample settings?
- It would be great if the author may consider adding a small-scale numerical experiment, even on a simple, synthetic problem.

**Ethical Concerns:**

["NO or VERY MINOR ethics concerns only"]

**Final Justification:**

My main concern was satisfactorily addressed. I remain my overall assessment.

**Limitations:**

See the limitations also in the above “Weakness” and “Questions” part.

**Paper Formatting Concerns:**

The formatting of Table 1 could be improved.

**Quality:**

3

**Strengths And Weaknesses:**

The analysis relies on classical smoothness and identifiability assumptions (e.g., differentiability of cost functions, uniqueness of minimizers, invertibility of Hessians, etc.), which seem natural and reasonable to me.

**Strengths:**

- This paper presents a technically solid analysis of data-driven stochastic optimization under local model misspecification.
- It provides asymptotic results for SAA, ETO, IEO under varying degree of model misspecification.
- They derive explicit expressions for decision bias and variance, offering insights into when and why model-based methods might fail or succeed.
- The main contribution is a local misspecification framework, which allows for a better understanding of how these methods perform when the underlying statistical model is "almost" correct.
- The paper appears novel both in problem framing and in technical methodology.


**Weakness:**

- Lack of empirical finding of finite sample behavior: The paper’s main results (Theorem 1, 3, and 6) describe the limiting behavior of the estimators as the $n \rightarrow \infty$. In practice, devisions are always made with a finite amount of data. This paper lack of evidence to show how well this asymptotic theory approximates the actual performance of SAA, ETO, and IEO for realistic finite sample sizes.
- It does not provide insight into the absolute magnitudes of bias vs variance. For a given problem, is the bias from a "mild" misspecification truly negligible compared to the variance? In the "balanced" regime, is the bias-variance tradeoff a dramatic one, or are the differences between the methods marginal?
- Theorem 5: misspecification in the direction of the model’s score function leads to zero asymptotic bias. This is a powerful results, but again its practical utility is not tested, the lack of empirical validation makes the work feel incomplete. It provides a map of the asymptotic world but offers no guidance on how that map corresponds to the finite-sample territories where practitioners actually operate.

---

> ### Author Rebuttal · Authors · 2025-07-31
>
> We sincerely appreciate your detailed review and thoughtful comments on our paper.
>
> W1: We completely understand and have added experiments to validate our theoretical results. We validate our findings by conducting numerical experiments on the newsvendor problem, a classic example in operations research with non-linear cost objectives. We show and compare the performances of the three data-driven methods under different local misspecification settings, including different directions and degrees of misspecification, to validate our theoretical analysis as well as providing additional insights about the effect of model parameters.
>
> The newsvendor problem has the objective function $c(w,z)={a}^\top(w-z)^++d^\top(z-w)^+$, where for each $j\in[d_z]$: (1) $z^{(j)}$ is the customers' random demand of product $j$; (2) $w^{(j)}$ is the decision variable, the ordering quantity for product $j$; (3) $a^{(j)}$  is the holding cost for product $j$; (4) $d^{(j)}$ is the backlogging cost for product $j$. We assume the random demand for each product are independent. We assume the holding cost and backlogging cost is uniform among all products by setting $a^{(j)}=5$ and $d^{(j)}=1$ for all $j\in[d_z]$.
>
> We describe the local misspecified setting by using the framework of Example 5 and building a model and generating a random demand dataset as follows. We denote the training dataset as $\{z_i^{(j)}\}_{i=1}^n$, where $n$ is the training sample size. The model assumes that the demand for each product $j\in[d_z]$ is normally distributed with the distribution $N(j\theta,1)$ where $\theta$ is unknown and needs to be learned. We first describe the well-specified setting, where the demand distribution for product $j$ is $N(3j,1)$. To describe the local misspecification, we need to specify the direction and degree of misspecification. We set (1) $\alpha=0.1$ for the severely misspecified setting, (2) $\alpha=0.5$ for the balanced setting and $\alpha=2$ for the mildly misspecified setting.
>
> We discuss two types of directions: (1) $u(z)=\sum_{j=1}^{d_z} (z^{(j)})^2$; (2)  ${u}(z)=\sum_{j=1}^{d_z} j(z^{(j)}-3j)$.
>
> We show experimental results in Tables 1-3 to support our theoretical results in Section 3, using the mean of the regret for three methods, under three directions and three misspecified regimes. When ${u}(z)=\sum_{j=1}^{d_z} (z^{(j)})^2$, in the mildly specified case, ETO has a lower regret than IEO, and IEO has a lower regret than SAA. However, in the severely misspecified regime, the ordering of the three methods flips. This is consistent with our theoretical comparison results in Theorems 3, 4 and 6. In the balanced regime, experimental results show that IEO has the lowest regret among the three methods. This is also consistent with the theoretical insight in Theorem 2 that IEO has the advantage of more readily achieving the bias-variance trade-off in terms of the decisions and corresponding regrets.  It also gives us some practical guidelines in choosing among the three methods: if the misspecification error is larger than the sampling error, then SAA should be used; if the model is almost well-specified, then ETO should be used; if the misspecification error and the sampling error are of the same scale, or are generally unknown, then IEO is usually a robust choice.
>
> W2: We note that (and we believe this is precisely what the reviewer thinks as well), as in other asymptotic analyses, our "negligibility" is theoretically framed by comparing the orders of different sequences. Consequently, the theory by itself does not reveal the absolute magnitudes of the various quantities; instead it serves primarily to provide comparative insights. While this is a typical limitation of any asymptotic study, through our new experiments we can get a sense of the absolute magnitudes, as shown in Tables 5-7, when $u(z)=\sum_{j=1}^{d_z} (z^{(j)})^2$ under three different misspecified regimes. We demonstrate the bias and variance terms as scalars by taking the norm of sample mean and sample covariance matrix of $n^{-\min\{1/2,\alpha\}}(\hat{w}^\square-w^*_n)$.  In the mildly misspecified case, the bias term is dominated by the variance term; in the severely misspecified regime, the dominance is flipped; in the balanced case, no significant dominance exists. Moreover, the results are consistent with the bias and variance comparison results in Theorems 1, 3, 4, 6.
>
> W3: In our new experiments, we include the setting when ${u}(z)=\sum_{j=1}^{d_z} j(z^{(j)}-3j)$, an impactless direction discussed in Theorem 5. In this case, in Table 4 the ordering of the three methods is different from previous scenarios (Table 1-3). This is because the direction of misspecification ${u}(z)$ in this case coincides with the score function itself. Note that the score function at $\theta\_0=3$ is $s_{\theta_0}(z)=\nabla\_{\theta}\log p\_{\theta\_0}(z)=\nabla\_{\theta}\log\exp(-\sum\_{j=1}^{d_z} (z^{(j)}-3j)^2/2)=\sum\_{j=1}^{d\_z} j(z^{(j)}-3j)$. Even in the severely misspecified setting, all three methods have asymptotically $o(n^{-\alpha})$ decision bias and corresponding $o(n^{-\alpha})$ regret, i.e., $b^\textup{SAA}=b^{\textup{IEO}}=b^\textup{ETO}=0$ and $R^\textup{SAA}=R^\textup{IEO}=R^\textup{ETO}=0$. Intuitively, through this particular direction, regardless of the degree of misspecification, the ordering of the performance of the three methods is consistent with the mildly misspecified setting, where ETO is better than IEO and IEO is better than SAA. The experiments validate our theory and show our theory is robust. Not only the theory holds in the asymptotic regime but also we have seen the same finite-sample behaviours and trends for a wide range of sample sizes.
>
> Q1 and Q2: As described in our responses to your comments above, we have provided and discussed new experimental results that aim to give a sense of the finite-sample manifestation.
>
> We also want to point out that, in terms of building additional theory that sheds light on finite-sample behaviors, we can likely derive corresponding bounds, but it is unclear that this would reveal better insights than our asymptotic analysis. Please see our response to W2 for Reviewer JYdJ regarding the comparison and motivation of our work with [1] that focuses on finite-sample bounds.
>
> [1] Elmachtoub, Adam N., Henry Lam, Haixiang Lan, and Haofeng Zhang. "Dissecting the Impact of Model Misspecification in Data-driven Optimization." arXiv preprint arXiv:2503.00626 (2025).
>
> Note: the caption stands for the table above.
>
> |n|SAA|IEO|ETO|
> |--|---|---|---|
> |20|7.92e-2|8.74e-3|3.85e-3|
> |40|2.75e-2|4.11e-3|2.13e-3|
> |60|2.53e-2|2.09e-3|1.75e-3|
> |80|1.80e-2|1.74e-3|1.25e-3|
> |100|1.62e-2|2.96e-3|1.49e-3|
> |120|1.45e-2|9.49e-4|4.05e-4|
> |140|9.45e-3|9.17e-4|3.44e-4|
> |160|1.25e-2|1.68e-3|6.49e-4|
> |180|8.50e-3|1.69e-3|1.91e-4|
> |200|8.66e-3|8.12e-4|3.71e-4|
>
> Table1: mild case, first direction
>
> |n|SAA|IEO|ETO|
> |--|---|---|---|
> |20|4.85e-2|1.37e-2|3.30e-2|
> |40|3.57e-2|7.36e-3|1.29e-2|
> |60|2.54e-2|5.51e-3|8.12e-3|
> |80|1.90e-2|3.14e-3|5.84e-3|
> |100|1.22e-2|2.21e-3|6.23e-3|
> |120|1.53e-2|1.63e-3|3.86e-3|
> |140|9.51e-3|1.82e-3|5.26e-3|
> |160|9.45e-3|2.47e-3|3.25e-3|
> |180|6.44e-3|1.70e-3|4.25e-3|
> |200|4.64e-3|1.14e-3|2.66e-3|
>
> Table 2: balanced case, first direction
>
> |n|SAA|IEO|ETO|
> |--|---|---|---|
> |20|4.64e-2|2.76e-2|1.18e-1|
> |40|2.19e-2|2.35e-2|1.03e-1|
> |60|1.49e-2|2.60e-2|1.05e-1|
> |80|8.80e-3|2.16e-2|9.99e-2|
> |100|1.05e-2|2.04e-2|9.86e-2|
> |120|9.34e-3|2.03e-2|1.01e-1|
> |140|5.80e-3|2.02e-2|9.69e-2|
> |160|5.51e-3|2.03e-2|9.50e-2|
> |180|5.46e-3|2.00e-2|9.23e-2|
> |200|5.35e-3|2.02e-2|8.91e-2|
>
> Table 3: severe case, first direction
>
> |n|SAA|IEO|ETO|
> |--|---|---|---|
> |20|7.39e-2|4.94e-3|6.25e-3|
> |40|4.16e-2|8.01e-3|5.45e-3|
> |60|2.76e-2|2.86e-3|1.12e-3|
> |80|1.93e-2|2.58e-3|7.16e-4|
> |100|1.38e-2|1.96e-3|7.24e-4|
> |120|1.64e-2|1.59e-3|1.26e-3|
> |140|1.09e-2|4.95e-4|4.77e-4|
> |160|7.81e-3|1.36e-3|1.02e-3|
> |180|9.29e-3|1.23e-3|5.04e-4|
> |200|6.89e-3|1.09e-3|2.12e-4|
>
> Table 4: severe case, second direction
>
> |n|SAA_b|SAA_v|ETO_b|ETO_v|IEO_b|IEO_v|
> |--|-----|-----|-----|-----|-----|-----|
> |20|4.41e-1|2.63e+0|4.86e-2|4.79e-1|3.44e-2|1.41e+0|
> |40|1.26e-1|2.71e+0|1.84e-1|8.30e-1|1.60e-1|1.48e+0|
> |60|1.50e-1|4.13e+0|5.95e-2|6.88e-1|3.13e-1|1.90e+0|
> |80|1.11e-1|3.30e+0|4.56e-1|1.17e+0|1.99e-1|2.18e+0|
> |100|4.94e-1|3.83e+0|5.07e-2|9.36e-1|3.53e-1|2.05e+0|
> |120|6.96e-1|2.64e+0|1.56e-2|6.68e-1|8.12e-2|1.33e+0|
> |140|3.42e-1|2.34e+0|7.10e-2|7.11e-1|2.38e-1|1.37e+0|
> |160|3.87e-1|3.79e+0|1.33e-2|8.47e-1|1.55e-1|1.72e+0|
> |180|2.38e-1|3.03e+0|1.07e-1|1.03e+0|1.90e-2|1.84e+0|
> |200|4.96e-1|3.99e+0|5.06e-2|8.29e-1|2.73e-2|2.43e+0|
>
> Table 5: mild case (b: bias; v: variance)
>
> |n|SAA_b|SAA_v|ETO_b|ETO_v|IEO_b|IEO_v|
> |--|-----|-----|-----|-----|-----|-----|
> |20|1.22e-1|2.10e+0|1.08e+0|6.37e-1|3.27e-1|1.60e+0|
> |40|5.92e-1|1.85e+0|1.41e+0|6.38e-1|3.75e-1|1.06e+0|
> |60|5.63e-1|2.65e+0|1.28e+0|4.21e-1|4.40e-1|1.71e+0|
> |80|4.66e-1|2.80e+0|9.79e-1|6.69e-1|4.00e-1|1.47e+0|
> |100|1.52e-1|2.62e+0|1.17e+0|1.04e+0|3.87e-1|1.93e+0|
> |120|2.29e-1|2.18e+0|1.09e+0|7.57e-1|3.84e-1|1.60e+0|
> |140|1.72e-1|2.97e+0|1.15e+0|9.11e-1|4.25e-1|1.80e+0|
> |160|2.96e-1|2.37e+0|1.41e+0|4.92e-1|4.17e-1|1.66e+0|
> |180|4.10e-1|2.82e+0|1.38e+0|8.91e-1|4.28e-1|1.81e+0|
> |200|3.41e-1|2.42e+0|1.24e+0|4.45e-1|4.51e-1|1.19e+0|
>
> Table 6: balanced case
>
> |n|SAA_b|SAA_v|ETO_b|ETO_v|IEO_b|IEO_v|
> |--|-----|-----|-----|-----|-----|-----|
> |20|8.84e-2|1.08e-1|6.60e-1|3.56e-2|2.15e-1|9.07e-2|
> |40|3.40e-2|6.37e-2|7.18e-1|1.77e-2|2.21e-1|3.27e-2|
> |60|5.77e-2|6.13e-2|7.13e-1|1.43e-2|2.28e-1|2.66e-2|
> |80|4.82e-2|4.11e-2|7.25e-1|1.17e-2|2.28e-1|2.32e-2|
> |100|1.87e-2|3.52e-2|7.50e-1|6.70e-3|2.31e-1|1.27e-2|
> |120|5.88e-2|3.25e-2|7.25e-1|9.10e-3|2.32e-1|1.49e-2|
> |140|1.45e-2|2.96e-2|7.56e-1|9.10e-3|2.35e-1|1.79e-2|
> |160|3.33e-2|2.06e-2|7.54e-1|5.40e-3|2.35e-1|1.22e-2|
> |180|2.12e-2|2.72e-2|7.40e-1|6.60e-3|2.36e-1|1.55e-2|
> |200|2.25e-2|1.92e-2|7.63e-1|4.45e-3|2.39e-1|9.57e-3|
>
> Table 7: severe case

---

> > ### Comment · Reviewer_JBev · 2025-08-05
> > **Thank you!**
> >
> > The new experiments are a valuable addition—they address my main concerns about finite-sample behavior and help ground the asymptotic insights. I especially appreciate the clear comparison across misspecification regimes and directions. This strengthens the practical relevance of the theoretical results. My overall assessment remains the same.

---

> > > ### Author Response · Authors · 2025-08-09
> > > **Thank you!**
> > >
> > > We are glad that our response and additional experiments have helped address the reviewer's concerns and strengthen our work. Thank you for your valuable comments and positive opinion on our work.

---

### Official Review · Reviewer_bf4G · 2025-07-17

**Clarity:** 3
**Significance:** 3
**Originality:** 3
**Rating:** 4
**Confidence:** 3

**Summary:**

This paper presents a novel theoretical framework to understand the bias-variance tradeoff in Data-Driven stochastic Optimization (DDO) under local model misspecification, an intermediate regime between the classical well-specified and globally misspecified settings. Specifically, the authors compare three popular methods: Sample Average Approximation (SAA), Estimate-Then-Optimize (ETO), and Integrated Estimation-Optimization (IEO), providing a unified asymptotic analysis that distinguishes mild, balanced, and severe misspecification regimes. The paper introduces an explicit characterization of decision bias and variance using influence functions and tools from contiguity theory. Importantly, it derives interpretable conditions under which model misspecification is approximately impactless, shedding light on when model-based methods can still perform well despite misspecification.

**Questions:**

1. The regret ordering results depend on specific expressions involving influence functions and score functions. How robust are these findings to changes in the parametric family or cost function structure? For example, would the same patterns hold under non-smooth costs or under heteroskedastic models?
2. The conclusion mentions potential extensions to constrained or contextual optimization settings. Could the authors clarify what the main challenges would be in applying their local misspecification framework to those more complex scenarios? Would feasibility constraints fundamentally alter the bias-variance tradeoff?

**Ethical Concerns:**

["NO or VERY MINOR ethics concerns only"]

**Final Justification:**

This paper presents a novel theoretical framework to understand the bias-variance trade-off in data-driven stochastic optimization. During the author-reviewer discussion, the authors provided detailed response, which addressed my concerns. So I will maintain my score that is inclined to acceptance.

**Limitations:**

While the theoretical analysis is thorough and well-executed, the paper unfortunately lacks empirical or numerical experiments to illustrate and validate the main findings—particularly the regret comparisons across different misspecification regimes—which limits the practical interpretability and accessibility of the results.

**Quality:**

3

**Strengths And Weaknesses:**

Strengths

1. The paper fills an important gap in the literature by providing a nuanced analysis of the interplay between model misspecification and statistical variability in DDO. This is a first-of-its-kind contribution in this context.
2. The division of results into mild, balanced, and severe misspecification regimes is both intuitive and insightful. Table 1 effectively summarizes the comparative behavior across these regimes.
3. The derivation of explicit bias expressions and the concept of "approximately impactless misspecification direction" provide valuable intuition and theoretical tools that could inform future method design.

Weaknesses

1. The paper is purely theoretical. While this is not a major flaw, a small numerical experiment (even a synthetic one) illustrating the main theoretical phenomena would be helpful and appreciated by a broader audience.
2. Some parts of the paper, especially around the statistical theory (e.g., contiguity, local LAN models), may be challenging for readers unfamiliar with advanced asymptotics. Additional intuition or diagrams (beyond Figure 1) could aid accessibility.

---

> ### Author Rebuttal · Authors · 2025-07-30
>
> Thank you for recognizing our significance and strengths, and for the thoughtful comments. We address your comments and questions as follows:
>
> W1: We understand your comment and have run new experiments. Please refer to our added experiment part in the response to reviewer JBev.
>
> W2: We have already introduced an important lemma, LeCam's third lemma in the appendix. We will add more explanations and references for the contiguity theory, if needed in the appendix. Specifically, we would start from the definition of contiguity for two sequences of measures $P_n$ and $Q_n$. Let $\{P_n\}$ and $\{Q_n\}$ be sequences of probability measures on a common measurable space. We say that $\{Q_n\}$ is contiguous with respect to $\{P_n\}$, written as $Q_n \triangleleft P_n$, if for every sequence of measurable events $A_n$, $P_n(A_n) \to 0 \quad \Rightarrow \quad Q_n(A_n) \to 0$. If both $Q_n \triangleleft P_n$ and $P_n \triangleleft Q_n$, then we say the sequences are mutually contiguous and write $P_n \sim Q_n$. We will introduce the criterion for contiguity (Le Cam's first lemma): Let $\{P_n\}$ and $\{Q_n\}$ be sequences of probability measures on $(\mathcal{X}_n, \mathcal{A}_n)$ such that $Q_n \ll P_n$ for all $n$, and define the likelihood ratio $Z_n = \frac{dQ_n}{dP_n}$. Then the following are equivalent:
> (1) $Q_n \triangleleft P_n$, i.e., $Q_n$ is contiguous with respect to $P_n$.
> (2) If along some subsequence, $\frac{dP_n}{dQ_n} \to U$ in distribution under $Q_n$, then $\mathbb{P}(U > 0) = 1$.
> (3) If along some subsequence, $\frac{dQ_n}{dP_n} \to V$ in distribution under $P_n$, then $\mathbb{E}\_P[V] = 1$.
> (4) For any sequence of statistics $T_n$, if $T_n \to 0$ in $P_n$-probability, then $T_n \to 0$ in $Q_n$-probability.
> We will introduce asymptotic log normality for two mutually contiguous $P_n$ and $Q_n$: $\log \frac{dP_n}{dQ_n} \xrightarrow{d} e^Z, \quad \text{under } Q_n, \quad \text{where } Z \sim \mathcal{N}(\mu, \sigma^2)$, then we have: (1) $Q_n \triangleleft P_n$, and (2) $P_n \triangleleft Q_n$ if and only if $\mu = -\frac{1}{2} \sigma^2$. We will review quadratic mean differentaiblity and local alternatives, definitions and properties of local asymptotic normality (LAN) in further details.
>
> Q1: Let us respond to four aspects of your question:
>
> 1) We emphasize that in Theorems 1-6, while the bias and variance terms depend on specific expressions involving influence functions and score functions, the orderings of the bias and variance are universal among all cost function structures (influence functions) and parametric families (score functions). The regret ordering in the mildly and severely regimes are also universal among all influence functions and score functions. The only undetermined regime is the balanced regime, where the orderings of bias and variance are opposite, thus making the regret (the sum of bias and variance) ordering undetermined. However, we empirically demonstrate the regret ordering in the added numerical experiments and find that IEO may achieve the best bias-variance tradeoff and the lowest regret among all three methods in the newsvendor problem.
>
> 2) We only assume the smoothness of the cost function \textit{in expectation}, instead of the cost function itself. The former assumption is much weaker and general than the latter. That is, the cost function itself can be possibly non-smooth in our framework. For example, the cost function of the newsvendor problem has kinks and is not smooth. Nonetheless, if the distribution is continuous, then the expected cost will be smooth and satisfy our framework.
>
> 3) If by "heteroskedastic" the reviewer means that the cost function taken in the expectation can have different noise depending on the decision, then yes our framework allows that. In fact, we don't even make an explicit distinction between "heteroskedastic" and "homoskedastic" in the paper.
>
> 4) Our framework focuses on nonlinear objective function, and does not apply to linear programming settings per se. However, we should point out that the latter setting is easier than our nonlinear setting from a statistical perspective. This is because if the cost function is linear, then it suffices to predict the expectation of the cost vector, which is a point prediction [1]. Because of this, fitting the whole distribution like we do is not needed. In contrast, our setting require the notion of model misspecification that is at the distributional level, since the nonlinearity of the objective function forces a more complex relation between the randomness and the decision variables.
>
> Q2: Our framework can be extended to other settings. For contextual optimization, the extension is quite immediate. Specifically, we introduce the feature $\boldsymbol{x}$ in addition to the random parameter $\boldsymbol{z}$. We write the joint distribution of $\boldsymbol{x}$ and $\boldsymbol{z}$ as $Q(\boldsymbol{x},\boldsymbol{z})=Q(\boldsymbol{x})Q(\boldsymbol{z}|\boldsymbol{x})$. In this case we need to estimate $Q(\boldsymbol{z}|\boldsymbol{x})$ using parametric family $Q_{\boldsymbol{\theta}}(\boldsymbol{z}|\boldsymbol{x})$. Note that in the contextual case, the SAA method is not interesting because it  allows to choose any map from context to decision, thus easily overfitting the
> finite-sample problem. In this case, IEO and ETO are the legitimate approaches. In the mildly misspecified case, ETO is better than IEO while in the balanced and misspecified case, IEO is better than ETO.
> For constrained optimization, if the constraints are deterministic, then we expect our analysis and insights to continue to hold, but it would introduce additional technical complexity from the KKT conditions instead of using simpler optimality conditions. On the other hand, if the constraints are stochastic, then there is an additional question of feasibility (e.g., in chance constrained settings where random safety conditions need to be enforced with high probability). These extensions to constrained settings warrant future work in our view.
>
> [1] Elmachtoub, Adam N., Henry Lam, Haofeng Zhang and Yunfan Zhao. "Estimate-Then-Optimize versus Integrated-Estimation-Optimization versus Sample Average Approximation: A Stochastic Dominance Perspective." arXiv preprint arXiv:2304.06833 (2023).

---

> > ### Comment · Reviewer_bf4G · 2025-08-05
> > **Reply to the Rebuttal and Further Comment**
> >
> > Thanks for the authors' response. Based on my reading of the response and other reviewers' comments, it seems that the assumption that the optimal solution lies in the interior of the feasible region does not apply to linear programs (LP). Note that LP is the most fundamental optimization problems in operations research. If that is the case, it will undermine the value and practicability of this work. The authors need to clarify this issue.

---

> > > ### Author Response · Authors · 2025-08-06
> > > **Further clarification about cost function (1/2)**
> > >
> > > We fully understand the reviewer's comments. We are happy to provide clarifications which, in fact, relate to a whole range of aspects beyond just about whether the solution is in the interior or not as the reviewer asked about.
> > >
> > > First of all, while linear optimization (LP) is clearly fundamental in operations research, nonlinear optimization is equally fundamental: Many operational applications, including revenue management and pricing, inventory control, and portfolio optimization, belong to nonlinear optimization. Please see the monographs [5], [6], [7] and [8] that devote to these nonlinear optimization problems.
> > >
> > > Our focus in this paper is nonlinear optimization which, as described above, is equally important as LP. To this end, we should clarify that analyzing the statistical performances of data integration, for linear and nonlinear problems, each has its own substantial challenges and respective techniques. In the literature, almost all papers focus on either linear or nonlinear settings, because the involved challenges and techniques are very distinct, and it is very difficult to investigate both settings in a single paper. For instance, among the recent works, [1] and [2] studied linear settings, while [3] and [4] studied nonlinear settings. That is, while we believe we can investigate a similar direction as our current paper for linear optimization problems, to our understanding of the literature, this would require substantially different approaches and belong to a different paper.
> > >
> > > Now, let us explain why linear and nonlinear optimization settings are statistically very different. The solution being interior or not is, in fact, only one of many differences between linear and nonlinear settings. More fundamentally, linear and nonlinear optimization problems exhibit different types of large-sample asymptotic convergences and, correspondingly, the techniques in arguing whether one data-driven approach is better than another are also different. For example, in arguing ETO is better than IEO in the well-specified setting for nonlinear optimization, [4] uses a stochastic dominance on the asymptotic regrets of ETO and IEO which turn out to exhibit the same order (in sample size) under the same type of central limit theorems, where the difference between ETO and IEO only lies in the variance in the Gaussian-type limit. In contrast, in arguing ETO is better than IEO in the well-specified setting for linear optimization, [2] derives that ETO and IEO have different rates in the sample size, by using a noise condition (Assumption 2 in [2], also called margin condition) that applies specifically to linear programs, and concludes the superiority of ETO via its faster rate of convergence. Technically, these differences between linear and nonlinear problems arise from the dual non-degeneracy of LP, in that [2] utilizes the distance to dual degenerate point as a problem-dependent parameter to get the fast rate of ETO. This structure is generally absent in nonlinear optimization, whose analysis is instead based on smoothness and first- and second-order optimality conditions that are very different from the linear setting where the structure is much closer to classification. Furthermore, besides the differences in asymptotic behaviors and techniques, the levels to which data are integrated into the optimization objectives are also different between linear and nonlinear optimization. As we have explained as part of our previous response to your Q1, the linear structure in the objective function of an LP means that it suffices to estimate the expectation of the underlying randomness (or conditional expectation in the extension to contextual optimization), while in the nonlinear optimization case typically the entire distribution of the randomness needs to be estimated due to the more complex relation between the randomness and the decision variables.
> > >
> > > We once again would like to express our great appreciation of the reviewer's comments. In view of our response above, we will add a clarification in the paper by modifying line 34 to "Our focus in this work is the statistical performance of model-based approaches for data-driven optimization with nonlinear cost objectives." and line 129 to "$c(\cdot,\cdot)$ is a known nonlinear cost function". We will also move our Examples 2 and 3 (newsvendor and portfolio optimization) in the appendix to the main body, along with the relevant citations like those above, to explain the importance of nonlinear cost objectives.
> > >
> > > The references of this response are in the next response.

---

> > > > ### Author Response · Authors · 2025-08-06
> > > > **Further clarification about cost function (2/2)**
> > > >
> > > > [1] Elmachtoub, Adam N., and Paul Grigas. "Smart “predict, then optimize”." Management Science 68.1 (2022): 9-26.
> > > >
> > > > [2] Hu, Yichun, Nathan Kallus, and Xiaojie Mao. "Fast rates for contextual linear optimization." Management Science 68.6 (2022): 4236-4245.
> > > >
> > > > [3] Qi, Meng, Paul Grigas, and Zuo-Jun Max Shen. ``Integrated conditional estimation-optimization.'' arXiv preprint arXiv:2110.12351 (2021).
> > > >
> > > > [4] Elmachtoub, Adam N., Henry Lam, Haofeng Zhang and Yunfan Zhao. "Estimate-Then-Optimize versus Integrated-Estimation-Optimization versus Sample Average Approximation: A Stochastic Dominance Perspective." arXiv preprint arXiv:2304.06833 (2023).
> > > >
> > > > [5] Bertsekas, Dimitri P. Nonlinear Programming. 3rd ed., Athena Scientific, 2016.
> > > >
> > > > [6] Markowitz, Harry. Portfolio Selection: Efficient Diversification of Investments. Yale University Press, 1959.
> > > >
> > > > [7] Talluri, Kalyan T., and Garrett J. van Ryzin. The Theory and Practice of Revenue Management. Springer, 2004.
> > > >
> > > > [8] Bazaraa, Mokhtar S., Hanif D. Sherali, and C. M. Shetty. Nonlinear Programming: Theory and Algorithms. 3rd ed., Wiley, 2006.

---

> > > > ### Comment · Reviewer_bf4G · 2025-08-08
> > > > **Reply**
> > > >
> > > > Thanks for the authors' clarifications, which have addressed my concerns.

---

> > > > > ### Author Response · Authors · 2025-08-09
> > > > > **Thank you!**
> > > > >
> > > > > We are glad that our response has helped clarify and addressed your concerns. Thank you again for all your valuable comments and positive opinion on our work.

---

### Official Review · Reviewer_jkSD · 2025-07-21

**Clarity:** 2
**Significance:** 3
**Originality:** 3
**Rating:** 4
**Confidence:** 4

**Summary:**

This paper analyzes two different data-driven stochastic optimization approaches -- estimation-then-optimize (ETO) and integrated estimation-optimization (IEO) -- in their comparison with a popular paradigm called the sample average approximation (SAA). The formulations of ETO and IEO are standard and follow existing literature and the settings of discussion combines the assumptions by Elmachtoub et al. [2023] with a misspecification model referred to as "local misspecification”. This model introduces three levels of misspecification: mild, balanced, and severe, which correspond to the presence of "deviations" from the true distribution at the rate of $o(n^{-2\alpha})$ with $\alpha>1/2$ (mild), $\alpha=1/2$ (balanced) and $0<\alpha<1/2$ (severe), respectively. These scenarios extend the discussion by Elmachtoub et al. [2023] in that the latter considers a well-specified case and a generic misspecified case. The paper claims to have explicated the asymptotic tradeoffs among ETO, IEO, and SAA in terms of their variance, bias, and regret. Discussions alike are of significance, as they can characterize the impact of misspecification on decision quality and inform the selection of data-driven approaches.


Ref:
A. N. Elmachtoub, H. Lam, H. Zhang, and Y. Zhao. Estimate-then-optimize versus integratedestimationoptimization: A stochastic dominance perspective. arXiv preprint arXiv:2304.06833, 2023.  (Please note that there is a more recent version of this paper made available after NeurIPS submission deadline).

**Questions:**

Q1. The authors should carefully check whether the structural assumption invoked in Line 901 is actually necessary for the proof of Lemma 4.

Q2. Some improvements in the wording of Table 1 are needed for accuracy. In particular, the description of the variance for ETO, IEO, and SAA in the severe misspecification case as “$\approx 0$” (and therefore “negligible”) may be misleading. It appears that the variance does not vanish, but rather becomes dominated by other terms --- such as bias --- even though the variance may still be non-trivial. Rephrasing this to reflect the relative scale, rather than suggesting the negligibility of variance, would provide a more accurate interpretation.

Q3. The paper assumes several technical conditions, including the invertibility of the Hessian matrix, twice continuous differentiability, and the full-rankness of $\Sigma$. The paper can be improved by providing concrete examples from applications where such conditions are satisfied.

Q4. There are multiple notational inconsistencies that need to be addressed. For instance, “$v(w,z)$” in Eq. (2) where $z$ is some parameters v.s. ``$v(w,Q_t)$’’ where $Q_t$ is some distribution in Line 903.

Q5. The paper would benefit from adding a section (such as “Assumptions and Preliminaries”) that summarizes and clarifies key concepts. This section could be used to consolidate notations with subtle distinctions, explicitly state key assumptions used throughout the paper, and discuss useful known results and those that differ marginally from existing findings.

Q6. Please update the information for all the references. E.g., A. N. Elmachtoub [2023]

Q7. Line 1009: typo: “mulptiplication”.

**Ethical Concerns:**

["NO or VERY MINOR ethics concerns only"]

**Final Justification:**

I would like to thank the authors again for their time and effort in carefully responding to my questions. While my initial main concern was that the paper might contain technical flaws in the proofs, the authors’ explanation clarified that the issues mostly stemmed from unintended oversight or lack of exposition. While my concerns regarding the clarity and rigor of the paper remain, I think many of them could be addressed according to the authors' rebuttal. Therefore, I consider reasons for acceptance to outweigh those for rejection in this case.

**Limitations:**

yes

**Paper Formatting Concerns:**

On several occasions, the tables or formulas are excessively long and extend beyond the page margins. E.g., Line 77-78; Line 985; Line 989.

**Quality:**

2

**Strengths And Weaknesses:**

*Strength:*

S1. The consideration of a new misspecification model, which captures varying degrees of deviation from the true underlying distribution (mild, balanced, or severe), appears to be a potentially original and meaningful contribution. Since misspecification may diminish as decision-makers acquire more knowledge from data, this modeling framework captures a more realistic aspect of data-driven decision-making in practice.

S2. The paper offers analytical comparisons of three important data-driven optimization schemes --- ETO, IEO, and SAA ---  in terms of three key performance metrics: bias, variance, and regret. The results can provide insights for selecting appropriate tools in different contexts of decision-making.


*Weaknesses:*

W1. On multiple occasions, the rigor of the analysis and proofs could be improved by adding explanations and justifications. Below are a few examples mostly in the proofs for Lemma 4 and 5, which serve as the bases for many  conclusions in the paper:

   1.a. (Line 901) The proof of Lemma 4 invokes a structural assumption similar to that of Example 5 in the appendix, which may render the proof inapplicable to the statement of Lemma 4. (See Question Q1 below).

   1.b. In line 904, how to obtain $\frac{\partial}{\partial t} \log q_t(\boldsymbol z)\vert_{t=0}=u(\boldsymbol z)$ from Definition 1 requires additional exposition, although the conclusion in Line 904 seems correct.

   1.c. The proof of Lemma 4 begins by setting the gradient of the cost function at $w_t$ to be zero. However, this argument is not valid in the presence of constraints $\Omega$.

   1.d.  In view of the assumption of an open feasible region $\Omega$, the analysis seems to have overlooked the possibility that $w_t$ (as well as $w^{\ast}$  and $w^{\ast}_n$ ) may not exist.

   1.e. It is non-trivial to see (i) why $\lim_{n\rightarrow \infty} \sqrt{n} (w_n^*-w_{\theta_0})=\lim_{t\rightarrow 0}\frac{1}{t}(w_t-w_0)$ at the end of Line 906 and (ii) how it generalizes to Line 907, particularly in view of the fact that “t” is substantiated differently into $\sqrt{n}$ v.s. $n$ in (i) and $n^{\alpha}$ v.s. $n$ in (ii), respectively.

   1.f. Similar issues (1.a--1.c) from above also apply to Lemma 5.

   1.g. How Line 966—967 are obtained requires further exposition.


W2. The paper can benefit from some careful and thorough editing to improve clarity. Below are a few examples:

  2.a Line 256—257: The formulations of variance, bias, and regret are not explicitly justified in the main text. While their interpretations can be implicitly inferred from the statements of the theorems. Providing clearer definitions or more direct justification would improve readability.

  2.b Critical assumptions (Assumptions 3 and 4) are not discussed until appendix.

  2.c. Related to 2.b, the justifications of some assumptions are provided within the proofs, rather than being stated upfront. (E.g., Line 969). For clarity, it would be helpful to motivate these assumptions earlier in the main text

  2.d Line 897—898 are pillar results for the rest of the analysis. How they are inferred from Lemma 4 is non-trivial to see, despite that the same reasoning as 897–898 is repeated in the proof of Lemma 4.

  2.e. Notations with only minor variations are introduced throughout the paper to represent conceptually distinct meanings used thereafter. Sometimes these notations are introduced on the fly as the discussion progresses. (E.g., $w_n^*$ and $v_n$ before Line 250). A more deliberate and consistent presentation of notations would improve clarity. (See Question Q5).

  2.f. The assumptions underlying Lemmata 2, 4, and 5 appear to differ, but they are not explicitly stated alongside the lemmas. It would be necessary to include the relevant assumptions directly in each lemma’s statement.

  2.g. Proposition 1 appears to be only marginally different from existing results in the literature and is focused on the ``well-specified'' case. In its current form, it would seem more appropriate to move this proposition to a section (See Question Q5) that reviews or introduces known results/concepts, rather than in the section titled “Mild Misspecification.”

 2.h. The wording of Definition 1 should be revised, as it does not explicitly define the term “local perturbation.” Instead, it introduces an assumption in Lines 193–194 without clearly stating the definition. Making the concept and terminology of “local perturbation” explicit in the statement of that definition would improve clarity.

  2.i. The use of the term “local misspecification” is not sufficiently justified. While the paper refers to Definition 1 as standard in asymptotic statistics, e.g., in [van der Vaart, 2000], this specific terminology does not appear to be introduced in that reference. Instead, the same mathematical formulation is presented in Chapter 7 of that reference in the context of local asymptotic normality (LAN). The use of LAN pertains to a somewhat different setting. A clearer exposition of the distinction between the adopted terminology and its original context as well as the justification for its current usage, would enhance rigor and clarity.



*Ref:*

W. van der Vaart. Asymptotic statistics, volume 3. Cambridge university press, 2000.

---

> ### Author Rebuttal · Authors · 2025-07-30
>
> Thank you for the detailed review and the list of thoughtful comments. In the following, we address all your comments/questions, including those in the proof details in the appendix.
> Regarding many of your presentation concerns, in case of acceptance, we will use the additional page allowed for the camera ready to move more intuition and definitions to the main body of the paper. Specifically, we will add Assumption 3 and 4, the explanation on line 969-970 and the numerical experiments in the additional page.
> Below, you can find our responses to each concern.
>
> W1.a: No, the structural assumption in line 901 (exponential tilt, essentially Example 5) is NOT necessary to prove Lemma 4. Definition 1 is more general and sufficient. We apologize for adding the exponential tilt assumption and making readers confused.  We will delete this redundant assumption in the revision.
>
> W1.b: The equality comes from the quadratic mean differentiability property in standard asymptotic statistics (Page 94, Section 7.2 of [1]). Specifically, Definition 1 indicates that $\frac{1}{2}u(z)\sqrt{p_{\theta_0}(z)}$ is the derivative of the map $t\to \sqrt{q_t(z)}$ at $t=0$ (note that $q_0=p_{\theta_0}$) for almost every $z$. In this case, $u(z)=2\frac{1}{\sqrt{q_0(z)}}\frac{\partial}{\partial t}\sqrt{q_t(z)}=\frac{\partial}{\partial t}\log q_t(z)$ at $t=0$.
>
> W1.c: In line 128, we assume that $\Omega$ is an open set, thus every point in $\Omega$ lies in the interior. Because of this, it is valid to set the gradient to $0$.
>
> W1.d: In line 866, we assumed the existence of minimizers ($w\_t$, $w\^{\*}$, $w\_n^\*$) in the open feasible region in Assumption 3, which is a standard assumption in asymptotic statistics (assumptions in Theorem 5.7 and Theorem 5.23 of [1]).
>
> W1.e: In line 906, once we get the functional limit for $t$, the sequential limit for $n$ is an immediate result (since $1/n^\alpha$ is a sequence that approaches to $0$ for all $\alpha>0$) from the sequential criterion for functional limits (Theorem 4.2.3 of [Abbott, 2015]). Please note that the notation for $\boldsymbol{w}_n^*$ and $Q_n$ already implicitly considers the order $\alpha$ in line 249-250 and Definition 2.
>
> W1.f: Very similar explanations as the above hold for Lemma 5.
>
> W1.g: Line 966 is the consequence of the former analysis (962-966) on the middle term. Line 967 is the beginning of analyzing the first term based on using definitions of $O_p(\cdot)$.
>
> W2.a: Thanks for your suggestions. We have provided some descriptions on bias and variance in Theorem 1 and provided the exact definition for regret in Definition 3. We will make it clearer in the final version.
>
> W2.b: We aim to demonstrate our own novel framework and results in the main body. We defer Assumptions 3 and 4 to Appendix (1) for the sake of page length; (2) since they are standard regularity assumptions in the existing asymptotic statistics literature.
> That said, we will use the additional page for the camera ready to discuss these in the body of the paper.
>
> W2.c: Thanks for your suggestions. We have justified the validity of the assumption by citing previous literature in line 262. We will move further explanations (currently in line 969) here.
>
> W2.d: We hope our explanations for W1.e answers these questions.
>
> W2.e: Thanks for your suggestions. We will display the notations in a more concentrated manner in the final version.
>
> W2.f: Lemma 2 is sufficient to prove using Example 5 and Definition 2 (they are already mentioned in the statement). Lemmas 4 and 5 are both based on the same assumptions in Definition 1 (and interchangeability assumptions). The difference is the result rather than the assumptions, where Lemma 4 focuses on $\boldsymbol{w}$ while Lemma 5 focuses on $\theta$. The limit is taken by sending $t\to 0$ thus also holds by taking $1/n^\alpha\to 0$ for any $\alpha>0$ and $n\to\infty$  (and thus for all $1/n^\alpha$ with $\alpha>0$).
>
> W2.g: We recognize that Proposition 1 is mostly an existing result from previous literature [3] (that is why we call it proposition instead of theorem). We aim to demonstrate that the existing result can be extended beyond the well-specified case to the mildly-misspecified case. We will add the reference in Proposition 1 and some clarifying text around it.
>
> W2.h: Definition 1 is to describe the relationship between the ground truth distribution and the parametric distribution (model class). Definition 1 is rather general and we have included many examples in the Appendix. We will make it more explicit by adding the description "the distribution $Q_t$ is a \textit{local perturbation} of the parametric family $\{P_{\boldsymbol{\theta}}:\boldsymbol{\theta}\in\Theta\}$ at $\boldsymbol{\theta}_0$ if it satisfies for all $t>0$, $q_t$ is differentiable and satisfies the following quadratic mean differentiablity condition (line 194)" in the revision.
>
> W2.i: We have extensively discussed the term "local misspecification" in related works section (line 112 to 124) with many classical literature in statistics, econometrics and machine learning, e.g. [2]. Specifically, [2] uses the terminology "local misspecification" in Section 2. We have also technically explained the terminology in line 206-217.  We build a universal, general and rigorous local misspecification theoretical framework based on these literature and on the technical tools from [1], despite the fact that the original "local asymptotic normality (LAN)" is used for other settings. For these reasons, we believe our use of the term "local misspecification" and the relation to past literature are sufficiently clear, but we are happy to revise the exposition accordingly if the reviewer still sees the need.
>
> Q1: Please see W1.a
>
> Q2: Thanks for your suggestions. The exposition in Table 1 is primarily to give readers a rough sense of our results, before loading them with many technical details. We have stated in the caption that $\approx 0$ means ``asymptotically negligible", and we have described in our main results much more accurately how it is in comparison to other terms and how the relative scale may depend on different regimes. Specifically, in the severe case, we build a convergence in probability result. In this case, the limit is a fixed vector (or, say, a degenerate distribution with zero variance). If you feel strongly, we can directly state the convergence result in the table, but it might hurt readability a bit.
>
> Q3: The technical assumptions including the invertibility of the Hessian/Fisher information matrix, twice continuous differentiability, and the full-rankness, mainly indicate the well-posedness of the problem and are often used in standard statistical literature (e.g. Theorems 4.6, 5.23, 5.39 in [1] have discussed the full-rankness, invertibility of Fisher information matrices and second-order Taylor expansion, etc.). We will add further discussions on the satisfaction of these assumptions in our application examples in Appendix A.1 (note that, on the other hand, we have provided many examples in Appendix A.2 regarding local perturbation, as this is the more interesting concept in this paper).
>
> Q4: The notation $v(\boldsymbol{w},\theta)$ (not $v(\boldsymbol{w}, z)$ as you wrote) is a shorthand for $v(\boldsymbol{w}, P\_{\theta})=\mathbb{E}\_{P_{\theta}}[c(\boldsymbol{w}, z)]$.  The notation $v(\boldsymbol{w}, Q_t)$ stands for $\mathbb{E}\_{Q_{t}}[c(\boldsymbol{w}, z)]$. They are consistent, but we will make it more clear in the revision. Specifically, we will stop using $v(\boldsymbol{w},\theta)$ and instead stick to $v(\boldsymbol{w}, P_{\theta})$.
>
> Q5: Please see W2.e
>
> Q6: Yes we will update all references, including [3] for which the reviewer has kindly noted there is an updated version.
>
> Q7: We will fix this typo in the revision.
>
> [1] Van der Vaart, Aad W. Asymptotic statistics. Vol. 3. Cambridge university press, 2000.
>
> [2] J. Fan, K. Imai, I. Lee, H. Liu, Y. Ning, and X. Yang. Optimal covariate balancing conditions in propensity score estimation. Journal of Business \& Economic Statistics, 41(1):97–110, 2022
>
> [3] Elmachtoub, Adam N., Henry Lam, Haofeng Zhang and Yunfan Zhao. "Estimate-Then-Optimize versus Integrated-Estimation-Optimization versus Sample Average Approximation: A Stochastic Dominance Perspective." arXiv preprint arXiv:2304.06833 (2023).

---

> ### Comment · Reviewer_jkSD · 2025-08-01
>
> I would like to thank the authors for going through my comments carefully. I have several remaining concerns/questions and will start with the ones that I found to be more critical. I may add more subsequently.
>
> Re: W1.c: Your statement seems at least not very rigorous. Consider (the many variations of) the one-dimensional problems: $\min  x: -1<x<1$ and $\min x^2: 1<x<2$. Here, the feasible regions are open sets. Assigning gradient of objective to zero does not work.
>
> Re: W1.d: It is unclear to me where the existence of these solutions was stated in Assumption 3, nor is there seemingly explicitly mentioning of interior of the feasible region. Meanwhile, there is a mysterious $\epsilon$ defined but unused in clause 2 of that assumption.
>
> Re: W1.e: Please provide some more relative detail. In particular, based on your notation, as $t$ changes, the sample size $n$ used to obtain, e.g., the SAA solution $w_t^{\ast}$ should vary accordingly. However, when you assign $t = 1/n^\alpha$, the notation $w_n^{\ast}$ does not appear to correctly represent the sample size for the corresponding SAA solution.
>
> Re: Q3: My concern was that the simultaneous satisfaction of the conditions in the list may appear uncommon and hard to verify, even though some subsets of (variations of) those conditions have been discussed in the literature. To resolve my concern, I wonder if the authors could provide some examples of problems/functions meaningful to machine learning or data-driven decision-making in their response.

---

> ### Author Response · Authors · 2025-08-01
> **Further technical clarification**
>
> We thank the reviewer for the response. We would like to respond as follows:
>
> Re: Re: 1.c: Since in Assumption 3 we assume the existence of the minimizer(or maximizer) that belongs to the open set, we can set the gradient to $0$. In your first example, the minimizer does not exist in the open interval $(-1,1)$. Our paper mainly focus on the case where the optimizer lies in the interior of the feasible region. The case is standard in statistical theory. As a remedy for the boundary case, potentially it is possible to extend the framework to the constrained case. In your example, we can take $x\in\mathbb{R}$ ($\mathbb{R}$ is an open set) and add two constraints $x+1\leq 0$ and $x-1\leq 0$. More generally, for the constrained case, we can add equality constraints and inequality constraints on the feasible set. To address the constraints, we can consider the Lagrangian function and corresponding Lagrangian multipliers. Under standard KKT conditions (such as Assumption A, B, C and D in [1], including complementary slackness, linear independent constraint qualification, and some assumptions on the relative interior, see Assumption B), potentially, the asymptotic behaviour of the three methods may still hold. This is left as our future research direction.
>
> Re: Re: 1.d: We apologize for missing a notation in the clause 2 in Assumption 3. The revised version is : there exists $\zeta$ $\zeta\^\ast=\arg\max\_{\zeta}\mathbb{E}\_Q[m\_\zeta(z)]$, for all $\varepsilon>0$,
>         $\sup\_{\zeta:\|\zeta-\zeta\^\ast\|\geq \varepsilon}$, .... The existence of these solutions is due to the existence of the maximizer $\zeta^\ast$. Also, we revise in line 877 "for IEO $m_\zeta(z)=-c(w_\theta,z)$" since we missed a minus sign there. For further technical details of Assumption 3, we refere to the assumptions in Theorem 5.7 and Theorem 5.23 in [2].
>
> Re: Re: 1e: We recall the definition of $w\_n^\ast:=\arg\min\mathbb{E}\_{Q^n}[\frac{1}{n} \sum_{i=1}\^n c(w, z)]$, where $Q\^n:=Q\_{1/n^{\alpha}}\^{\otimes n}$ is defined in line 214 as the $n$-fold product distribution. By setting $t=1/n^\alpha$, now $w\_n^\ast=\boldsymbol{w}\_{1/n\^\alpha}$, by the definition of $w\_t$ in line 895-896. To be even more explicit, for fixed a $\alpha$, we have $w\_n^\ast=\arg\min_{w\in\Omega}\mathbb{E}\_{Q\_{1/n\^\alpha}}[c(w, z)]$.
>
> Re: Re: Q3: We verify our assumptions by (1) citing a generic example and  (2) providing a more concrete example. The cited generic example is from [3]. Consider the newsvendor problem $c(w,z)=a^\top(w-z)^++d^\top(z-w)^+$. We can see that Proposition 3 in [3] shows that the newsvendor problem where each product $j$ with demand distribution $N(\theta j, \sigma_j)$ satisfies their Assumptions 1, 2 (which corresponds to our Assumption 3) and Assumption 3 (which corresponds to  our Assumptions 1 and 4).
>
> For a concrete example, we consider the portfolio optimization problem with $c(w, z) = \frac{1}{2} w^\top (z - \mathbb{E}[z])(z - \mathbb{E}[z])^\top w - w^\top z + \frac{1}{2} w^\top w$, where $z \in \mathbb{R}^3$ denotes the assets’ return and follows a multivariate Gaussian distribution $z \sim \mathcal{N}(B(\theta), \Gamma)$ where $\theta \in \mathbb{R}$is an unknown scalar parameter, $B(\theta) = (\theta, 2\theta, 3\theta)^\top \in \mathbb{R}^3$, $\Gamma \in \mathbb{R}^{3 \times 3}$ is a known, positive definite covariance matrix. We get: $\mathbb{E}\_\theta[c(w, z)] = \frac{1}{2} w^\top (\Gamma + I) w - w^\top B(\theta).$ We define this expected cost as: $v(w, \theta) := \frac{1}{2} w\^\top (\Gamma + I) w - w\^\top B(\theta)$. We now calculate the gradient and Hessian. Let $Q :=\Gamma + I$, which is symmetric and positive definite. Then: $\nabla_w v(w, \theta) = Q w - B(\theta)$, $\frac{\partial v}{\partial \theta} = -w^\top \frac{d B(\theta)}{d \theta} = - (w_1 + 2w_2 + 3w_3)$ The Hessian of $v(w, \theta)$ with respect to $(w, \theta)$ has the blocks corresponding to our Assumption 1: $V=Q$ (invertible), and $\Sigma= -(1,2,3)$ (full rank). To minimize $v(w, \theta)$, solve the first-order condition: $\nabla_w v(w, \theta) = Q w - B(\theta) = 0$. Using $B(\theta) = \theta \cdot b$, where $b = (1, 2, 3)^\top$, we get: $w_\theta = \theta Q^{-1} b$. First derivative: $\frac{d w_\theta}{d\theta} = Q^{-1} b$. Second derivative is $0$. Fisher information: $\mathcal{I}(\theta) = \left( \frac{d B(\theta)}{d\theta} \right)^\top \Gamma^{-1} \left( \frac{d B(\theta)}{d\theta} \right)=b^\top \Gamma^{-1} b$, which is positive, thus invertible (it is a scalar).
>
> [1] Duchi, John C., and Feng Ruan. "Asymptotic optimality in stochastic optimization." (2021): 21-48.
>
> [2] Van der Vaart, Aad W. Asymptotic statistics. Vol. 3. Cambridge university press, 2000.
>
> [3] Elmachtoub, Adam N., Henry Lam, Haofeng Zhang and Yunfan Zhao. "Estimate-Then-Optimize versus Integrated-Estimation-Optimization versus Sample Average Approximation: A Stochastic Dominance Perspective." arXiv preprint arXiv:2304.06833 (2023).

---

> ### Comment · Reviewer_jkSD · 2025-08-04
>
> Thank you for your response; please see my questions regarding your second-round responses below:
>
> Re: Re: 1.d: Thanks for your explanation and for pointing out how to correct the notations. However, if my understanding were correct, even after fixing the previous mistake in the notations, there seem to be several remaining notational issues. In particular, Clause 2 in Assumption 3 does not ensure the existence of optimal solutions within the interior of the feasible region in that there is no mentioning of feasible set in the statement.
>
> Re: Re: 1e: In view of your response, could you explain why the conclusions before line 907 would be consistent with the second formula in Line 939--940?
>
>
> Please see my additional questions regarding your first-round responses below:
>
> Re: W1.b:  Please provide proper references to the sources to improve clarity and give proper credit.
>
> Re: W2.f:  Definition 1 alone is not sufficient for Lemma 4 and 5. For an example, the existence of minimizers/maximizers in the interior of the feasible region has to be assumed.

---

> > ### Author Response · Authors · 2025-08-06
> > **Further technical clarification**
> >
> > We thank the reviewer for the response. We would like to respond as follows:
> >
> > Re: Re: Re: 1d: We understand the reviewer's point; however, we have stated clearly right after Assumption 3 (which is a placeholder assumption) what $\zeta$ stands for in the three cases SAA, ETO and IEO. In these cases, $\xi$ is either $w$ or $\theta$. Since throughout the paper we have assumed $ w$ is in the open set $\Omega$ and $\theta$ is in the open set $\Theta$ (as stated clearly in line 128 and line 133), a reader who is following our paper can only understand our application of Assumption 3 as having the corresponding region $\Omega$ or $\Theta$. We have chosen not to introduce further notation because of this obvious reason. However, if the reviewer feels we should, we can certainly introduce an extra notation, say $\Xi$ (open set), to denote the space of $\zeta$ in Assumption 3 so that the statements hold under the condition $\zeta\in\Xi$.
> >
> > Re: Re: Re: 1e: They are consistent. As we mentioned in our previous response, $w_n^\ast$ is a vector that depends on $\alpha$:  for fixed $\alpha$, $w_n^\ast=\arg\min_{w\in\Omega}\mathbb{E}\_{Q_{1/n^\alpha}}[c(w, z)]$ defined at the bottom of page 6, which translates to $w_n^*=w_{t}$ at $t =1/n^\alpha$ by the definition of $w_t$ in line 895-896. The conclusion before line 907 asked by the reviewer is under the balanced case ($\alpha=1/2$), while the second formula between line 939-940 asked by the reviewer is under the mildly misspecified case ($\alpha>1/2$).
> >
> > Re: Re: W1.b: We will cite the exact page and section of the reference mentioned in our previous response (i.e., Page 94, Section 7.2 of [1]) in our paper. However, we also want to make clear that the calculation is actually very elementary -- it involves only standard calculus to take the derivative of the square root of a density function. In other words, we are happy to add the reference, but we doubt it is at all meaningful to do so.
> >
> > Re: Re: W2.f: We want to politely point out to the reviewer that in the statements of Lemmas 4 and 5, we have explicitly defined $w_t:=\arg\min_{w\in\Omega}\mathbb{E}\_{Q_t}[c(w,z)]$, $\theta_{t}^\textup{KL}:=\arg\max_{\theta\in\Theta}\mathbb{E}\_{Q_t}[\log p_\theta(z)]$ and $\theta_{t}^\ast:=\arg\min_{\theta\in\Theta}\mathbb{E}\_{Q_t}[c(w_\theta,z)]$. For a reader who is following our paper and has the usual math training, $w_t$, $\theta_{t}^\textup{KL}$ and $\theta_{t}^\ast$ would be understood to exist and be unique (otherwise, these definitions of $w_t$, $\theta_{t}^\textup{KL}$ and $\theta_{t}^\ast$ inside Lemmas 4 and 5 would not make sense). Note also that we have assumed throughout the paper that $\Omega$ and $\Theta$ are open (as stated clearly in line 128 and line 133), so $w_t^\ast$, $\theta_t^{\textup{KL}}$ and $\theta_t^{\ast}$, which exist and are unique, must be in the interior.
> >
> > [1] Van der Vaart, Aad W. Asymptotic statistics. Vol. 3. Cambridge university press, 2000.

---

> ### Comment · Reviewer_jkSD · 2025-08-04
>
> One last additional comment:
>
> Re: Re: Q3: It appears that neither of the examples (newsvendor and portfolio optimization) provided in your response satisfies all your assumptions. In particular, the newsvendor problem does not seem to admit an invertible hessian matrix and the distribution estimation problem in the portfolio optimization example does not admit an optimal solution in the interior of $\Theta$.

---

> > ### Author Response · Authors · 2025-08-06
> > **Further technical clarification**
> >
> > Re: Re: Q3: Both the newsvendor and the portfolio examples raised in our previous response satisfy our assumptions. With all due respect, we do not understand how the reviewer claims our examples violate the mentioned conditions, and we would appreciate some explanations from the reviewer. In particular,
> >
> > **Newsvendor problem**: As described in our previous response, we consider a cost function $c(w,z)=a^\top(w-z)^++d^\top (z-w)^+$ borrowed from [1] and for each product $j\in[d_z]$, the demand $z^{(j)}$ has a Gaussian distribution $N(j\theta,\sigma_j)$ where $\theta$ is unknown and needs to be learned. In this example, the Hessian matrix can be readily deduced to be a diagonal matrix, where the $j$-th diagonal term of the matrix is $V_{jj}=(a_j+d_j)p_{\theta}^j(w)$, with $a_j$ and $d_j$ denoting the entries of the vectors $a$ and $d$ respectively and $p_{\theta}^j(\cdot)$ denoting the probability density function of $N(j\theta, \sigma_j)$. These diagonal entries $V_{jj}$ are all positive since $a_j>0, d_j>0$ and the density function of normal distribution is always positive. Therefore the Hessian matrix is positive definite and invertible.
> >
> > **Portfolio optimization problem:** As described in our previous response, the cost function is $c(w, z) = \frac{1}{2} w^\top (z - \mathbb{E}[z])(z - \mathbb{E}[z])^\top w - w^\top z + \frac{1}{2} w^\top w$, where $z \in \mathbb{R}^3$ denotes the assets’ return and follows a multivariate Gaussian distribution $z \sim N(B(\theta), \Gamma)$ where $\theta \in \mathbb{R}$ is an unknown scalar parameter, $B(\theta) = (\theta, 2\theta, 3\theta)^\top \in \mathbb{R}^3$, and $\Gamma \in \mathbb{R}^{3 \times 3}$ is a known, positive definite covariance matrix. In this problem, the solution can be readily derived to be $w_\theta=(\Gamma+I)^{-1}B(\theta)$ ($I$ is the identity matrix).  Since this is an unconstrained problem, i.e., the feasible region is $\Omega=\mathbb{R}^3$, the solution is clearly in the interior. Similarly, in the parameter space $\Theta=\mathbb{R}$, which is open, every distribution parameter $\theta\in\Theta$, including $\theta_0$, $\theta_n^\textup{KL}=\arg\max_{\theta\in\Theta}\mathbb{E}\_{Q^n}[\frac{1}{n}\sum_{i=1}^n\log p_\theta(z_i)]$ and $\theta_n^\ast=\arg\min_{\theta\in\Theta}\mathbb{E}\_{Q^n}[\frac{1}{n}\sum_{i=1}^n c(w_{\theta}, z_i)]$ lie in the interior of $\Theta$.
> >
> > [1] Elmachtoub, Adam N., Henry Lam, Haofeng Zhang and Yunfan Zhao. "Estimate-Then-Optimize versus Integrated-Estimation-Optimization versus Sample Average Approximation: A Stochastic Dominance Perspective." arXiv preprint arXiv:2304.06833 (2023).

---

> ### Comment · Reviewer_jkSD · 2025-08-06
>
> Again, I appreciate the author's time and effort in carefully responding to my questions and concerns.
>
> This reviewer remains concerned that the presentation of Assumption 3 and the implicit assumptions claimed in the authors’ response, among others, reflect room for improving the rigor in the analysis and for adding essential expositions in the presentation. The reviewer is also concerned that the applicability of the results is either limited or not clearly articulated, due to the seemingly restrictive nature of the assumptions.
>
> **Re: Portfolio optimization problem:** In view of the authors' response, the authors might revisit the proof of Lemma 5 using their example, where a linear cost function is involved. In this special case, the step of setting the gradient to zero appears problematic and warrant reconsideration.

---

> > ### Author Response · Authors · 2025-08-06
> > **Further Technical Clarification**
> >
> > Thank you for the further comment. As mentioned in our previous response, as the reviewer insists, we are happy to introduce an extra notation $\Xi$ to denote the space of $\zeta$ in Assumption 3 so that the statements hold for $\zeta\in\Xi$. Similarly, we are happy to add a clarification, right after the statement in each of Lemmas 4 and 5, to note to the reader that our definition of $w_t$, or $\theta_t^\textup{KL}$ and $\theta_t^*$, implies these quantities exist and are unique. We hope these completely address the concerns of the reviewer.
> >
> > For the portfolio optimization example we raised, the cost function is obviously quadratic and nonlinear. There is not even a special case where the cost function is linear (if this is what the reviewer was thinking).

---

> ### Comment · Reviewer_jkSD · 2025-08-06
>
> Thank you. Based on the author's rebuttal, $\mathbb{E}_\theta[c(w, z)] = \frac{1}{2} w^\top (\Gamma + I) w - w^\top B(\theta)$, which is linear (affine) in $\theta$.

---

> > ### Author Response · Authors · 2025-08-08
> > **Further Technical Clarification about Portofolio Optimization, and Beyond (1/2)**
> >
> > We again thank the reviewer for the reply and continued discussion.
> >
> > The reviewer raised that the objective function $\mathbb{E}\_{P\_\theta}[c(w,z)]$ is a linear function of $\theta$ in the portfolio optimization example and, in the previous round's comment, questioned the validity of setting its gradient to zero in Lemma 5. If we interpret correctly, then, with respect, we think that the reviewer might have misunderstood the IEO method. As described in line 163-165 (with more explanations in line 284-285), IEO solves $\min\_{\theta\in\Theta}\frac{1}{n}\sum_{i=1}^nc(w_{\theta},z_i)$, where 1) $z_i$ are i.i.d. with the ground truth distribution $Q_t$ (NOT $P_{\theta}$); 2) $w_{\theta}$ is the oracle solution that solves $\min_{w\in\Omega}\mathbb{E}\_{P\_\theta}[c(w,z)]$ defined in Equation (2). This obtains $\theta^{{IEO}}$ and hence the decision $w_{{IEO}}$. In this pipeline, we need to solve $\min_{\theta\in\Theta}\frac{1}{n}\sum_{i=1}^nc(w_{\theta},z_i)$, but never solve $\min_{\theta\in\Theta}\mathbb{E}\_{P_\theta}[c(w,z)]$. Nor does any of our mathematical results or analyses require solving $\min_{\theta\in\Theta}\mathbb{E}\_{P_\theta}[c(w,z)]$.
> >
> > We would also like to clarify regarding the Hessian matrix $\nabla\_\theta\mathcal{G}\^\ast(\theta_0,0)=\nabla\_{\theta\theta}\mathbb{E}\_{\theta_0}[c(w_{\theta_0},z)]=\Phi$ in the series of equations between line 928-929 in Lemma 5, in case this is the part that has confused the reviewer. This matrix  $\Phi=\nabla\_{\theta\theta}\mathbb{E}\_{\theta_0}[c(w_{\theta_0},z)]$ is $\nabla\_{\theta_1\theta_1}\int c(w_{\theta_1},z)p_{\theta_0}(z)dz$ at $\theta_1=\theta_0$, where the notation $\theta_0$ in the density function $p_{\theta_0}$ is fixed and is not a variable. This has been mentioned explicitly in line 240 in the main body. To avoid any misunderstanding in the proof of Lemma 5, we will add a recall of our definition from line 240 in that series of equations. Moreover, to further alleviate concerns of the reviewer about our implicit assumptions, we will add Assumptions 3 and 4 back at the beginning of Lemma 5 to make all assumptions explicit. We thank the reviewer for all the valuable related comments.
> >
> > Next, we would make two additional response items to even further address the reviewer's previous comments. First regards the verification of solution existence/uniqueness and being in the interior asked by the reviewer. Here, we verify directly from scratch the existence, uniqueness, and interiority of $\theta_t^{KL}$ and $\theta_t^\ast$ for the portfolio optimization example. In particular, we can do so even though $\mathbb{E}\_{P_\theta}[c(w,z)]$ is a linear function of $\theta$. To execute, we first verify $\theta_t^\ast$ that minimizes $\mathbb{E}\_{Q_t}[c(w_{\theta}, z)]$ exists and is unique, where the expectation is with respect to the ground truth distribution $Q_t$, not the distribution model class $P_{\theta}$. Recall that the distribution model class $P\_{\theta}$ is $N(B(\theta), \Gamma)$. As our previous response wrote, we know $w_{\theta}=(\Gamma+I)^{-1}(\theta, 2\theta, 3\theta)^\top$. For the ease of exposition, we denote $b=(1,2,3)^\top$ so that $B(\theta)=\theta b$. The objective function now is $\mathbb{E}\_{Q_t}[c(w_{\theta},z)]=\frac{1}{2}w_{\theta}\mathbb{V}\_{Q_t}[z]w_\theta-w^\top\mathbb{E}\_{Q_t}[z]+\frac{1}{2}w_{\theta}^\top w_{\theta}$. Note that here the variance $\mathbb{V}\_{Q_t}(z)$ is not necessarily equal to $\Gamma$, since $P_{\theta}$ is our model, not necessarily the ground truth ${Q_t}$. Similarly, $\mathbb{E}\_{Q_t}[z]$ is not necessarily equal to $\theta b$ (moreover, $\mathbb{E}\_{Q_t}[z]$ is not a function of $\theta$). By plugging in $w_\theta$, we have that $\mathbb{E}\_{Q_t}[c(w_{\theta},z)]=\frac{1}{2}\theta^2b^\top(\Gamma+I)^{-1}(\mathbb{V}\_{Q_t}[z]+I)(\Gamma+I)^{-1}b-\theta b^\top(\Gamma+I)^{-1}\mathbb{E}\_{Q_t}[z]$ is a unconstrained quadratic optimization problem. In this case, setting the gradient to zero with respect to $\theta$ is valid in optimizing over $\theta$. Moreover, the optimal solution $\theta_t^\ast$ exists, is unique and in the interior of $\Theta=\mathbb{R}$. Along a similar vein, we also verify $\theta_t^{{KL}}$ that maximizes $\mathbb{E}\_{Q_t}[\log p_{\theta}(z)]$ exists and is unique. Note that $\mathbb{E}\_{Q_t}[\log p_\theta(z)]=\text{constant} - \frac{1}{2} \mathbb{E}\_{Q_t}[(z - \theta b)^\top \Gamma^{-1}(z - \theta b)] =\text{constant} - \frac{1}{2} (\mathbb{E}\_{Q_t}[z^\top \Gamma^{-1} z] - 2\theta b^\top \Gamma^{-1} \mathbb{E}\_{Q_t}[z] + \theta^2 b^\top \Gamma^{-1} b)$ and is a quadratic function with respect to $\theta$. Hence the optimal solution $\theta_t^{{KL}}$ exists and is unique and in the interior of $\Theta=\mathbb{R}$. Thus, the existence, uniqueness and interiority of the solution apply to the portfolio optimization setting despite the linearity of $v(w, \theta)$ with respect to $\theta$ (what we need is the nonlinearity of $v(w, \theta)$ with respect to $w$).

---

> > ### Author Response · Authors · 2025-08-08
> > **Further Technical Clarification about Portofolio Optimization, and Beyond (2/2)**
> >
> > Our second additional response item regards the reviewer's concern on the restrictiveness of our assumption, which we hope to further alleviate here. In fact, we can remove the invertibility assumption of the total Hessian matrix of $v(w, \theta)$ defined in line 232, consisting of $V, \Sigma, \Sigma^\top, \ast$, where $v(w, \theta)$ is the function $\mathbb{E}\_{\theta}[c(w,z)]$ mentioned by the reviewer. That is, we only need the full rankness of $\Sigma$ and the invertibility of $V$, and we do not require properties of the $\ast$ submatrix (as we mentioned in Assumption 1 that this part is not of interest). The reason is that, in all our results and proofs throughout the entire paper, we have never used the ``invertibility" of the total Hessian matrix. This is because in our IEO method and other analysis, we have never jointly minimized $w$ and $\theta$ with respect to the function $v(w, \theta)$ and we have never minimized $\theta$ with respect to the function $v(w, \theta)$. Hence, we do not need the corresponding second-order optimality conditions, e.g., the positive definiteness or invertibility of the total Hessian matrix or the $\ast$ sub-matrix. On the other hand, in both ETO and IEO, we need the oracle decision $w_\theta$ to be well-defined. If we fix $\theta$, then $w_{\theta}$ is an optimal solution to the function $v(w,\theta)$ with respect to $w$. We therefore need the first- and second-order optimality conditions to hold for this minimization problem, i.e., the Hessian matrix $V=\nabla_{ww}v(w, \theta)$ needs to be positive definite, thus invertible, which is stated in Assumption 1.  The previous newsvendor and portfolio optimization examples that we elaborated satisfy the twice differentiability, the invertibility of $V$, and the full rankness of $\Sigma$. We hope this helps alleviate further the reviewer's concern on the restrictiveness of our assumptions. Moreover, we would like to once again thank the reviewer, since your valuable comment has driven us to investigate further to remove this unneeded invertibility of the total Hessian matrix in Assumption 1 that makes our results more generally applicable.

---

> > > ### Comment · Reviewer_jkSD · 2025-08-08
> > >
> > > Thank you for your careful response. I will raise the score accordingly.

---

> > > > ### Author Response · Authors · 2025-08-09
> > > > **Thank you!**
> > > >
> > > > We thank again the reviewer for all the very thoughtful comments and discussions which help improve our work. We are also glad that our responses are helpful, and would like to thank the reviewer for increasing our score.

---

### Official Review · Reviewer_JYdJ · 2025-07-23

**Clarity:** 3
**Significance:** 2
**Originality:** 3
**Rating:** 4
**Confidence:** 3

**Summary:**

This work compared SAA, ETO and IEO performances from the bias-variance tradeoff perspectives, under the setting of local model misspecification. They provided the ordering comparison results of bias, variance and regret under balanced, severe and mild misspecification scenarios, respectively.

**Questions:**

1. This work is more about the asymptotic results on the comparison, and you mentioned you leveraged tools from asymptotic statistics, is it possible to extend the current results into finite-sample regime based on current toolsets, or whether it is an important problem?

**Ethical Concerns:**

["NO or VERY MINOR ethics concerns only"]

**Quality:**

3

**Strengths And Weaknesses:**

Strength:
1. The work has a comprehensive comparison on the performance of SAA, ETO and IEO in data-driven optimization
2. The work leverage several tools from statistics, which should be uncommon in the literature and is novel in terms of the techniques.
3. The writting of the paper looks good to me.

Weakness:
1. Some assumptions can be further rationalized:
   - Line 173, why do you require the minimizer to be in the interior of the space? I am wondering in the common linear programming cases, the solution should be at a vertex, and the assumption may not hold (also uniqueness due to common nonconvexity in ML objective functions)?
   - Line 178, you assumed Q could deviate from the model in a certain “direction", I am wondering whether it is a reasonable assumption, whether it is implementable in real applications, is there any approach to detect the direction.
2. Comparison with existing work. A closely related work should be Elmachtoub et al. [2025] as you cited, they divided their discussion by the dgree of misspecification $B_0$, and here authors divided the discussion by the magnitude of $t$ (exponent $\alpha$), which looks similar to me, just like set $B_0=1/n^\alpha$ (is that true?). So I am confused why do you say "there is no smooth transition in between that captures the impact of varying the misspecification amount." in Line 57. Even though authors mentioned the direction $u$ may take effect, but Theorem 5 seems to work in some specical case. So I hope to have more illustrations on the improvement from the techniques you employed.

Typo:
   - Line 285, what is $O_{Q^n}$, also why there is no norm for the three errors?

---

> ### Author Rebuttal · Authors · 2025-07-30
>
> Thank you for your recognition of our significance and strengths, and your detailed review. We address your comments and questions as follows:
>
> W1.1: Let us respond in three parts:
> 1) Our paper focuses on non-linear cost objectives, and our assumptions are in line with the standard statistics [2, 3] and optimization literature [4, 5, 6] that presume openness of the decision space and hence that an optimal solution is an interior point. Note that this assumption can be relaxed by adding equality constraints and inequality constraints -- We expect analogous results hold by utilizing KKT conditions, but the conceptual insights would be similar while adding significant technical complexity, and hence we leave this extension as future work.
> 2) The uniqueness assumption is also standard in nonlinear optimization; See [4, 5, 6, 7] (some of them even assume strong convexity). Note that this assumption may be weaker than you would initially think, because the uniqueness applies to the expected cost, not the cost itself. For example, in the newsvendor problem, a classic example in operations research, the cost function itself is piecewise linear and is not strongly convex. However, the expected cost is indeed strongly convex if the distribution of customer demand is continuous and has a lower bounded density, therefore, the optimal solution becomes unique.
> 3) In fact, the linear programming setting raised by the reviewer is easier than our nonlinear setting from a statistical perspective. This is because when the cost function is linear, it suffices to predict the expectation of the cost vector, which is a point prediction [1]. Because of this, it is not necessary to fit the whole distribution, as we must. In contrast, our developments require the notion of model misspecification that is at the distributional level, since the nonlinearity of the objective function forces a more complex relation between the randomness and the decision variables.
>
> W1.2: Yes, the assumption on $Q$ is reasonable. The form originates from the local minimax theory and semi-parametric statistics in [2] (Section 25, Eq.(25.13)). In other words, it is a smoothness condition saying that the distribution $Q_t$ is differentiable at $t=0$ with score function $u$ in the $L_2$ norm sense. This condition plays a central role in ensuring that the log-likelihood ratio behaves locally like a quadratic form, which is the cornerstone of the local asymptotic normality (LAN) framework. On the other hand, in practice this misspecification direction is not known (otherwise, if it is known to the modeler, it would have been incorporated in the fitting so that the misspecification is reduced). Nonetheless, we highlight that, as  a strength of our main results, the orderings of the bias and variance of the three methods are universal regardless of (almost) all directions $u(\boldsymbol{z})$, but only depend on the degree of misspecification. That is, the deviating directions $u$ of $Q$ turn out to be an invariant factor under most scenarios.
>
> W2: First, note that the statement "there is no smooth transition in between that captures the impact of varying the misspecification amount." in Line 57 refers to [5], not [8]. Indeed, [5] divides the analysis into either misspecified or well-specified cases and, unlike the current work, there is no notion of ``local misspecification" there that allows the ground-truth data distribution and the model to be close in a certain asymptotic sense.
>
> Now, as the reviewer rightly points out, [8] has a similar favor as us in that they consider the degree of misspecification. However, the derived bounds of [8] are opaque and loose. Specifically, their degree of misspecification is denoted by $\delta$ and $B_0$, which is problem and method dependent, is not interpretable and just for technical convenience. They use the so-called Berry-Esseen bounds, a uniform tool, to compare the performance between ETO and IEO by tail probability bounds, thus making the analysis coarse. Even though they can conclude the partial ordering between IEO and ETO, the involved conditions including the threshold for $B_0$ and $\delta$ in Assumption 4 and Corollary 1 are difficult to understand. In contrast, and in fact as a remedy to [8], we introduce the notion of local misspecification that allows us to explicitly derive asymptotic limits, dissect the bias and variance, and even understand the impact from the directions of misspecification that are well beyond [8].
>
> Finally, we cannot set $B_0=1/n^\alpha$ in [8] because $B_0$ there is assumed (and required by the analysis) to be a fixed number that cannot be changed with respect to the sample size $n$. In our framework, however, we allow  dynamics where, roughly speaking, the distance between the ground truth distribution $Q$ and the model $P_{\theta}$ has a comparable order of magnitude with the sampling error in estimating the parameter $\theta$. This, together with our developed asymptotic framework, empowers us to dissect bias-variance accurately, especially in cases where these errors are comparable (the interesting case), as well as directional impacts.
>
> Q1: As explained in our response to W2 above, finite-sample results are interesting, but like in [8], they are anticipated to be intricate and do not offer the clean analysis that allows us to dissect bias-variance and the impact of misspecification directions. We do understand that there are merits for finite-sample bounds (just like in other ML problems), but at the same time it is also the case that these bounds very often are conservative and loosely depend on model factors. A tight enough finite-sample analysis that allows us to dissect bias-variance tradeoff meaningfully would be very interesting. Lastly, please kindly refer to our response to Reviewer JBev on our new experimental results that validate our asymptotic theory and its demonstration in finite-sample manifestation.
>
> Typo: Yes we will add the norm for the three terms in our final version.
>
> [1] Hu, Yichun, Nathan Kallus, and Xiaojie Mao. "Fast rates for contextual linear optimization." Management Science 68.6 (2022): 4236-4245.
>
> [2] Van der Vaart, Aad W. Asymptotic statistics. Vol. 3. Cambridge university press, 2000.
>
> [3] Shao, Jun. Mathematical statistics. Springer Science \& Business Media, 2008.
>
> [4] Duchi, John C., and Feng Ruan. "Asymptotic optimality in stochastic optimization." (2021): 21-48.
>
> [5] Elmachtoub, Adam N., Henry Lam, Haofeng Zhang and Yunfan Zhao. "Estimate-Then-Optimize versus Integrated-Estimation-Optimization versus Sample Average Approximation: A Stochastic Dominance Perspective." arXiv preprint arXiv:2304.06833 (2023).
>
> [6] Nocedal, Jorge, and Stephen J. Wright, eds. Numerical optimization. New York, NY: Springer New York, 1999.
>
> [7] Qi, Meng, Paul Grigas, and Zuo-Jun Max Shen. ``Integrated conditional estimation-optimization.'' arXiv preprint arXiv:2110.12351 (2021).
>
> [8] Elmachtoub, Adam N., Henry Lam, Haixiang Lan, and Haofeng Zhang. "Dissecting the Impact of Model Misspecification in Data-driven Optimization." arXiv preprint arXiv:2503.00626 (2025).

---

> > ### Comment · Reviewer_JYdJ · 2025-08-07
> >
> > Thank you for the response, which resolves my questions, I will keep the score.

---

> > > ### Author Response · Authors · 2025-08-09
> > > **Thank you!**
> > >
> > > We sincerely thank the reviewer again for your valuable comments and positive view on our work, and are glad that our response has helped resolve your questions.

---

### Decision · Program_Chairs · 2025-09-17

**Decision:**

Accept (poster)

**Comment:**

This paper provides a new analysis studying the impact of model misspecification on the asymptotic performance of sample average approximation, estimate-then-optimize, and integrated estimate-then-optimize.

Pros:
- This paper provides a novel and significant view on model misspecification in an important broader context
- The analysis provides interpretable insights that are likely to be useful to practitioners

Weakness:
- While the paper is a theory paper, the empirical evaluation could be stronger. The authors added an empirical evaluation focused on finite-time performance during the review process but the paper would be stronger if there were additional investigations.

I'm recommending acceptance because of the novelty, significance, and usefulness of the theory

There was a robust conversation between the reviewers and authors that will improve the paper's clarity, level of rigor, and the strength of its empirical evaluation. Reviewer comments focused on clarifying a number of technical points in the paper and on the lack of empirical evaluation.